# Caterpillar GNN: Replacing Message Passing with Efficient Aggregation

## Abstract

Message-passing graph neural networks (MPGNNs) dominate modern graph learning. Typical efforts enhance MPGNN's expressive power by enriching the adjacency-based aggregation. In contrast, we introduce an *efficient aggregation* over walk incidence-based matrices that are constructed to deliberately trade off some expressivity for stronger and more structured inductive bias. Our approach allows for seamless scaling between classical message-passing and simpler methods based on walks. We rigorously characterize the expressive power at each intermediate step using homomorphism counts over a hierarchy of generalized *caterpillar graphs*. Based on this foundation, we propose Caterpillar GNNs, whose robust graph-level aggregation successfully tackles a benchmark specifically designed to challenge MPGNNs. Moreover, we demonstrate that, on real-world datasets, Caterpillar GNNs achieve comparable predictive performance while significantly reducing the number of nodes in the hidden layers of the computational graph.

## 1 Introduction

Graphs are a powerful structure, capable of representing relational information across various domains such as biology, chemistry, databases, or social sciences. Graph inference carries variability in that its structure is governed by the underlying distribution, unlike inference on sequential text or gridded images. The established incorporation of this variability relies on the inductive bias of (equivariant) message-passing (MP) in graph neural networks (MPGNNs). Prior work has shown the limits of MP in capturing structural biases (Xu et al., 2019; Morris et al., 2019). Consequently, MPGNNs may suffer from restricted expressivity, leading to many extensions of MP. On the other hand, MPGNNs may also fail to learn properly due to phenomena such as nodal over-smoothing (Oono and Suzuki, 2020) and over-squashing (Alon and Yahav, 2021).

Namely, aggregation in MP causes a bottleneck that subsequent work mitigates by modifying the graph topology, e.g., rewiring (Topping et al., 2022; Di Giovanni et al., 2023). We consider an alternative walk incidence-based topology that reveals another kind of bottleneck. Guided by this topology, we construct a benchmark that empirically uncovers the consequent limitation of MPGNNs. Surprisingly, our benchmark only requires small unlabeled acyclic graphs that seem nearly trivial to distinguish from an expressivity standpoint.

To study such disparities between topology and lower expressivity, we rely on a more algebraic definition of expressive power. Concretely, the expressivity of some architectures extending MPGNN can be bounded using graph homomorphism counts over a restricted class of graphs $\mathcal{F}$ (see Table 1). In the limit, extending $\mathcal{F}$ from trees upwards to all graphs yields the maximum equivariant expressivity, namely, graph isomorphism, as shown by Lovász (1967). *Our work answers the converse:* which inductive biases arise when $\mathcal{F}$ is restricted downwards to subclasses of trees, such as *caterpillars*? A caterpillar is a tree graph for which deleting its leaves yields a path.

| MPGNN extension | Bound over $\mathcal{F}$ |
|---|---|
| Vanilla[a] | trees[g] |
| Higher ($k$) order[b] | treewidth[g] ($\leqslant k$) |
| $\mathcal{P}$-enabled[c] | $\mathcal{P}$-pattern trees[c] |
| Subgraph agg.[d,e] | apex trees[h] |
| Spectral inv.[f] | parallel trees[i] |
| Caterpillar (ours) | caterpillars |

Table 1: [a]Gilmer et al. (2017), [b]Morris et al. (2019), [c]Barceló et al. (2021), [d]Qian et al. (2022) [e]Frasca et al. (2022), [f]Zhang et al. (2024), [g]Dvořák (2010), [h]Rattan and Seppelt (2021), [i]Gai et al. (2025).

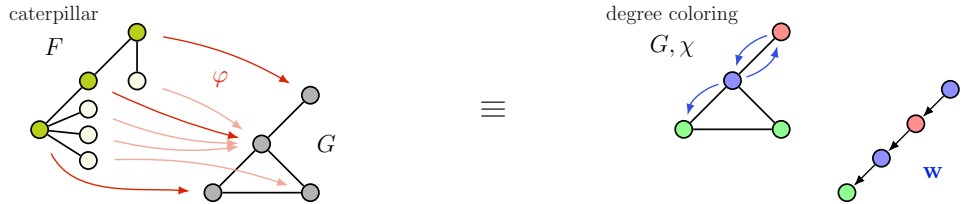

Figure 1: Counting over either of the two depicted notions is exactly as expressive on graphs. Left: *A graph homomorphism* $\varphi\colon F \to G$, where $F$ is a caterpillar graph. Right: *An occurrence of a colored walk* $\mathbf{w} =$ brbg in the same graph $G$ with a coloring $\chi$ according to vertex degrees.

**Theorem 4.1** (informal, example case). *For an input graph $G$, counting homomorphisms to $G$ over caterpillars is exactly as expressive as* coloring $G$ according to vertex degrees and then counting colored walks. *See Figure 1 for illustration and Section 4 for the formal statement.*

While the above characterization clarifies the notion of lower expressivity, the number of colored walks grows exponentially in the worst case. As a result, existing architectures process sequential patterns in graphs (Tönshoff et al. (2023); Zeng et al. (2023); Chen et al. (2024)) by random-walk sampling, which sacrifices equivariance. To retain equivariance, we propose efficient aggregation (EA), improving the asymptotical complexity to polynomial upon the exponential enumeration, and achieving the desired expressivity, which we can parametrize. Subsequently, we introduce Caterpillar GNNs, incorporating EA in the same way that MPGNN incorporates MP. Caterpillar GNNs pioneer the study of performance under lower expressivity. Within our motivating benchmark, we find that a less expressive inductive bias mitigates the bottleneck of information alignment, whereas increasing expressivity further degrades performance. Moreover, EA may downscale the computation graph after each layer, in which is reminiscent of downscaling in convolutional neural networks. Main contributions are as follows:

- We introduce EA (Section 3). We prove its tractability (Theorem 3.1) and desired expressivity (Theorem 4.2). The challenge of its complete derivation and proofs we address by developing techniques in automata theory (Appendix A).
- We characterize expressivity of EA using a hierarchy of *generalized caterpillar graphs* and its graph homomorphism counts (Section 4, Theorem 4.1). For this, we develop novel combinatorial arguments in graph theory (Appendix B).
- We incorporate EA into Caterpillar GNNs (Section 3.2, Eq. (5)), and investigate its parametric scaling (Section 3.3). Using walk incidence-based topology (Section 5.1), we illustrate that the effect of stronger inductive bias can outweigh lower expressivity.

Empirically, we investigate how parametric scaling of EA impacts the dataset-specific tradeoffs between performance and nodal efficiency on real-world tasks. Enabled for such tradeoffs, Caterpillar GNNs achieve comparable performance while using fewer nodes of the computational graph (Fig. 13).

## 2 PRELIMINARIES

Let $G = (V, E)$ be an undirected graph with a finite vertex set $V$ and an edge set $E \subseteq V^2$. Loops are not assumed, and an edge between $u$ and $v$ is denoted by $uv$. The degree of a vertex $u$ is $\deg(u) = |\{v \mid uv \text{ in } E\}|$, and $n = |V|$. A path is a connected acyclic graph with vertices of degree at most two. We denote the class of all paths by $\mathcal{P}$, and by $\mathcal{P}_t \subset \mathcal{P}$ the subclass of paths of length at most $t$ where length means $|E|$. A tree is a connected acyclic graph. We denote by $\mathcal{T}$ the class of all trees. By $\mathcal{T}^{\bullet}$, we mean the class of rooted trees, and by $\mathcal{T}_h^{\bullet} \subset \mathcal{T}^{\bullet}$, the class where every vertex is at distance at most $h$ from the root, that is, at most $h$ edges from a root (e.g. Diestel (2025, page 8)).

Multisets are represented using symbols $\{\!\{$ , and $\}\!\}$. Let $X, Y$ be sets, and $x$ in $X$, $y$ in $Y$. The family of all multisets of elements from $X$ is denoted by $\mathbb{N}^X$. For a multiset $m$ in $\mathbb{N}^X$, we access multiplicity of $x$ by $m[x]$. For a vector $\boldsymbol{v}$ in $\mathbb{R}^X$, we access its $x$-th component by $\boldsymbol{v}[x]$. For a matrix $\boldsymbol{M}$ in $\mathbb{R}^{X \times Y}$ (of shape $X \times Y$), we access its entries by $\boldsymbol{M}[x, y]$, rows by $\boldsymbol{M}[x]$ and columns by $\boldsymbol{M}[-, y]$. Finally, the notation $[k]$ stands for the set $\{1, 2, \ldots, k\}$ for $k \in \mathbb{N}$. For the graph $G$, we denote its *adjacency matrix* by $\boldsymbol{A}$ in $\mathbb{R}^{V \times V}$, that is, $\boldsymbol{A}[u, v] = 1$ if $uv$ in $E$ and 0 otherwise. Its *identity* or *self-loop matrix* is denoted by $\boldsymbol{I}$ in $\mathbb{R}^{V \times V}$, and the all-ones vector by $\mathbf{1}$ in $\mathbb{R}^{V \times 1}$.

**MPGNNs.** In what follows, we often represent graphs by matrices and hence adopt a specific notation. For a matrix $\boldsymbol{M}$ in $\mathbb{R}^{X \times Y}$, features of $d$ channels $\boldsymbol{h}$ in $\mathbb{R}^{Y \times d}$ and $x$ in $X$, we define

$$\text{mult}(x, \boldsymbol{M}, \boldsymbol{h}) := \{\!\!\{ (\boldsymbol{M}[x,y], \boldsymbol{h}[y]) \mid y \text{ in } Y, \ \boldsymbol{M}[x,y] \neq 0 \}\!\!\}.$$

Let $\boldsymbol{h}_{\text{MP}}^{(0)}$ be features in $\mathbb{R}^{V \times d}$ if given and $\mathbf{1}$ otherwise. Then we define MPGNN of $L$ layers for each $u$ in $V$ and integer $\ell$ such that $0 \leqslant \ell < L$ as follows

$$\boldsymbol{h}_{\text{MP}}^{(\ell+1)}[u] = \text{UPDATE} \left( \text{mult}(u, \boldsymbol{I}, \boldsymbol{h}_{\text{MP}}^{(\ell)}), \ \text{AGG} \left( \text{mult}(u, \boldsymbol{A}, \boldsymbol{h}_{\text{MP}}^{(\ell)}) \right) \right),$$

$$\boldsymbol{h}_{\text{MP}} = \text{READOUT} \left( \text{mult}(0, \tfrac{1}{n}\mathbf{1}^{\top}, \boldsymbol{h}_{\text{MP}}^{(L)}) \right),$$

where we use $\text{mult}$ to self-loop with $\boldsymbol{I}$, and to range over adjacent nodes with $\boldsymbol{A}$ of shape $V \times V$, and to collect all nodes with $\tfrac{1}{n}\mathbf{1}^{\top}$ of shape $\{0\} \times V$. The functions AGG, UPDATE, and READOUT are specific to each layer. We omit their indexing and learnable parameters for readability.

**Expressivity and homomorphism counts.** Let $\mathcal{G}$ denote the class of all graphs and let f and g be two functions on $\mathcal{G}$. Then the function f *is at least as expressive as* g, denoted by $\mathsf{f} \sqsupseteq \mathsf{g}$, if for every two graphs $G$ and $H$ holds that $\mathsf{f}(G) = \mathsf{f}(H)$ implies $\mathsf{g}(G) = \mathsf{g}(H)$. Next, f *is (exactly) as expressive as* g, denoted by $\mathsf{f} \equiv \mathsf{g}$, if $\mathsf{f} \sqsupseteq \mathsf{g}$ and $\mathsf{g} \sqsupseteq \mathsf{f}$. Furthermore, f *is (strictly) more expressive than* g, denoted by $\mathsf{f} \sqsupsetneq \mathsf{g}$, if $\mathsf{f} \sqsupseteq \mathsf{g}$ and $\mathsf{f} \not\equiv \mathsf{g}$. The expressivity relation is a partial ordering on the family of functions on $\mathcal{G}$.

For a source graph $F = (V_s, E_s)$, a function $\varphi \colon V_s \to V$ is a *graph homomorphism* $F \to G$ if every edge $uv$ in $E_s$ implies edge $\varphi(u)\varphi(v)$ in $E$. See Figure 1. For a class of source graphs $\mathcal{F} \subseteq \mathcal{G}$, we define a (possibly infinite) vector of *homomorphism counts over* $\mathcal{F}$, denoted by $\hom(\mathcal{F}, G)$ in $\mathbb{N}^{\mathcal{F}}$, as $\hom(\mathcal{F}, G)[F] = |\{\varphi \mid \varphi \colon F \to G\}|$ for all $F$ in $\mathcal{F}$. Note that every class $\mathcal{F} \subseteq \mathcal{G}$ induces the function $\hom(\mathcal{F}, -) \colon \mathcal{G} \to \mathbb{N}^{\mathcal{F}}$, assigning $\hom(\mathcal{F}, G)$ to the target graph $G$. It always holds that $\mathcal{F} \supseteq \mathcal{F}'$ implies $\hom(\mathcal{F}, -) \sqsupseteq \hom(\mathcal{F}', -)$.

**Graph colorings and color refinement.** A coloring $\chi$ is a map that assigns specific colors to the vertices. Formally, for each graph $G = (V, E)$, we have a function $\chi(G, -) \colon V \to \Sigma'$ where $\Sigma'$ denotes a color set. We say coloring $\chi$ is a $\Sigma$-*coloring on* $G$ if $\Sigma' \subseteq \Sigma$. A coloring example is the trivial coloring $\chi_{\text{triv}}$, which assigns 0 to every vertex $u$ of $G$, $\chi_{\text{triv}}(G, u) = 0$. Therefore, $\chi_{\text{triv}}$ is a $\{0\}$-coloring on every graph. Another "extreme" is the *identity coloring* $\chi_{\text{id}}$ assigning identities on vertices, $\chi_{\text{id}}(G, u) = u$ for vertex $u$, and thus $\chi_{\text{id}}$ is a $V$-coloring on $G$. The *degree coloring* $\chi_{\text{deg}}$, assigns degree to every vertex in $G$, which can be written as $\chi_{\text{deg}}(G, -) = \deg(-)$.

A *color refinement* constructs a sequence of graph colorings: $\chi_{\text{cr}}^{(0)}(G, u) = 1$, and for all $h \geqslant 0$ and each $u$ in $V$ as $\chi_{\text{cr}}^{(h+1)}(G, u) = (\chi_{\text{cr}}^{(h)}(G, u), \{\!\!\{ \chi_{\text{cr}}^{(h)}(G, v) \mid uv \text{ in } E \}\!\!\})$. Secondly, it defines a sequence of functions on graphs $\text{cr}^{(h)}(G) = \{\!\!\{ \chi_{\text{cr}}^{(h)}(G, u) \mid u \text{ in } V \}\!\!\}$; and, finally, the function on graphs: $\text{cr}(G) = \{\text{cr}^{(h)}(G) \mid h \text{ in } \mathbb{N}\}$. For instance, the first coloring is as expressive as the trivial: $\chi_{\text{cr}}^{(0)}(G, -) \equiv \chi_{\text{triv}}(G, -)$, and the second exactly as the degree coloring: $\chi_{\text{cr}}^{(1)}(G, -) \equiv \chi_{\text{deg}}(G, -)$.

Let $\chi$ be a $\Sigma$-coloring on $G$. A *walk* in $G$ is a sequence of vertices $v_1, v_2, \ldots, v_t$ such that $v_i v_{i+1}$ in $E$ for $i$ in $[t-1]$. A special case is a *path* in $G$, which is a walk with all vertices distinct. A *colored walk* is a word $\boldsymbol{a} = a_1 a_2 \ldots a_t$ such that $a_i = \chi(G, v_i)$ in $\Sigma$ for $i$ in $[t]$. At the same time, the sequence $v_1, v_2, \ldots, v_t$ is an *occurrence* of $\boldsymbol{a}$ in $G$. See Figure 1. We say that vertex $u$ is *incident* to colored walk $\boldsymbol{a}$ if $u = v_t$, and *adjacent* if $uv_t \in E$. We denote by $\Sigma^t$, resp. $\Sigma^{\leqslant t}$ the set of all words over $\Sigma$ of length exactly $t$, resp. at most $t$; and by $\Sigma^*$ the set of all words. Note that $\Sigma^0 = \{\lambda\}$ where $\lambda$ is the *empty word*.

## 3 EFFICIENT AGGREGATION: THE DEFINITION

This section introduces efficient aggregation (EA), the matrix-based replacement at the core of Caterpillar GNNs. EA is grounded in sequential graph patterns (Part I), but is formulated using layer-specific matrices (Part II) to provably aggregate these patterns. Part III is a short user-guide to scaling by a single height parameter controlling the strength of our inductive bias. Omitted proofs are given in Appendix A.

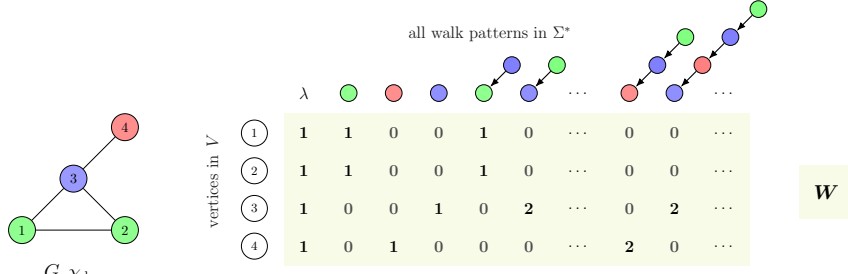

Figure 2: Left: Graph $G$ with vertices colored by $\chi_{\mathrm{deg}}$. Colors red, green and blue depict degrees 1, 2 and 3, respectively. Right: Walk incidence matrix $W$ of shape $V \times \Sigma^*$. The entry $W[3, \mathrm{gb}]$ equals 2, since vertex 3 terminates two occurrences of the colored walk gb in $G$.

### 3.1 PART I: SEQUENTIAL PATTERNS

For tractable incorporation of lower-order inductive biases, we innovate processing of sequential patterns, as motivated in Theorem 4.1 and analyzed further in Section 4. In the language of colored walks, many successful machine learning approaches first *sample* a tractable number of random walks and then process the visited colors as sequences with either kernels (Borgwardt et al., 2005; Kriege, 2022) or neural networks (Tönshoff et al., 2023; Zeng et al., 2023; Chen et al., 2024). Our approach is a fundamental *reversal* of these steps: given a prescribed colored walk, we count its occurrences.

Crucially, we consider a tractable and canonical subset of colored walks. As shown later (Theorem 4.2), this subset suffices to determine all other colored walks. In contrast to prior sampling-heavy methods, we preserve determinism, equivariance and intended expressivity. To formalize our reversal, we relate vertices and colored walks using incidence matrix.

**Walk incidences.** For a given graph $G$ with $\chi$ a $\Sigma$-coloring, and a given length $t \geqslant 0$, we define the *walk-incidence matrix* $W_t$ of shape $V \times \Sigma^t$ for each $u$ in $V$ and $\boldsymbol{a}$ in $\Sigma^t$ by

$$W_t[u, \boldsymbol{a}] \text{ is the number of occurrences of } \boldsymbol{a} \text{ that terminate in vertex } u. \quad (1)$$

Each column $W_t[-, \boldsymbol{a}]$ in $\mathbb{N}^V \subseteq \mathbb{R}^V$ corresponds to a *multiset of vertices* incident to colored walk $\boldsymbol{a}$. For instance, the column $W_1[-, c]$ coincides with vertices $u$ of color $c = \chi(G, u)$ in $\Sigma$. By *convention*, the empty walk is incident to every vertex, $W_0[u, \lambda] = 1$ for $u$ in $V$. See Figure 2, for an illustration of $W = [W_0 | W_1 | \cdots]$ of shape $V \times \Sigma^*$.

**Walk selection.** The row dimension $V$ of incidence matrices $W_t$ remains fixed, while the column dimension $\Sigma^t$ grows exponentially in $t$. We avoid this growth by selecting subsets of $\Sigma^t$, such that the induced columns of $W_t$ form a basis of the column space of $W_t$. The definition proceeds inductively: $S_0 = \{\lambda\} = \Sigma^0$, and for known $S_t$, the set $S_{t+1} \subseteq \Sigma^{t+1}$ satisfies the following conditions:

(i) for every $\boldsymbol{a}c$ in $S_{t+1}$ there is $\boldsymbol{a}$ in $S_t$ (*prefix-closedness*),

(ii) the columns of $W_{t+1}$ induced by $S_{t+1}$ are *linearly independent*, and

(iii) the set $S_{t+1}$ is *lexicographically minimal* among other sets satisfying (i) and (ii).

The last Condition (iii) together with $S_0 = \{\lambda\}$ ensures uniqueness, making this selection canonical. Condition (ii) implies the upper bound $|S_t| \leqslant \mathrm{rank}(W_t) \leqslant |V|$, reducing the representation of potentially $|V|^t$ columns to at most $|V|$. Finally, Condition (i) allows for a time-efficient algorithm reminiscent of breadth-first search with linear independence checking using matrix decomposition.

**Theorem 3.1.** *Let $\chi$ be a $\Sigma$-coloring on graph $G$ with $n$ vertices, and $T$ in $\mathbb{N}$ a limit then the canonical subsets $(S_t)_{t=0}^{T}$ defined above are computable in time $\mathcal{O}(Tn^{\omega+\epsilon})$ for any $\epsilon > 0$ where $\omega$ is the exponent of matrix multiplication.*[1]

### 3.2 PART II: EFFICIENT MATRICES

Up to this point, we have considered walks as sequences processed one by one. However, such a representation is inefficient, in particular because it ignores shared prefix structure, as is well-known

---

[1]As of 2025, $\omega \leqslant 2.371339$ (Alman et al., 2024). For instance, we can assume that $\omega + \epsilon \leqslant 2.372 < 3$.

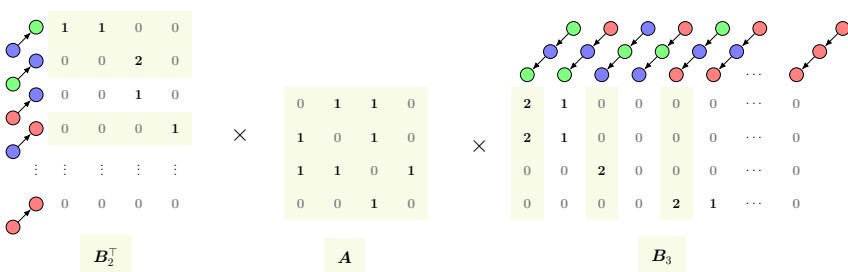

Figure 3: Locking the adjacency matrix $\boldsymbol{A}$ of the graph $G$ as in Figure 2. Matrix $\boldsymbol{B}_3$ of shape $V \times S_3$ is a submatrix of $\boldsymbol{W}_3$ (of shape $V \times \Sigma^3$) induced by highlighted columns, similarly, $\boldsymbol{B}_2^\top$ and $\boldsymbol{W}_2^\top$.

from string-searching algorithms (e.g., suffix trees (Weiner, 1973)). To overcome this inefficiency, we organize walk-incidence statistics into matrices, where selected colored walks correspond to columns and also rows. This matrix formulation enables hierarchical aggregation of walk patterns, analogous to how message passing (MP) aggregates over neighborhoods, but dropping the assumption of repeating fixed neighborhood structure at each layer.

MP on graph $G = (V, E)$ with $n$ vertices consists on two steps: aggregation via adjacency operator $\boldsymbol{A}$ and update via self-looping operator $\boldsymbol{I}$ both $\mathbb{R}^V \to \mathbb{R}^V$. We aim to deliberately restrict this repeating mechanism: informally, we lock corresponding vector space $\mathbb{R}^V$ by projecting those operators into $\mathbb{R}^{S_{t+1}} \to \mathbb{R}^{S_t}$ of possibly lower dimension as implied by Condition (ii).

For a $\Sigma$-coloring on $G$, and integer $t \geqslant 0$, we denote by $\boldsymbol{B}_t$ of shape $V \times S_t$ the submatrix of $\boldsymbol{W}_t$ (of shape $V \times \Sigma^t$) that keeps only the columns indexed by $S_t$ (see Figure 3 for an illustration). Condition (ii) guarantees that every matrix $\boldsymbol{B}_t$ is tractable and has full rank.

**Efficient aggregation.** Let $M$ be a matrix of shape $V \times V$, such as $\boldsymbol{A}$ or $\boldsymbol{I}$ above, then a $t$-th *efficient $M$-matrix* $\boldsymbol{C}_t^M$ of shape $S_t \times S_{t+1}$ is defined as

$$\boldsymbol{C}_t^M = (\boldsymbol{B}_t^\top \boldsymbol{B}_t)^{-1} \boldsymbol{B}_t^\top \, M \, \boldsymbol{B}_{t+1}, \tag{2}$$

which solves the least-squares problem $\arg\min_{\boldsymbol{C}} \| \boldsymbol{B}_t \boldsymbol{C} - M \boldsymbol{B}_{t+1} \|_F$. Informally, the unique operator $\boldsymbol{C}_t^M : \mathbb{R}^{S_{t+1}} \to \mathbb{R}^{S_t}$ is the best approximation of $M$ in the basis indexed by canonical $S_t$.

We call an *efficient aggregation (EA)* the collection of the first $n$ efficient adjacency and identity matrices into the graph invariant

$$\mathcal{I}_{\text{EA}}(G, \chi) = \left\{ \left( \boldsymbol{C}_t^A, \boldsymbol{C}_t^I \right) \mid 0 \leqslant t < n \right\}. \tag{3}$$

Since efficient matrices are indexed by colored walks, we compare $\mathcal{I}_{\text{EA}}$ directly across graphs. Its expressive power (Section 4) as the function on graphs $\mathcal{I}_{\text{EA}}(-, \chi)$ motivates the following model.

**Caterpillar GNNs.** We now describe how efficient matrices are used across $L$ layers. *Caterpillar GNN* initializes with $\boldsymbol{h}_{\text{EA}}^{(0)}(ac) = \text{REDUCE}(c, \{\}, \mathbf{1})$ for colored walk $ac$ in $S_L$. Then, at each layer $\ell$ such that $0 \leqslant \ell \leqslant L - 1$ with $t_\ell = L - \ell$ and for each colored walk $ac$ in $S_{t_\ell}$ we have

$$\boldsymbol{h}_{\text{EA}}^{(\ell+1)}[ac] = \text{REDUCE}\left(c, \, \text{mult}(ac, \boldsymbol{C}_{t_\ell}^I, \boldsymbol{h}_{\text{EA}}^{(\ell)}), \text{AGG}\left( \text{mult}(ac, \boldsymbol{C}_{t_\ell}^A, \boldsymbol{h}_{\text{EA}}^{(\ell)}) \right) \right), \tag{4}$$

$$\boldsymbol{h}_{\text{EA}} = \text{READOUT}\left( \text{mult}(\lambda, \boldsymbol{C}_0^I, \boldsymbol{h}_{\text{EA}}^{(L)}) \right). \tag{5}$$

In this definition, the function REDUCE replaces the usual UPDATE: it targets a colored walk $ac$ instead of a fixed vertex and requires color $c$ as an additional input. A visual side-by-side comparison with the standard MP is given in Figure 4.

### 3.3 Part III: Parametric Scaling

The vertex coloring $\chi$ controls the coarseness of distinguished colored walks and thus governs the resulting inductive bias of EA. In our approach to EA, we utilize colorings of the color refinement $\chi = \chi_{\text{cr}}^{(h)}$, simplifying the choice for end-users to the parameter $h \geqslant 0$ called *height*. To guide our exploration, we analyze two extreme cases: trivial coloring $\chi_{\text{triv}}$ and identity coloring $\chi_{\text{id}}$.

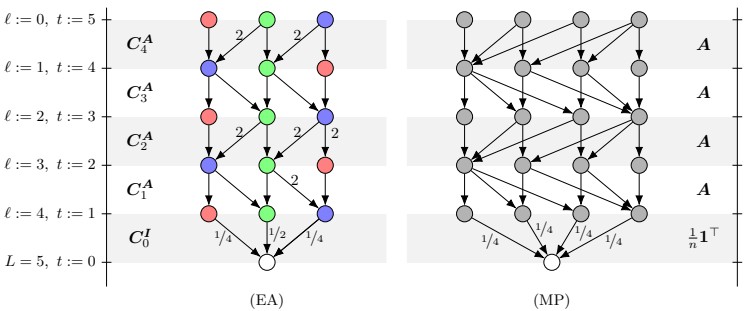

Figure 4: Comparison of computational graphs (without self-loops): (EA) *efficient aggregation* (ours), and (MP) *message-passing* for the graph $G$ and coloring $\chi_{\text{deg}}$ as given in Figure 2 (left). Connections between layers are given by (EA) $t$-th efficient graph matrices; (MP) copies of the adjacency matrix and the global readout. For unit weights, we omit labels.

Under $\chi_{\text{triv}}$, all vertices share the same color 0. Thus, every walk of length $t$ has color $\boldsymbol{z}_t = 00\cdots 0$ (constant word of length $t$). Hence, every set $S_t$ collapses to the singleton of $\boldsymbol{z}_t$, and computation over $L$ layers collapses to a linear sequence of length $L$. The REDUCE function receives the color 0 together with multisets of form $\text{mult}(\boldsymbol{z}_t, \boldsymbol{C}, \boldsymbol{h})$ containing a single pair $(m, \boldsymbol{h}[\boldsymbol{z}_t])$, where $m$ is a normalized count of plain walks (c.f., walk partition (Chung, 1997)). Formally, we have:

**Observation 3.2.** *Let $\chi_{\text{triv}}$ be the $\{0\}$-coloring on a graph $G$ with at least one edge. Then for every $t \geq 0$: (a) it holds that $|S_t| = 1$; (b) the only entries, $\boldsymbol{C}_t^{\boldsymbol{I}}[\boldsymbol{z}_t, \boldsymbol{z}_{t+1}] = \frac{\mathbf{1}^\top \boldsymbol{A}^{2t+1}\mathbf{1}}{\mathbf{1}^\top \boldsymbol{A}^{2t}\mathbf{1}}$, and $\boldsymbol{C}_t^{\boldsymbol{A}}[\boldsymbol{z}_t, \boldsymbol{z}_{t+1}] = \frac{\mathbf{1}^\top \boldsymbol{A}^{2t+2}\mathbf{1}}{\mathbf{1}^\top \boldsymbol{A}^{2t}\mathbf{1}}$.*

When each vertex is assigned a distinct color under $\chi_{\text{id}}$, every colored walk in the graph has its unique occurrence. Hence, every set $S_t$ reaches the maximum size $|V|$, with one colored walk per vertex. In this regime, the efficient matrices coincide entry-wise with the original matrices. Moreover, if REDUCE ignores its first parameter then EA reaches semantically the classical MP.

**Proposition 3.3.** *Let $\chi_{\text{id}}$ be the $V$-coloring on a graph $G = (V, E)$ with $n$ vertices. By $\boldsymbol{v}_{t,u}$, we denote the (unique) word in $S_t$ with the last color $u$ in $V$. Then for every $t \geq 1$: (a) it holds that $|S_0| = 1$, and $|S_t| = |V|$; (b) for entries $\boldsymbol{C}_0^{\boldsymbol{I}}[\lambda, u] = \frac{1}{n}$, and $\boldsymbol{C}_t^{\boldsymbol{I}}[\boldsymbol{v}_{t,u}, \boldsymbol{v}_{t+1,v}] = \boldsymbol{I}[u, v]$ and $\boldsymbol{C}_t^{\boldsymbol{A}}[\boldsymbol{v}_{t,u}, \boldsymbol{v}_{t+1,v}] = \boldsymbol{A}[u, v]$.*

## 4 EXPRESSIVITY CHARACTERIZATION

In this section, we characterize the expressivity of efficient aggregation (EA) through homomorphism counts. The motivation is to position our approach structurally within a hierarchy of graph classes ranging from paths to trees. This contrasts with existing approaches that begin with trees by default, recall Table 1. We first define caterpillar graphs and provide an explanatory diagram that summarizes our main results. These follow from two main theorems, each established in a separate subsection.

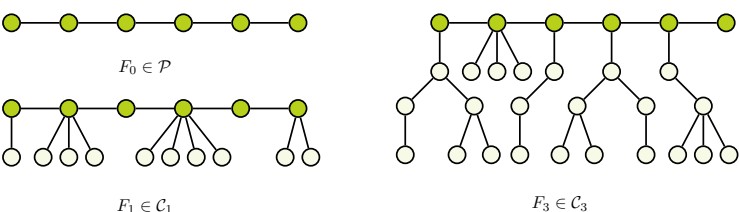

Figure 5: Caterpillar graphs with highlighted (possible) spine (green). Graph $F_0$ is a path of length 6, and also a $(0, 6)$-caterpillar. Graph $F_1$ is a $(1, 6)$-caterpillar, and graph $F_3$ is a $(3, 6)$-caterpillar.

**Caterpillar graphs.** A *caterpillar of height at most $h$ and length at most $t$*, or shortly $(h, t)$-*caterpillar* is a graph $F$ constructable as follows: take a sequence of rooted trees in $\mathcal{T}_h^\bullet$, i.e., $(L_1, s_1), \ldots, (L_t, s_t)$ and connect consequent roots with edges so that vertices $s_1, s_2, \ldots, s_t$ form a path $S$. We call $S$ a *spine*, the rooted trees *legs*, and their sequence a *leg sequence*. We denote the class of all caterpillars of height at most $h$ by $\mathcal{C}_h$, and by $\mathcal{C}_{h,t}$ the subclass of $(h, t)$-caterpillars.

For instance, every caterpillar in $\mathcal{C}_0$ is a path, $\mathcal{P} = \mathcal{C}_0$, see examples in Figure 5. The "folklore" caterpillars here correspond to $\mathcal{C}_1$ and are often used in graph theory (Harary and Schwenk, 1973; El-Basil, 1987). Other generalization of caterpillars using hair-length is due to Monien (1986).

**Expressivity Hierarchy.** Our main findings on expressivity of EA (Equation 3), we situate diagrammatically in the context of homomorphism expressivity. This provides a *scale* clarifying the expressive power of the associated inductive biases:

$$
\begin{array}{ccccccccccc}
\mathrm{hom}(\mathcal{P}, -) & \subsetneqq & \mathrm{hom}(\mathcal{C}_1, -) & \subsetneqq & \mathrm{hom}(\mathcal{C}_2, -) & \sqsubseteq \cdots \sqsubseteq & \mathrm{hom}(\mathcal{C}_h, -) & \sqsubseteq \cdots \sqsubseteq & \mathrm{hom}(\mathcal{T}, -) \\
\| & & \| & & \| & & \| & & \| \\
\mathcal{I}_{\mathrm{EA}}(-, \chi_{\mathrm{triv}}) & \subsetneqq & \mathcal{I}_{\mathrm{EA}}(-, \chi_{\deg}) & \subsetneqq & \mathcal{I}_{\mathrm{EA}}(-, \chi_{\mathrm{cr}}^{(2)}) & \sqsubseteq \cdots \sqsubseteq & \mathcal{I}_{\mathrm{EA}}(-, \chi_{\mathrm{cr}}^{(h)}) & \sqsubseteq \cdots \sqsubseteq & \mathrm{cr}(-),
\end{array}
$$

where height $h \geqslant 3$. The vertical equivalences follow from Theorem 4.1, and Theorem 4.2 which we establish in Section 4.1 and Section 4.2, respectively. The last one involving $\mathcal{T}$ is due to Dvořák (2010, Theorem 7). Note that color refinement cr symbolizes message-passing (MP). The horizontal bounds follow by definition from $\mathcal{C}_h \subset \mathcal{C}_{h+1} \subset \mathcal{T}$, while the strictness of the first two bounds follows from Lemma B.13 adopting the results of Roberson (2022); Schindling (2025).

## 4.1 CATERPILLAR HOMOMORPHISMS AS EXPRESSIVE AS COLORED WALKS

*Colored walk refinement:* Let $\chi$ be a $\Sigma$-coloring on a graph $G$ with $n$ vertices. We define a sequence of multisets $\mathrm{wr}^{(t)}(G, \chi)$ in $\mathbb{N}^{\Sigma^t}$ for each $t \geqslant 0$ with $\boldsymbol{a}$ in $\Sigma^t$ as: $\mathrm{wr}^{(0)}(G, \chi)[\lambda] = n$, $\mathrm{wr}^{(t)}(G, \chi)[\boldsymbol{a}]$ equals the number of occurrences of $\boldsymbol{a}$ in $G$, and $\mathrm{wr}(G, \chi) = \{\mathrm{wr}^{(t)}(G, \chi) \mid t \text{ in } \mathbb{N}\}$.

The reader may recall walk-incidence matrices in Equation 1, then multiplicity in $\mathrm{wr}^{(t)}$ is a sum of entries in the corresponding column of $\boldsymbol{W}_t$. Note that our colored walk refinement is distinct from what is usually called walk refinement, i.e. (Lichter et al., 2019). The following result motivates our use of colored walks that is not ad-hoc but due to its correspondence with homomorphisms:

**Theorem 4.1.** *For every $h, t \geqslant 0$, it holds that* $\mathrm{hom}(\mathcal{C}_{h,t}, -) \equiv \mathrm{wr}^{(t)}(-, \chi_{\mathrm{cr}}^{(h)})$.

*(Proof in Appendix B).* A direct consequence of Theorem 4.1 is: to capture the expressive power of homomorphism counts over folklore caterpillars (for instance) of length $t$, it suffices to color the vertices by their degrees and record every occurrence of a colored walk of length $t$ by $\mathrm{wr}^{(t)}(G, \chi_{\deg})$, see Figure 1 for example of such a graph homomorphism.

## 4.2 EFFICIENT AGGREGATION IS AS EXPRESSIVE BUT TRACTABLE

The previous result depicted more clearly the semantics of caterpillar homomorphisms, however, that is still not computationally tractable. As we observe, the number of distinct colored walks in a graph can be large, exponential in the worst case. Therefore, it is crucial that we introduced more efficient but as expressive representation of $\mathrm{wr}(-, \chi)$, which we state for *any* coloring of vertices.

**Theorem 4.2.** *For every coloring $\chi$ it holds that* $\mathrm{wr}(-, \chi) \equiv \mathcal{I}_{EA}(-, \chi)$.

*(Proof in Appendix A). Sketch of proof.* Central to our arguments is that all the colored walks occurring together in a graph $G$ with a $\Sigma$-coloring $\chi$ form a regular language. As a consequence, rather than (naively) enumerating all elements of the language, we construct an automaton $\mathcal{A}(G, \chi)$ recognizing such language. To reflect multiplicities of occurrences in the language, we use a weighted variant of finite automata introduced by Tzeng (1996).

The "$\sqsupseteq$-direction" of the equivalence requires that $\mathcal{I}_{\mathrm{EA}}(G, \chi)$ derived from $\mathcal{A}(G, \chi)$ depends purely on the recognized language and ordering on $\Sigma$ via the canonical walk selection ($S_t$ in Section 3.1). For the other "$\sqsubseteq$-direction", given two automata recognizing distinct languages, subsequent matrices of $\mathcal{I}_{\mathrm{EA}}$ decompose back to automata transitions (Lemma A.13) to detect the difference.

## 5 EXPERIMENTS

We next turn to an empirical analysis of Caterpillar GNN (Equation 5) incorporating efficient aggregation (EA). Because expressivity of EA (Section 4) is controlled by its height, we propose experiments to empirically evaluate behavior of subsequent inductive bias. Two scenarios are considered: (I.) a controlled benchmark isolating topology-driven preference for stronger inductive bias, and (II.) real-world graph-level tasks investigating the impact of height (Section 3.3) on the trade-off between nodal efficiency and performance. We defer full implementation details to Appendix C, and the training setup to Appendix D.

### 5.1 SCENARIO I: REDUCING A BOTTLENECK

Prior to any processing of a graph $(V, E)$, the neighborhood topology $\tau(E)$ on $V$ specifies which vertices are considered close, namely those in neighborhoods. We instead consider an alternative *incidence topology* $\tau(\chi)$ on $V$, induced by a coloring $\chi$: two vertices are considered close if they are incident *or* adjacent to a common colored walk of length $T$. Since a colored walk may have multiple occurrences, this captures relationships beyond direct neighbors. We use $\tau(\chi)$ as a model to study different inductive biases in graph learning, grounded in lower-order concepts such as colored walks as shown by Theorem 4.1.

We illustrate this with our NSTEPADDITION benchmark. Given two integers of at most $T$ bits, take a graph with two occurrences of a colored walk $a_1 \cdots a_T$. We associate each number with one occurrence as follows: encode the $i$-th bit of the integer in a vertex adjacent to $a_1 \cdots a_i$. This yields a graph embedding of two $T$-bit integers, and the classification task is whether their sum equals a target integer $N$. Under the incidence topology $\tau(\chi)$ corresponding bit positions are naturally close, while standard topology $\tau(E)$ may obscure such alignments. Therefore, we evaluated Caterpillar GNN on NSTEPADDITION with increasing height $h$, and compared to MPGNN. The results are presented in Figure 6, detailed information is provided in Appendix D.

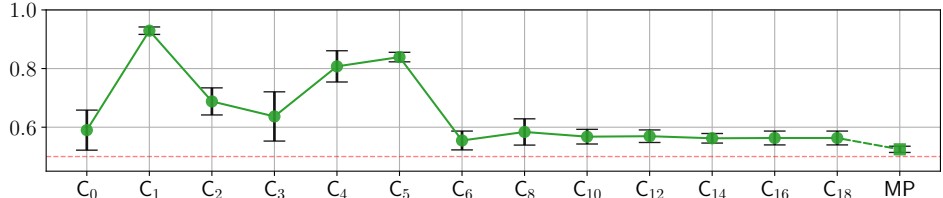

Figure 6: NSTEPADDITION: more expressivity hurts. $\mathsf{C}_h$ denotes Caterpillar GNN with height $h$ (ours), while $\mathsf{MP}$ refers to MPGNN. The y-axis shows validation accuracy.

**Importance of topology.** The extremal results for $\mathsf{C}_1$ and $\mathsf{MP}$ (in Figure 6) highlight a clear difference between the two models. In MPGNN, information propagates according to $\tau(E)$. A hypothesis arises that Caterpillar GNN propagates information according to topology $\tau(\chi)$. We validate empirically on NSTEPADDITION that model $\mathsf{C}_1$, as well as topology $\tau(\chi_{\mathrm{deg}})$, aligns bit positions for effective digit-by-digit addition, while $\mathsf{MP}$ within $\tau(E)$ effectively promotes learning values separately for each input pair, resulting in almost missing generalization.

**Paradoxically reversed descent.** We also observe a double descent, which we attribute to training oscillation between two regimes: digit-by-digit processing, and a higher-level aggregation, producing the high-variance performance dip. As we scale our models (cf. Section 3.3, Section 4), incidence topology scales analogically from $\tau(\chi_{\mathrm{triv}})$ up to $\tau(\chi_{\mathrm{id}}) = \tau(E)$. Unlike rewiring strategies, e.g., Topping et al. (2022), changing edges $E'$ to operate in $\tau(E')$, our approach restructures the computational graph into a less expressive one (Section 4). Finally, given the systematically decreasing performance of models $\mathsf{C}_{10}$ up to $\mathsf{MP}$, a bottleneck of *information alignment* of $\tau(E)$ is revealed by the topology $\tau(\chi_{\mathrm{deg}})$ that qualitatively differs from, e.g., oversquashing (Alon and Yahav, 2021).

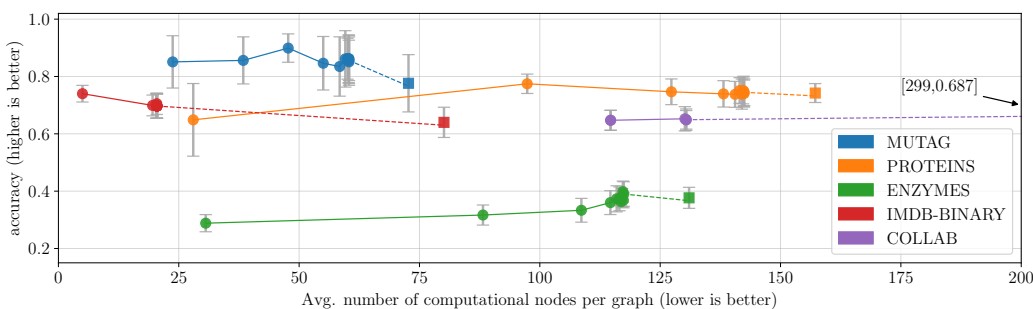

Figure 7: Computational nodes vs. accuracy. Solid segments connect models of Caterpillar GNN, with height $h = 0$ ($\mathsf{C}_0$, circle) up to $h = 10$ ($\mathsf{C}_{10}$, circle), the last dashed is to MPGNN (square).

## 5.2 Scenario II: Nodal Efficiency

We evaluate GNNs (Figure 7) on common real-world classification datasets (Morris et al., 2020) in dependence to *nodal efficiency*, i.e. the average number of nodes of the computational graph (Figure 4). We fixed the number of layers for models to ensure a relative comparison. To this end, hyperparameters are per-dataset, so that the behavior can be attributed solely to the height parameter. In our experiments (Figure 7), every real-world dataset exhibits a unique behavior under increasing height. This suggests that every type of data may contain patterns organized in varying topologies resulting in distinct preferences for inductive biases. Effectively, the height parameter *shapes the model performance and nodal efficiency*. We remind that incorporated EA requires one initial precomputation (Theorem 3.1) of efficient matrices per dataset and height. Overall, Caterpillar GNNs of the optimum height achieved comparable or superior accuracy compared to MPGNN as detailed in Table 7.

## 6 Related Work

Graph homomorphisms are an active area of research in graph learning. One line of work uses homomorphism counts directly as *features* (Barceló et al., 2021; Maehara and NT, 2024; Jin et al., 2024), or as *embeddings* (Nguyen and Maehara, 2020; Thiessen et al., 2022). Several *extensions* of MPGNNs formally demonstrate the expressivity of homomorphism counts over classes extending beyond trees, listed in Table 1, including (Zhang et al., 2023a; Paolino et al., 2024). Other line of work enhances expressivity without relating to homomorphisms. This includes cycle representations (Yan et al., 2022; Bause et al., 2025), path representations (Michel et al., 2023; Graziani et al., 2024), distance encodings (Li et al., 2020; Zhang et al., 2023b), and spectral information such as (Defferrard et al., 2017; Kreuzer et al., 2021), which is in contrast to our study of lower expressivity.

In our results, we rely on theoretical study of homomorphism counts that traces back to Lovász (1967; 2012), and their connection to Weisfeiler-Leman refinement (Weisfeiler and Leman, 1968) which is due to Dvořák (2010); Dell et al. (2018). Further developments include quantum isomorphism via homomorphisms over planar graphs (Mančinska and Roberson, 2020), further expanded by Grohe et al. (2022); Kar et al. (2025), as well as algorithmic results on the tractability of homomorphism indistinguishability over restricted classes (Seppelt, 2024).

Learning on sequential patterns such as walks has been also approached via non-equivariant random-walk kernels (Borgwardt et al., 2005; Kriege, 2022). Other work investigates slowing down message-passing as a regularizing inductive bias (Bause and Kriege, 2022). Recently, the role of computational graph in deep learning has been explored (Vitvitskyi et al., 2025). In addition, least squares-based operators have been applied to cross-network optimization (Wang et al., 2024), or graph coarsening (Jin et al., 2020; Stamm et al., 2023). These operators target graphs at a different level of abstraction, not considering layer-specific walk incidence matrices or homomorphism counts.

## 7 CONCLUSION

In this paper[2], we introduced mechanism that scales GNN's computational graph using the parameter height. Subsequent *Caterpillar GNNs* enable controlled trade-off between expressivity, strength of inductive bias and its nodal efficiency. Beyond the empirical gains, such as accuracy-increasing reduction of computational nodes to $6\%$ on unattributed IMDB-BINARY, our work gives broader insight: *less expressive but strongly organized aggregation can outperform unconstrained message passing*. Finally, our mechanism, its derivation and its rigorous theoretical analysis using colored walks and homomorphism counts over caterpillar graphs are stated in general terms and remain independent of most implementation choices. This provides basis for further applications such as integration into state-of-the art backbones and ensembles, where height parameter enables additional space for optimization via computational graph. A notable limitation of our expressivity characterization is its assumption of undirected graphs, which does not directly extend to directed ones.

---

[2]Use of Large Language Models. We used LLMs exclusively for grammar checking and wording improvements. All conceptual content, results, and analyses were developed by the authors.

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

## NOTATION AND SYMBOLS

$\boldsymbol{A} \in \mathbb{R}^{V \times V}$      adjacency matrix of $G$ sort. (pg. 2)

$\boldsymbol{1} \in \mathbb{R}^{V \times 1}$      all-ones column vector indexed by $V$. (pg. 2)

$\boldsymbol{B}_t \in \mathbb{R}^{V \times S_t}$      column submatrix of walk-incidence matrix $\boldsymbol{W}_t$ induced by $S_t$. (pg. 5)

$\boldsymbol{C}_t^{\boldsymbol{A}} \in \mathbb{R}^{S_t \times S_{t+1}}$      $t$-th efficient adjacency matrix for $\boldsymbol{A} \in \mathbb{R}^{V \times V}$. (pg. 5)

$\boldsymbol{C}_t^{\boldsymbol{I}} \in \mathbb{R}^{S_t \times S_{t+1}}$      $t$-th efficient self-loop matrix for $\boldsymbol{I} \in \mathbb{R}^{V \times V}$. (pg. 5)

$\boldsymbol{C}_t^{\boldsymbol{M}} \in \mathbb{R}^{S_t \times S_{t+1}}$      $t$-th efficient $\boldsymbol{M}$-matrix for $\boldsymbol{M} \in \mathbb{R}^{V \times V}$. (pg. 5)

$\mathcal{C}_h$      class of all caterpillars of height at most $h$. (pg. 6)

$\mathcal{C}_{h,t} \subseteq \mathcal{C}_h$      class of all $(h,t)$-caterpillars. (pg. 6)

$\chi$      coloring function $\chi(G,-)\colon V \to \Sigma'$ for $G$. (pg. 3)

$\chi_{\mathrm{cr}}^{(h)}$      color refinement coloring at height $h$. (pg. 3)

$\chi_{\mathrm{deg}}$      degree coloring. (pg. 3)

$\chi_{\mathrm{id}}$      identity coloring. (pg. 3)

$\chi_{\mathrm{triv}}$      trivial coloring. (pg. 3)

$\mathrm{cr}(-)$      function on graphs of color refinement. (pg. 3)

$G = (V, E)$      undirected graph with finite vertex set $V$ and edge set $E \subseteq V^2$. (pg. 2)

$\mathcal{G}$      class of all finite undirected graphs. (pg. 3)

$\boldsymbol{h}_{\mathrm{EA}}^{(\ell)} \in \mathbb{R}^{S_{t_\ell} \times d}$      features after $\ell$ layers of a Caterpillar GNN with $d$ channels. (pg. 5)

$\boldsymbol{h}_{\mathrm{EA}} \in \mathbb{R}^{\{\lambda\} \times d}$      vector computed by a Caterpillar GNN with $d$ channels. (pg. 5)

$\boldsymbol{h}_{\mathrm{MP}}^{(\ell)} \in \mathbb{R}^{V \times d}$      vertex features after $\ell$ layers of an MPGNN with $d$ channels. (pg. 3)

$\boldsymbol{h}_{\mathrm{MP}} \in \mathbb{R}^{1 \times d}$      vector computed by an MPGNN with $d$ channels. (pg. 3)

$\mathrm{hom}(\mathcal{F}, -)$      function of graphs assigning vector in $\mathbb{N}^{\mathcal{F}}$ for a class of source graphs $\mathcal{F}$. (pg. 3)

$\boldsymbol{I} \in \mathbb{R}^{V \times V}$      self-loop (identity) matrix of $G$. (pg. 2)

$\mathcal{I}_{\mathrm{EA}}(-, \chi)$      function of graphs of efficient aggregation invariant for a coloring $\chi$. (pg. 5)

$[k]$      (index) set for $k \in \mathbb{N}$. (pg. 2)

$\{\!\!\{ - \}\!\!\}$      multiset. (pg. 2)

$\mathcal{P} \subset \mathcal{G}$      class of all finite undirected paths. (pg. 2)

$\mathcal{P}_t \subset \mathcal{P}$      class of paths of length at most $t$, where length means $|E|$. (pg. 2)

$S_t \subseteq \Sigma^t$      canonical subset of colored walks of length $t$ in $G$ for a $\Sigma$-coloring $\chi$. (pg. 4)

$\mathcal{T} \subset \mathcal{G}$      class of all trees. (pg. 2)

$\mathcal{T}^{\bullet}$      class of rooted trees. (pg. 2)

$\mathcal{T}_h^{\bullet} \subset \mathcal{T}^{\bullet}$      class of rooted trees where every vertex is at distance at most $h$ from the root. (pg. 2)

$W_t \in \mathbb{R}^{V \times \Sigma^t}$     walk-incidence matrix of length $t$ colored walks in $G$ for a $\Sigma$-coloring $\chi$. (pg. 4)

$\mathrm{wr}(-, \chi)$     function on graphs of colored walk refinement for a coloring $\chi$. (pg. 7)

## TABLE OF CONTENTS

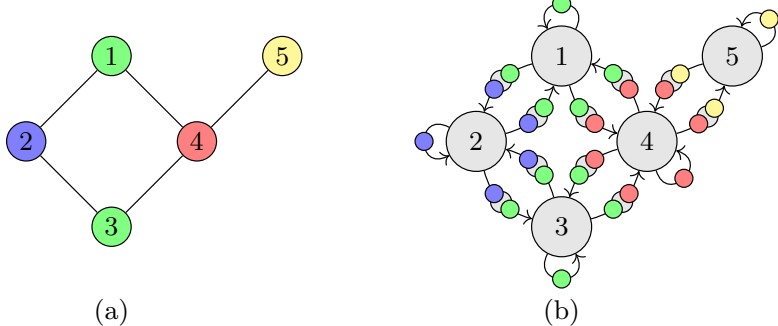

(a)                          (b)

Figure 8: (a) An example graph $G$ on vertices $\{1, 2, 3, 4, 5\}$ with a $\Sigma$-coloring $\chi$, where $\Sigma = \{r, b, g, y\}$. (b) The weighted automaton $\mathcal{A}(G, \chi)$ that accepts a weighted language of words corresponding to colored walks in $G$ (Lemma A.5). The weights of the language represent the number of occurrences of each walk in $G$. Transitions corresponding to directed edges $uv$ in $E(G)$ are represented by matrices $M(\chi(u)|\chi(v)) = P_{\chi(u)} A P_{\chi(v)}$, while transitions corresponding to loops at vertex $u$ in $V(G)$ are given by matrices $M(\chi(u)) = P_{\chi(u)} I P_{\chi(u)}$.

## A    WEIGHTED AUTOMATA

In this section, we briefly recall the concept of weighted finite automata (cf. Tzeng (1996); Kiefer et al. (2013)). Then, we apply insights from automata theory, using them as a key technical tool to establish the results of Section 3 and Theorem 4.2 from Section 4.

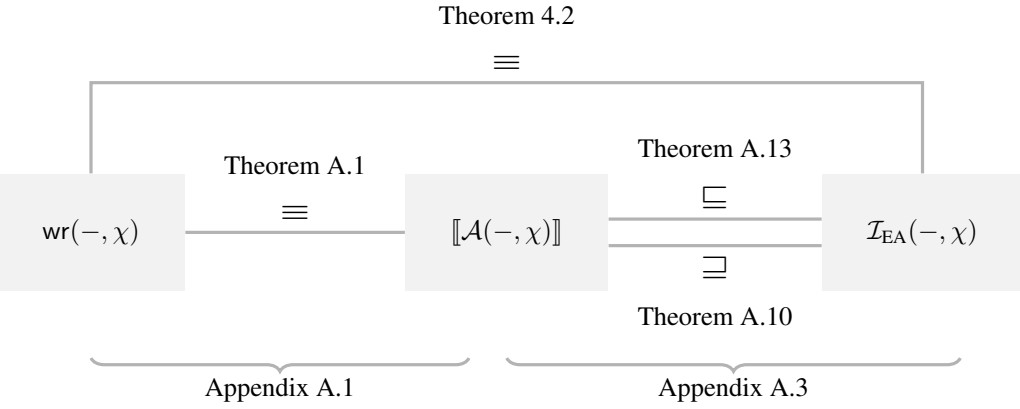

Figure 9: Sections A.1 and A.3 establish the result of Theorem 4.2 for a coloring $\chi$ by relating the following three functions on graphs: Colored walk refinement $\mathsf{wr}(-, \chi)$, semantics of graph walk automaton $[\![\mathcal{A}(-, \chi)]\!]$, and the matrices of efficient aggregation $\mathcal{I}_{\mathrm{EA}}(-, \chi)$.

**Automata.**    A *weighted finite automaton*, or here simply *automaton*, is a tuple

$$\mathcal{A} = (Q, \Sigma, M, \alpha, \omega), \tag{6}$$

where $Q$ is a finite set of states; $\Sigma$ is a finite alphabet; $M(-) \colon \Sigma \to \mathbb{R}^{Q \times Q}$ is a per-symbol mapping of transition matrices, $\alpha$ in $\mathbb{R}^{1 \times Q}$ is the initial state (row) vector, and $\omega$ in $\mathbb{R}^{Q \times 1}$ is the final state (column) vector.

**Semantics.**    Given an automaton $\mathcal{A} = (Q, \Sigma, M, \alpha, \omega)$, we extend the mapping $M \colon \Sigma \to \mathbb{R}^{Q \times Q}$ to words as follows: for a given word $w = w_1 w_2 \ldots w_t$ in $\Sigma^*$, we define $M(w) = M(w_1) M(w_2) \cdots M(w_t) \in \mathbb{R}^{Q \times Q}$, and $M(\lambda) = I \in \mathbb{R}^{Q \times Q}$ for the empty word $\lambda$ in $\Sigma$. The

*semantics* of the automaton $\mathcal{A}$ is a function $[\![\mathcal{A}]\!] \colon \Sigma^* \to \mathbb{R}$, interpreted as a formal series, defined by

$$[\![\mathcal{A}]\!](\boldsymbol{w}) = \alpha \boldsymbol{M}(\boldsymbol{w})\omega \in \mathbb{R} \quad \text{for all } \boldsymbol{w} \text{ in } \Sigma^*. \tag{7}$$

Two automata $\mathcal{A}$ and $\mathcal{A}'$ are *equivalent* if their semantics are equivalent, that is, $[\![\mathcal{A}]\!](\boldsymbol{w}) = [\![\mathcal{A}']\!](\boldsymbol{w})$ for all $\boldsymbol{w}$ in $\Sigma^*$. The value $[\![\mathcal{A}]\!](\boldsymbol{w})$ is called the *weight* of $\boldsymbol{w}$. For a symbol $a$ in $\Sigma$, let $a^k$ denote the word formed by repeating $a$ exactly $k$-times.

### A.1 GRAPH WALKS AND AUTOMATA SEMANTICS

As in the main text, we assume a graph $G$ and $\chi$ a $\Sigma$-coloring on $G$, for which we now define a weighted finite automaton $\mathcal{A}(G, \chi) = (Q, \overline{\Sigma}, \boldsymbol{M}, \mathbf{1}^\top, \mathbf{1})$ defined as follows. The states are corresponding to the vertices of the graph $Q := V(G)$, and the alphabet is induced by the colors of the vertices and edges as follows:

$$\overline{\Sigma} := \{\chi(u) \mid u \in V(G)\} \cup \{\chi(u) | \chi(v) \mid uv \in E(G)\}, \tag{8}$$

where we consider both $a$ and $a|b$ as a single symbol in $\overline{\Sigma}$, for some original colors in $a, b$ in $\Sigma$. For $a$ in $\overline{\Sigma}$, the *partition matrix* $\boldsymbol{P}_a$ in $\mathbb{R}^{V \times V}$ is the diagonal matrix defined as $\boldsymbol{P}_a[u, u] = 1$ if $\chi(u) = a$ and 0 otherwise. For each symbol $a$ or $a|b$ in $\overline{\Sigma}$, we define the transition matrices using the adjacency matrix of $G$ and the coloring-dependent partition matrices as

$$\boldsymbol{M}(a) := \boldsymbol{P}_a = \boldsymbol{P}_a \boldsymbol{I} \boldsymbol{P}_a \text{ in } \mathbb{R}^{V \times V} \text{ and } \boldsymbol{M}(a|b) := \boldsymbol{P}_a \boldsymbol{A} \boldsymbol{P}_b \text{ in } \mathbb{R}^{V \times V}.$$

We set the initial and final vectors to the all-one vectors. We depict an example of a graph and a coloring and the corresponding automata in Figure 8.

We also recall the following from the main paper. For a given graph $G$ with $\chi$ a $\Sigma$-coloring, and a given length $t \geqslant 0$, we define the *walk-incidence matrix* $\boldsymbol{W}_t = \boldsymbol{W}_t(G, \chi)$ of shape $V \times \Sigma^t$ for each $u$ in $V$ and $\boldsymbol{a}$ in $\Sigma^t$ by

$$\boldsymbol{W}_t[u, \boldsymbol{a}] \text{ is the number of occurrences of } \boldsymbol{a} \text{ that terminate in vertex } u. \tag{9}$$

*Colored walk refinement:* Let $\chi$ be a $\Sigma$-coloring on a graph $G$ with $n$ vertices. We define a sequence of multisets $\mathrm{wr}^{(t)}(G, \chi)$ in $\mathbb{N}^{\Sigma^t}$ for each $t \geqslant 0$ with $\boldsymbol{a}$ in $\Sigma^t$ as: Let $\chi$ be a $\Sigma$-coloring on a graph $G$ with $n$ vertices. A *colored walk refinement* is a sequence of multisets $\mathrm{wr}^{(t)}(G, \chi)$ in $\mathbb{N}^{\Sigma^t}$ defined for each $t \geqslant 0$ with $\boldsymbol{a}$ in $\Sigma^t$ as

$$\mathrm{wr}^{(0)}(G, \chi)[\lambda] = n,$$
$$\mathrm{wr}^{(t)}(G, \chi)[\boldsymbol{a}] \quad \text{equals the number of occurrences of } \boldsymbol{a} \text{ in } \mathrm{G}, \text{ and}$$
$$\mathrm{wr}(G, \chi) = \{\mathrm{wr}^{(t)}(G, \chi) \mid t \text{ in } \mathbb{N}\}.$$

The main result of this subsection is the equivalence of graph-induced weighted finite automata and colored walk refinement.

**Theorem A.1.** *For every coloring $\chi$ it holds that*

$$[\![\mathcal{A}(-, \chi)]\!] \equiv \mathrm{wr}(-, \chi).$$

An implication of Theorem A.1 is that instead of the sums of columns of walk-incidence matrices, we can walk directly with the automata semantics, for which we can *choose* the representation of the automaton potentially more suitable for our application.

To prove this theorem, we need a couple of intermediate results. We first show that the partition matrices $\boldsymbol{P}_a$ are projection matrices.

**Proposition A.2** (Partition matrices). *Let $a$ and $b$ be colors in $\Sigma$ then for the partition matrices the following hold:*

1. $\boldsymbol{P}_a = \boldsymbol{P}_a^2$ *is a projection, and*

2. $\boldsymbol{P}_a \boldsymbol{P}_b$ *is all-zero if $a \neq b$.*

*Proof.* For the first part, we have for all $u, v$ in $V(G)$:

$$(\boldsymbol{P}_a \boldsymbol{P}_a)[u, v] = \sum_{w \in V(G)} \boldsymbol{P}_a[u, w] \boldsymbol{P}_a[w, v] = \sum_{w = u} 1 \cdot \boldsymbol{P}_a[w, v] = \sum_{u = w = v} 1 \cdot 1,$$

which is 1 if $u = v$ and 0 otherwise. Similarly, for the second part, given the colors $a, b$ in $\Sigma$ we have for all $u, v$ in $V(G)$:

$$(\boldsymbol{P}_a \boldsymbol{P}_b)[u, v] = \sum_{w \in V(G)} \boldsymbol{P}_a[u, w] \boldsymbol{P}_b[w, v] = \sum_{\substack{u = w \\ a = \chi(w)}} 1 \cdot \boldsymbol{P}_b[w, v] = \sum_{\substack{u = w, w = v \\ a = \chi(w), \chi(w) = b}} 1 \cdot 1,$$

which is 1 if $u = v$ and $a = b$, and 0 otherwise. Thus, $\boldsymbol{P}_a \boldsymbol{P}_b$ is all-zero matrix if $a \neq b$. □

**Observation A.3.** *Let $\boldsymbol{w}$ in $\overline{\Sigma}$ be a word with a non-zero weight, $[\![\mathcal{A}(G, \chi)]\!](\boldsymbol{w}) > 0$, then $\boldsymbol{w}$ is of the following form:*

$$(a_1^{k_1})(a_1 | a_2)(a_2^{k_2})(a_2 | a_3)(a_3^{k_3}) \cdots (a_{t-1}^{k_{t-1}})(a_{t-1} | a_t^{k_t})(a_t^{k_t}),$$

*where $a_i$ in $\Sigma$ are colors, $k_i$ in $\mathbb{N}$ are non-negative integers,*

*Proof.* For the sake of contradiction, consider a word $\boldsymbol{w}$ that contains a subword $ab|c$ in $\overline{\Sigma}$, $c|ba$ in $\overline{\Sigma}$, or $ab$ in $\overline{\Sigma}$ such that $a \neq b$ and weight $[\![\mathcal{A}(G, \chi)]\!](\boldsymbol{w}) > 0$. In the first case, by the definition of semantics, we get for any $\boldsymbol{w}_1, \boldsymbol{w}_2$ in $\Sigma^*$:

$$\begin{aligned}
[\![\mathcal{A}(G, \chi)]\!](\boldsymbol{w}_1 ab|c \, \boldsymbol{w}_2) &= \boldsymbol{1}^\top \boldsymbol{M}(\boldsymbol{w}_1) \boldsymbol{M}(a) \boldsymbol{M}(b|c) \boldsymbol{M}(\boldsymbol{w}_2) \boldsymbol{1} \\
&= \boldsymbol{1}^\top \boldsymbol{M}(\boldsymbol{w}_1) \boldsymbol{P}_a \boldsymbol{P}_b \boldsymbol{A} \boldsymbol{P}_c \boldsymbol{M}(\boldsymbol{w}_2) \boldsymbol{1} \\
&= \boldsymbol{1}^\top \boldsymbol{M}(\boldsymbol{w}_1) \boldsymbol{0} \boldsymbol{M}(\boldsymbol{w}_2) \boldsymbol{1} = 0,
\end{aligned}$$

where we used (1.) of Proposition A.2 for the second equality. Thus, we have $[\![\mathcal{A}(G, \chi)]\!](\boldsymbol{w}_1 ab|c \, \boldsymbol{w}_2) = 0$, and symmetrically $[\![\mathcal{A}(G, \chi)]\!](\boldsymbol{w}_1 c|ba \, \boldsymbol{w}_2) = 0$, and analogically $[\![\mathcal{A}(G, \chi)]\!](\boldsymbol{w}_1 ab \, \boldsymbol{w}_2) = 0$. □

**Lemma A.4.** *For all $t$ in $\mathbb{N}$, $a_i$ in $\Sigma$ and $k_i$ in $\mathbb{N}$ for $i$ in $[t]$ the weight of the following words is equal:*

1. $(a_1^{k_1})(a_1 | a_2)(a_2^{k_2})(a_2 | a_3)(a_3^{k_3}) \cdots (a_{t-1}^{k_{t-1}})(a_{t-1} | a_t^{k_t})(a_t^{k_t})$, *and*

2. $(a_1 | a_2)(a_2 | a_3) \cdots (a_{t-1} | a_t)$.

*Proof.* This follows from (1) of Proposition A.2. Indeed, it suffices to observe that

$$\boldsymbol{M}\left(a_i^{k_i}\right) = \boldsymbol{M}(a_i), \text{ and}$$

$$\boldsymbol{M}\left((a_i^{k_i})(a_i | a_{i+1})\left(a_{i+1}^{k_{i+1}}\right)\right) = \boldsymbol{P}_{a_i}^{k_i} \boldsymbol{P}_{a_i} \boldsymbol{A} \boldsymbol{P}_{a_{i+1}} \boldsymbol{P}_{a_{i+1}}^{k_{i+1}} = \boldsymbol{P}_{a_i} \boldsymbol{A} \boldsymbol{P}_{a_{i+1}} = \boldsymbol{M}(a_i | a_{i+1}).$$

Then, from the semantics of automata, the statement on the equal weights follows. □

Therefore, by Lemma A.4, we can characterize the semantics of automata $\mathcal{A}(G, \chi)$, using only words of the form as in Item 2 without loss of generality. Consequently, by Theorem A.3, we shall use a more natural *simplified notation*

$$a_1 | a_2 | a_3 \cdots a_{t-1} | a_t$$

for the word $(a_1 | a_2)(a_2 | a_3) \cdots (a_{t-1} | a_t)$.

We now make the first connection between weighted automata and walk-incidence matrices.

**Lemma A.5.** *Let $G$ be a graph, $\chi$ a $\Sigma$-coloring on $G$, and $(w_1, w_2, \ldots, w_t)$ in $\Sigma^t$ a colored walk. Then the following holds:*

$$\boldsymbol{M}(w_t | w_{t-1} | \ldots | w_1) \boldsymbol{1} = \boldsymbol{W}_t(G, \chi)[-, w_1 w_2 \cdots w_t]. \tag{10}$$

*Proof.* By induction on the length $t$. For the base case $t = 0$, it holds $M(\lambda)\mathbf{1} = I\mathbf{1} = \mathbf{1} = W_0(G, \chi)[-, \lambda]$.

For the induction step, suppose that the lemma holds for $t$. We take a word $w_1|w_2|\ldots|w_{t+1}$ and consider a vertex $u$ in $V(G)$. Then,

$$
\begin{aligned}
(M(w_{t+1}|w_t|\ldots|w_1)\mathbf{1})[u] &= (M(w_{t+1}|w_t)M(w_t|w_{t-1}|\ldots|w_1)\mathbf{1})[u] \\
&= \sum_{v \in V(G)} M(w_{t+1}|w_t)W_t(G, \chi)[v, w_1 w_2 \cdots w_t] \\
&= \sum_{v \in V(G)} P_{w_{t+1}}[u, u]A[u, v]P_{w_t}[v, v]W_t(G, \chi)[v, w_1 w_2 \cdots w_t] \\
&= \sum_{\substack{v \in V(G) \\ w_{t+1} = \chi(v) \\ w_t = \chi(u)}} A[v, u]W_t(G, \chi)[v, w_1 w_2 \cdots w_t] \\
&= \sum_{\substack{uv \in E(G) \\ \chi(v) = w_t \\ \chi(u) = w_{t+1}}} W_t(G, \chi)[u, w_1 w_2 \cdots w_t] \\
&= W_{t+1}(G, \chi)[u, w_1 w_2 \cdots w_{t+1}],
\end{aligned}
$$

where the first and third equality follows from the definition of $M$, the second from the induction hypothesis, the forth and fifth follows from the definition of $P_{w_{t+1}}$ and $A$. Finally, the last equation follows from the observation that we can find every occurrence of the walk of color $w_1 w_2 \cdots w_t w_{t+1}$ ending in the vertex $u$, given that $\chi(u) = w_{t+1}$, as an occurrence of the walk of color $w_1 w_2 \cdots w_t$ ending at one of its neighbors $v$, which is of color $\chi(v) = w_t$. $\qquad\square$

**Theorem A.6.** *Let $G$ be a graph, $\chi$ a $\Sigma$-coloring on $G$, and $(w_1, w_2, \ldots, w_t)$ in $\Sigma^t$ a colored walk. Then the following holds:*

$$
[\![\mathcal{A}(G, \chi)]\!](w_t|w_{t-1}|\ldots|w_1) = \sum_{u \in V(G)} W_t(G, \chi)[u, w_1 w_2 \cdots w_t]. \tag{11}
$$

*Proof.* By the definition of semantics, we have:

$$
\begin{aligned}
[\![\mathcal{A}(G, \chi)]\!](w_t|w_{t-1}|\ldots|w_1) &= \mathbf{1}^\top M(w_1|w_2|\ldots|w_t)\mathbf{1} \\
&= \mathbf{1}^\top (M(w_1|w_2|\ldots|w_t))^\top \mathbf{1} \\
&= \mathbf{1}^\top M(w_t|w_{t-1}|\ldots|w_1)\mathbf{1} \\
&= \mathbf{1}^\top W_t(G, \chi)[u, w_1 w_2 \cdots w_t] \\
&= \sum_{u \in V(G)} W_t(G, \chi)[u, w_1 w_2 \cdots w_t],
\end{aligned}
$$

where the second equality follows from the symmetry of all matrices $M$, since $P_a$ and $A$ are symmetric matrices, $a \in \Sigma$, the fourth equality follows from Lemma A.5. $\qquad\square$

*Proof of Theorem A.1.* Follows from Theorem A.6, by definition of colored walk refinement. $\qquad\square$

A.2 MINIMIZATION OF WEIGHTED AUTOMATA

In this subsection, we recall the definition of the *minimal weighted automata* (cf. Kiefer et al. (2013), Kiefer (2020)). Unlike for the standard finite automata, in the weighted case, the concrete variant is unique only up the change of the basis of the vector space $\mathbb{R}^Q$. On the bright side, every such minimal weighted automata has the unique dimension, that is the number of states $|Q|$, and the unique (canonical) word subset $S \subseteq \Sigma^*$ of size at most $|Q|$. Also, the minimal weighted automata can be computed in time $\mathcal{O}(n^3|\Sigma|)$.

In our case, the graph induced automata $\mathcal{A}(G, \chi)$ are completely symmetric in the sense that $[\![\mathcal{A}(G, \chi)]\!](w_1|w_2|\ldots|w_t) = [\![\mathcal{A}(G, \chi)]\!](w_t|w_{t-1}|\cdots|w_1)$ for $w_i$ in $\Sigma$, but also in the sense of

---

**Algorithm 1:** Layered canonical word search

**Input:** Number of layers $T$, ordered alphabet $(\Sigma, \leqslant)$, graph $G$, coloring $\chi$
**Output:** $S_0, S_1, \ldots S_T$ finite subsets of $\Sigma^*$

1 **for** $t \leftarrow 0$ **to** $T$ **do**
2     $S_t \leftarrow \varnothing$
3     $\mathcal{B}_t \leftarrow \varnothing$
4     queue $Q_t \leftarrow []$
5 $Q_0.push(\lambda)$
6 **for** $t \leftarrow 0$ **to** $T$ **do**
7     **while not** $Q_t.empty()$ **do**
8        $\boldsymbol{w} \leftarrow Q_t.pop()$
9        $\gamma \leftarrow \boldsymbol{W}_t(G, \chi)[-, \boldsymbol{w}]$
10        **if** $\mathrm{rank}\,(\mathcal{B}_t \cup \{\gamma\}) > |\mathcal{B}_t|$ **then**
11           $\mathcal{B}_t \leftarrow \mathcal{B}_t \cup \{\gamma\}$
12           $S_t \leftarrow S_t \cup \{\boldsymbol{w}\}$
13           **if** $t < T$ **then**
14              **foreach** $a$ **in** $\Sigma$ **do**
15                 $Q_{t+1}.push(\boldsymbol{w}a)$

16 **return** $S_0, S_1, \ldots, S_T$

---

that the initial vector can be interchanged as $\alpha^\top = \omega = \mathbf{1}$, and similarly matrices $\boldsymbol{M}(a) = \boldsymbol{M}(a)^\top$ and $\boldsymbol{M}(a|b) = \boldsymbol{M}(b|a)^\top$. It follows that the forward and backward steps of the automata minimization described are spanning the same vector space. And thus if there is a minimal automata $A_1 = (Q_1, \overline{\Sigma}, \boldsymbol{M}_1, \mathbf{1}^\top, \mathbf{1})$ for the graph induced automata $\mathcal{A}(G, \chi)$, it holds that there is a matrix of a full rank $\boldsymbol{F}$ in $\mathbb{R}^{V \times Q_1}$ such that for all $a$ in $\Sigma$:

$$\mathbf{1}^\top \boldsymbol{F} = \mathbf{1}^\top, \quad \mathbf{1} = \boldsymbol{F}\mathbf{1}, \quad \boldsymbol{F}\boldsymbol{M}_1(a) = \boldsymbol{M}(a)\boldsymbol{F}, \quad \boldsymbol{F}\boldsymbol{M}_1(a|b) = \boldsymbol{M}(a|b)\boldsymbol{F}. \tag{12}$$

In general, by e.g. Kiefer (2020), if we have two minimal automata $\mathcal{A}_i = (Q_i, \overline{\Sigma}, \boldsymbol{M}_i, \alpha_i, \omega_i)$ for $i = 1, 2$, then there is an invertible matrix $\boldsymbol{Q}$ in $\mathbb{R}^{Q_2 \times Q_1}$ such that for all $a$ in $\Sigma$:

$$\alpha_2 \boldsymbol{Q} = \alpha_1, \quad \omega_2 = \boldsymbol{Q}\omega_1, \quad \boldsymbol{Q}\boldsymbol{M}_2(a) = \boldsymbol{M}_1(a)\boldsymbol{Q}, \quad \boldsymbol{Q}\boldsymbol{M}_2(a|b) = \boldsymbol{M}_1(a|b)\boldsymbol{Q}. \tag{13}$$

Inspired by the minimization procedure of weighed automata, also cf. Kiefer et al. (2013), we propose a similar procedure *layered canonical word search*, see Algorithm 1, to compute the canonical word subsets $S_t(G, \chi)$ for all $t$ in $\mathbb{N}$, as given in Section 3.1. The main distinction from the automata minimization algorithm is that we keep separate queue, base, and the word subset for each layer $t$, and thus we are ensuring linear independence for each layer $t$ separately.

We recall the Conditions given in Section 3.1: The definition proceeds inductively: $S_0(G, \chi) = \{\lambda\} = \Sigma^0$, and for known $S_t(G, \chi)$, the set $S_{t+1}(G, \chi) \subseteq \Sigma^{t+1}(G, \chi)$ satisfies the following:

   (i) for every $\boldsymbol{a}c$ in $S_{t+1}(G, \chi)$ there is $\boldsymbol{a}$ in $S_t(G, \chi)$ (*prefix-closedness*),

   (ii) the columns of $\boldsymbol{W}_{t+1}$ induced by $S_{t+1}(G, \chi)$ are *linearly independent*, and

   (iii) the set $S_{t+1}(G, \chi)$ is *lexicographically minimal* among other sets satisfying (i) and (ii).

**Lemma A.7** (Correctness)**.** *Let $G$ be a graph, $\chi$ a $\Sigma$-coloring on $G$ then the Algorithm 1 computes the canonical word subsets $S_t(G, \chi)$, satisfying Conditions (i), (ii), and (iii), for all $t$ in $\mathbb{N}$.*

*Proof.* We proceed by induction on $t$. It is observed that in the for-loop of the algorithm ranging over $t$, we only add to the list $S_t$ and to the set $\mathcal{B}_t$, working only with the elements from the queue $Q_t$, and adding new elements to the queue $Q_{t+1}$ based on the words we added to $S_t$. For the base case $t = 0$, the queue $Q_0$ is initialized with the empty word $\lambda$, for which $\boldsymbol{W}_0[-, \lambda]$ is the all-one vector $\mathbf{1}$, and as the base $\mathcal{B}_0$ is empty and its rank 0. Thus, we have $\mathcal{B}_0 = \{\mathbf{1}\}$ and $S_0 = \{\lambda\}$.

For the induction step, we assume that the algorithm computes $S_t$ and $\mathcal{B}_t$ correctly. In the beginning of the $(t + 1)$-th iteration, the queue $Q_{t+1}$ contains the words of length $t + 1$, of the form $\boldsymbol{w}a$ for

all $a$ in $\Sigma$ and $\boldsymbol{w}$ in $S_t$, thus satisfying the condition (i). If we add $\gamma = \boldsymbol{W}_{t+1}[-, \boldsymbol{w}a]$ to $\mathcal{B}_{t+1}$, and $\boldsymbol{w}$ to $S_{t+1}$, then we have $\mathrm{rank}(\mathcal{B}_{t+1} \cup \{\gamma\}) > |\mathcal{B}_{t+1}|$, thus words in $S_{t+1}$ are satisfying condition (ii). The last condition (iii) follows from the fact that always adding to $Q_{t+1}$ possible candidates by the foreach-loop over $\Sigma$ in a lexicographical order, and we keep the order while processing the queue. $\qquad\square$

**Lemma A.8** (Complexity). *There is an implementation of Algorithm 1 that for a given $T$ in $\mathbb{N}$, computes the canonical word subsets $S_1, S_2, \ldots, S_T$, in time $\mathcal{O}(Tn^\omega)$, where $n$ is the number of vertices in $G$.*

*Proof. 1. Quartic bound.* We focus on the cost of an iteration of the for-loop over $t$, the base case $t = 0$ is trivially $\mathcal{O}(|\Sigma| + n)$. The size of the queue $Q_{t+1}$ is exactly $|S_t||\Sigma| \leqslant n|\Sigma|$. For every word $\boldsymbol{w}a$ in $Q_{t+1}$, we compute the corresponding vector of walk incidence matrix $\boldsymbol{W}_t[-, \boldsymbol{w}a]$ in $\mathbb{R}^{V(G)}$, from the vector $\boldsymbol{W}_t[-, \boldsymbol{w}]$ in for the word $\boldsymbol{w}$ in $S_t$ as shown in the proof Lemma A.5, by multiplying by matrix $\boldsymbol{P}_a$ if $t = 1$, and by $\boldsymbol{P}_a\boldsymbol{A}$ if $t \geqslant 2$. The expensive part of the algorithm is the computation of the rank of $\mathcal{B}_{t+1} \cup \{\gamma\}$, this can be done in $\mathcal{O}(n^2)$ time, if we maintain the representation of the linearly independent vectors of $\mathcal{B}_t$ in a row echelon form. Thus, for a limit $T$ in $\mathbb{N}$, we have $T$ iterations of the for-loop over $t$, which has the complexity of $\mathcal{O}(|S_t||\Sigma| + |S_t|n^2 + |S_{t+1}||\Sigma|) = \mathcal{O}(n^3|\Sigma|)$. And finally, the total complexity of the algorithm is $\mathcal{O}(Tn^3|\Sigma|)$. This yields $\mathcal{O}(Tn^4)$ using that $|\Sigma| \leqslant n$.

*2. Cubic bound.* To improve upon the above quartic bound, we further exploit the structure of our transition matrices $\boldsymbol{P}_a\boldsymbol{A}$ and $\boldsymbol{P}_a\boldsymbol{I}$. In each iteration over $t$ Partition matrices on the left side of $\boldsymbol{A}$ effectively zero out rows of vertices that are not of color $a$. Therefore, instead of performing for a given base $\boldsymbol{B}_t$ multiplication $\boldsymbol{P}_a\boldsymbol{A}\boldsymbol{B}_t$ for all $a$ in $\Sigma$, we only multiply by $\boldsymbol{A}\boldsymbol{B}_t$ and then choose the row-submatrix $\boldsymbol{A}_a$ of shape $V_a \times V$ where $V_a = \{u \mid u \in V, \chi(u) = a\}$. To that end, for each $\boldsymbol{A}_a$ we select the first subset of independent columns for $S_t$. This is without loss on generality since the row partition ($V_{a_1} \sqcup \cdots \sqcup V_{a_{|\Sigma|}} = V$) gives orthogonal decomposition of the columns space. Using the row echelon form technique as above, this accounts for $\mathcal{O}(n|V_a|^2)$ steps.

In total, we sum $\sum_a n|V_a|^2$ over all colors in $\Sigma$, and since $n\sum_a |V_a|^2 \leqslant n \cdot n^2$, we get an algorithm of complexity $\mathcal{O}(Tn^3)$.

*3. QR-bound.* To obtain this bound, we focus on the linear dependence queries that cost $\mathcal{O}(|V_a|^2)$ each. Instead of running a greedy loop over columns for each $\boldsymbol{A}_a$, we use matrix decomposition.

Recall a general **QR-decomposition** of square and/or tall matrix $\boldsymbol{X}$ of shape $k \times l$ such that $k \geqslant l$. It computes a square matrix $\boldsymbol{Q}$ of shape $k \times k$ and an upper-triangular matrix $\boldsymbol{R}$ of shape $k \times l$ such that $\boldsymbol{X} = \boldsymbol{Q}\boldsymbol{R}$. Diagonal $\delta$ of matrix $\boldsymbol{R}$ is a vector of length $l$, where $\delta[j] = \boldsymbol{R}[j, j]$ is zero if and only if the $j$-th column of $\boldsymbol{X}$ is linearly dependent on the previous columns.

It remains to clarify usage of QR-decomposition in our case. Matrices $\boldsymbol{A}_a$ of shape $V_a \times V$ such that $|V_a| \leqslant n$, are rather wide than tall. Therefore, we instead apply the following iterative approach:

Initially, duplicate rows of $\boldsymbol{A}_a$ to obtain matrix $\boldsymbol{A}_a^{(0)}$ of shape $V_a^{(0)} \times V$, where $V_a^{(0)} = V_a \uplus V_a$. Next, split columns of $\boldsymbol{A}_a^{(0)}$ into $m_0 := \lceil n/|V_a^{(0)}| \rceil$ square blocks $\boldsymbol{A}_{a,1}^{(0)}, \ldots, \boldsymbol{A}_{a,m_0}^{(0)}$, padding the last one with zero columns if needed.

In the $s$-th step, compute QR-decompositon for each square matrix $\boldsymbol{A}_{a,i}^{(s)}$ independently, selecting its linear independent columns. Crucially, due to the initial row duplication, $\mathrm{rank}(\boldsymbol{A}_{a,i}^{(s)}) \leqslant \frac{1}{2}|V_a^{(s)}|$, hence obtaining at most $\frac{1}{2}|V_a^{(s)}|$ of independent columns. From consecutive blocks $\boldsymbol{A}_{a,2i'}^{(s)}$ and $\boldsymbol{A}_{a,2i'+1}^{(s)}$, construct one square block $\boldsymbol{A}_{a,i'}^{(s+1)}$ containing at most $2 \cdot \frac{1}{2}|V_a^{(s)}|$ selected independent columns (padding rest with zeros if needed) of $\boldsymbol{A}_{a,2i'}^{(s)}$ and $\boldsymbol{A}_{a,2i'+1}^{(s)}$. This way, we obtain $m_{s+1} := \lceil \frac{m_s}{2} \rceil$ blocks with rows remaining duplicated for the $(s + 1)$-step.

Since the number of blocks $m_s$ halves after every step, after a finite number of steps $S$, we obtain a single square block. Its final QR-decomposition gives the lexicographically first column base marking the desired columns of $\boldsymbol{A}$ to obtain $B_{t+1}$. During this process we perform at most $\sum_{s=0}^{S} m_s = \sum_{s=0}^{S} \lceil m_0 \cdot 2^{-s} \rceil \leqslant 3m_0$ decompositions. Each decomposition has cost $\mathcal{O}((2|V_a|)^{\omega+\epsilon})$, as shown by Demmel et al. (2007), for any $\epsilon > 0$, where $\omega < 3$ is the exponent of matrix multiplication.

Finally, accounting for all row partitions and considering for a suitable constant $C$, we have

$$\sum_{a \in \Sigma} 3\lceil n/|V_a^{(0)}|\rceil (2|V_a^{(0)}|)^{\omega+\epsilon} \leqslant n \sum_{a \in \Sigma} C|V_a|^{\omega+\epsilon-1} \leqslant n \cdot Cn^{\omega+\epsilon-1} \leqslant Cn^{\omega+\epsilon}.$$

Therefore, to obtain $B_{t+1}$ from $B_t$, we perform $\mathcal{O}(n^{\omega+\epsilon})$ operations and $\mathcal{O}(Tn^{\omega+\epsilon})$ in total. $\quad\square$

Note that the last bound in the above proof of Lemma A.8 is not only of theoretical interest, but also practical one. The QR-decomposition is a standard building block of many numerical libraries (e.g., LAPACK and SciPy), providing highly optimized implementations are available. Moreover, splitting wide submatrix into a batch of square blocks allows for highly parallel processing of the algorithm on modern hardware architectures.

**Theorem 3.1.** *Let $\chi$ be a $\Sigma$-coloring on graph $G$ with $n$ vertices, and $T$ in $\mathbb{N}$ a limit then the canonical subsets $(S_t)_{t=0}^{T}$, (satisfying (i), (ii), and (iii)), are computable in time $\mathcal{O}(Tn^{\omega+\epsilon})$ for any $\epsilon > 0$ where $\omega$ is the exponent of matrix multiplication.*

*Proof.* We use the Algorithm 1, which is correct by Lemma A.7 and satifying the time complexity by Lemma A.8. $\quad\square$

In the following part, we prove two statements from the end of Section 3. For $t \geqslant 0$, we recall that the columns of matrix $B_t = B_t(G, \chi)$ span the same space as the columns of the walk incidence matrix $W_t(G, \chi)$. The columns of $B_t$ are indexed by the words in $S_t(G, \chi)$. Let us denote by $z_t = 00 \cdots 0$ a constant word of length $t$.

**Observation 3.2.** *Let $\chi_{\text{triv}}$ be the $\{0\}$-coloring on a graph $G$ with at least one edge. Then for every $t \geqslant 0$: (a) it holds that $|S_t| = 1$; (b) the only entries, $C_t^I[z_t, z_{t+1}] = \frac{\mathbf{1}^\top A^{2t+1} \mathbf{1}}{\mathbf{1}^\top A^{2t} \mathbf{1}}$, and $C_t^A[z_t, z_{t+1}] = \frac{\mathbf{1}^\top A^{2t+2} \mathbf{1}}{\mathbf{1}^\top A^{2t} \mathbf{1}}$.*

*Proof.* The first part (a) follows from the fact that the only word in $S_t(G, \chi_{\text{triv}})$ is $z_t$, and thus every queue $Q_{t+1}$ of the Algorithm 1 contains exactly the one word $z_{t+1}$. Note that the only partition matrix $P_a = I$.

For the second part (b), we follow the definition of the matrices $C_t^I$ and $C_t^A$ in $\mathbb{R}^{\{z_t\} \times \{z_{t+1}\}}$:

$$\begin{aligned}
C_t^M[z_t, z_{t+1}] = \mathbf{1}C_t^M \mathbf{1}^\top = B_t^+ M B_{t+1} &= \\
&= ((P_a A P_a)^t \mathbf{1})^+ M((P_a A P_a)^{t+1} \mathbf{1}) = \\
&= (A^t \mathbf{1})^+ M(A^{t+1} \mathbf{1}) = \\
&= ((A^t \mathbf{1})^\top A^t \mathbf{1})^{-1} \cdot (A^t \mathbf{1})^\top M A^t \mathbf{1} = \\
&= (\mathbf{1}^\top A^{2t} \mathbf{1})^{-1} \cdot \mathbf{1}^\top A^t M A^{t+1} \mathbf{1}.
\end{aligned}$$

By setting $M := I$ we obtain the result for the $C_t^I$, and by $M := A$ for the $C_t^A$. $\quad\square$

**Proposition 3.3.** *Let $\chi_{\text{id}}$ be the $V$-coloring on a graph $G = (V, E)$ with $n$ vertices. By $v_{t,u}$, we denote the (unique) word in $S_t$ with the last color $u$ in $V$. Then for every $t \geqslant 1$: (a) it holds that $|S_0| = 1$, and $|S_t| = |V|$; (b) for entries $C_0^I[\lambda, u] = \frac{1}{n}$, and $C_t^I[v_{t,u}, v_{t+1,v}] = I[u, v]$ and $C_t^A[v_{t,u}, v_{t+1,v}] = A[u, v]$.*

*Proof.* We denote the vertices $V$ by $\{u_1, u_2, \ldots, u_n\}$, which here coincides with the set of colors. To prove (a), we proceed by induction on $t$. For the base case $t = 0$, we have trivially $|S_0| = 1$, as $S_0 = \{\lambda\}$. For the case $t = 1$, we have $S_1 = V$, since $P_{u_i} = e_{u_i} e_{u_i}^\top$ for $u_i$ in $V$, and furthermore, $u_i$-th column of $B_1$ is $P_{u_i} \mathbf{1} = e_{u_i}$, and thus we have $B_1 = I$. For, the induction step, we assume that $S_t = \{v_{t,u} \mid u \text{ in } V\}$. Since $W_{t+1}[-, wu] = P_u A W_t[-, w]$, there is for each $u$ in $V$ a unique word $wu$ in $S_{t+1}$, and the base $\mathcal{B}_{t+1}$ is also canonical, that it $B_{t+1} = I$. Thus, we have $|S_{t+1}| = |V|$.

For the second part (b), we have $C_0^I = \mathbf{1}^+ I B_1 = (\mathbf{1}^\top \mathbf{1})^{-1} \mathbf{1}^\top I I = \frac{1}{n} \mathbf{1}^\top$. Next, for $t \geqslant 1$ and $M$ in $\{A, I\}$, we get $C_t^M = B_t^+ M B_{t+1} = I^+ M I = M$. $\quad\square$

### A.3 Constraints on Walk-Incidence Matrices

In the main text, we have noted that the matrices $\boldsymbol{W}_t(G, \chi)$ are not completely arbitrary. We now show in detail, how the structure of the matrices $\boldsymbol{W}_t(G, \chi)$ is influenced by the weighted finite automata $\mathcal{A}(G, \chi)$. Suppose we have two graphs $G$ and $H$, and that the automata $\mathcal{A}(G, \chi)$ and $\mathcal{A}(H, \chi)$ are equivalent, $[\![\mathcal{A}(G, \chi)]\!] = [\![\mathcal{A}(H, \chi)]\!]$. Then there is a minimal weighed automata $\mathcal{A}_1$ common for $\mathcal{A}(G, \chi)$ and $\mathcal{A}(H, \chi)$. By Equation 12 and Equation 13, there are matrices $\boldsymbol{F}^G$ in $\mathbb{R}^{V(G) \times Q_1}$ and $\boldsymbol{F}^H$ in $\mathbb{R}^{V(H) \times Q_1}$ mapping automata induced by $G$ and $H$ to the common minimal automata $\mathcal{A}_1$. Here, we denote the matrix between the two automata $\mathcal{A}(G, \chi)$ and $\mathcal{A}(H, \chi)$ by the following:

$$\boldsymbol{F} = (\boldsymbol{F}^G)(\boldsymbol{F}^H)^\top \text{ in } \mathbb{R}^{V(G) \times V(H)}.$$

Note that $\boldsymbol{F}\mathbf{1} = \boldsymbol{F}^G(\boldsymbol{F}^H)^\top \mathbf{1} = \boldsymbol{F}^G \mathbf{1} = \mathbf{1}$, and similarly $\mathbf{1}^\top \boldsymbol{F} = \mathbf{1}^\top$.

**Lemma A.9.** *Using the notation above, we have for every $t \geqslant 0$:*

$$\boldsymbol{W}_t(G, \chi) = \boldsymbol{F}\boldsymbol{W}_t(H, \chi).$$

*Proof.* As the $\Sigma$-coloring $\chi$ is fixed, we write $\boldsymbol{W}_t^G = \boldsymbol{W}_t(G, \chi)$ and $\boldsymbol{W}_t^H = \boldsymbol{W}_t(H, \chi)$. In addition, we denote the matrices of the first graph $\boldsymbol{A}^G$, $\boldsymbol{P}_a^G$, and similarly for the second graph $\boldsymbol{A}^H$, $\boldsymbol{P}_a^H$, given that $a$ in $\Sigma$. We proceed by induction on $t$, proving the following statement for each word $\boldsymbol{w}$ in $\Sigma^t$: $\boldsymbol{W}_t^G[-, \boldsymbol{w}] = \boldsymbol{F}\boldsymbol{W}_t^H[-, \boldsymbol{w}]$.

For $t = 0$, from Lemma A.5 it follows that

$$\boldsymbol{W}_0^G[-, \lambda] = \mathbf{1} = \boldsymbol{F}\mathbf{1} = \boldsymbol{F}\boldsymbol{W}_0^H[-, \lambda].$$

Moreover, using Equation 12, for $t = 1$, we get for any $a$ in $\Sigma$

$$\boldsymbol{W}_1^G[-, a] = \boldsymbol{P}_a^G \mathbf{1} = \boldsymbol{P}_a^G \boldsymbol{F}\mathbf{1} = \boldsymbol{F}\boldsymbol{P}_a^H \mathbf{1} = \boldsymbol{F}\boldsymbol{W}_1^H[-, a].$$

For the induction step, we assume that the lemma holds for all words of length $t$. We take a word $\boldsymbol{w}a$ of length $t + 1$, which denotes the last color of $\boldsymbol{w}$ by $b$. It follows:

$$\begin{aligned}
\boldsymbol{W}_{t+1}^G[-, \boldsymbol{w}a] = \boldsymbol{P}_a^G \boldsymbol{A}^G \boldsymbol{P}_b^G \boldsymbol{W}_t^G[-, \boldsymbol{w}] &= \\
= \boldsymbol{P}_a^G \boldsymbol{A}^G \boldsymbol{P}_b^G \boldsymbol{F}\boldsymbol{W}_t^H[-, \boldsymbol{w}] &= \\
= \boldsymbol{F}\boldsymbol{P}_a^H \boldsymbol{A}^H \boldsymbol{P}_b^H \boldsymbol{W}_t^H[-, \boldsymbol{w}] &= \\
= \boldsymbol{F}\boldsymbol{W}_{t+1}^H[-, \boldsymbol{w}a],
\end{aligned}$$

which finishes the proof. $\square$

**Characterization of Efficient Matrices.** We recall that for graph $G$ and a coloring $\chi$, the *graph invariant* of efficient matrices $\mathcal{I}_{\text{EA}}(G, \chi)$ is defined as the set of pairs of matrices

$$\mathcal{I}_{\text{EA}}(G, \chi) = \left\{ \left( \boldsymbol{C}_t^{\boldsymbol{A}}, \boldsymbol{C}_t^{\boldsymbol{I}} \right) \mid 0 \leqslant t < n \right\},$$

to state the following lemma:

**Lemma A.10.** *For any coloring $\chi$ we have $\mathcal{I}_{EA}(-, \chi) \sqsubseteq [\![\mathcal{A}(-, \chi)]\!]$.*

*Proof.* Suppose we have two graphs $G$ and $H$ that are not distinguished by the semantics of their induced automata. Then there is a matrix a $\boldsymbol{F}$ in $\mathbb{R}^{V(G) \times V(H)}$ adjoining these two automata. Given a $\Sigma$-coloring $\chi$, and two graphs $G$, $H$ we have from the Lemma A.9 that $\boldsymbol{B}_t^G = \boldsymbol{F}\boldsymbol{B}_t^H$ for all $t \geqslant 0$. For $\boldsymbol{M}$ in $\{\boldsymbol{A}, \boldsymbol{I}\}$, we have

$$\begin{aligned}
\boldsymbol{C}_t^{\boldsymbol{M}}(G, \chi) &= (\boldsymbol{B}_t^G)^+ \boldsymbol{M}^G \boldsymbol{B}_{t+1}^G = (\boldsymbol{B}_t^G)^+ \boldsymbol{M}^G \boldsymbol{F}\boldsymbol{B}_{t+1}^H \\
&= (\boldsymbol{B}_t^G)^+ \boldsymbol{F}\boldsymbol{M}^H \boldsymbol{B}_{t+1}^H = (\boldsymbol{B}_t^G)^+ \boldsymbol{F} \left( \boldsymbol{B}_t^H (\boldsymbol{B}_t^H)^+ \right) \boldsymbol{M}^H \boldsymbol{B}_{t+1}^H \\
&= (\boldsymbol{B}_t^G)^+ \boldsymbol{F}\boldsymbol{B}_t^H (\boldsymbol{B}_t^H)^+ \boldsymbol{M}^H \boldsymbol{B}_{t+1}^H = (\boldsymbol{B}_t^G)^+ \left( \boldsymbol{F}\boldsymbol{B}_t^H \right) (\boldsymbol{B}_t^H)^+ \boldsymbol{M}^H \boldsymbol{B}_{t+1}^H \\
&= (\boldsymbol{B}_t^G)^+ \left( \boldsymbol{B}_t^G \right) (\boldsymbol{B}_t^H)^+ \boldsymbol{M}^H \boldsymbol{B}_{t+1}^H = (\boldsymbol{B}_t^H)^+ \boldsymbol{M}^H \boldsymbol{B}_{t+1}^H.
\end{aligned}$$

The final expression is equal to $\boldsymbol{C}_t^{\boldsymbol{M}}(H, \chi)$, which finishes the proof. $\square$

In the previous lemma, we have shown that the efficient matrices are invariant under the equivalence of the automata. To state the other direction, we first give a simpler lemma for $\chi = \chi_{\text{triv}}$ to illustrate the structure of the more general lemma.

**Lemma A.11.** *Let $\chi_{\text{triv}}$ be the trivial coloring, then $\mathcal{I}_{EA}(-, \chi_{\text{triv}}) \sqsupseteq [\![\mathcal{A}(-, \chi_{\text{triv}})]\!]$.*

*Proof.* As shown in Theorem 3.2, the matrices $C_t^I(G, \chi_{\text{triv}})$ and $C_t^A(G, \chi_{\text{triv}})$ are of a single entry $(\mathbf{1}^\top A^{2t}\mathbf{1})^{-1}\mathbf{1}^\top A^{2t+1}\mathbf{1}$ and $(\mathbf{1}^\top A^{2t}\mathbf{1})^{-1}\mathbf{1}^\top A^{2t+2}\mathbf{1}$, respectively. We let the symbol $a = 0$, so that $\chi_{\text{triv}}$ is the $\{a\}$-coloring on $G$. Note that $|V(G)| = n = |\mathcal{I}_{EA}|$.

Next, the transition matrices of $\mathcal{A}(G, \chi_{\text{triv}})$ are $M(a) = P_a = I$, and $M(a|a) = P_a A P_a = A$, and generally for $k$-th repetition of $a$, $M(a|\cdots|a) = A^k$. Thus, the semantics of the automata $\mathcal{A}(G, \chi_{\text{triv}})$ is determined by the formal series given as $a^k \mapsto \mathbf{1}^\top A^k \mathbf{1}$.

For simplicity, we define the following three functions for all $m$ in $\mathbb{N}$:

$$f(m) = \mathbf{1}^\top A^m \mathbf{1}, \quad g_1(m) = \frac{\mathbf{1}^\top A^{2m+1}\mathbf{1}}{\mathbf{1}^\top A^{2m}\mathbf{1}}, \quad g_2(m) = \frac{\mathbf{1}^\top A^{2m+2}\mathbf{1}}{\mathbf{1}^\top A^{2m}\mathbf{1}}.$$

In the language of such notation, it is sufficient to show all values of $f$ are determined by values of function $g_1, g_2$ and single value $n$. Moreover, it is sufficient to show values of $f(k)$ for $k < n$, since for values $k \geqslant n$, we can apply Cayley-Hamilton theorem and express $A^k$ using the powers of $A$ up to $n-1$.

We shall proceed by the following induction on $k$. For the base case $k = 0$, we have $f(0) = n$. For the induction step, assume that the value $f(k')$ are determined for each $k' < k$, we distinguished two cases: (a) $k = 2l + 1$ is odd, and (b) $k = 2l + 2$ is even.

For the odd case (a), we have $f(k) = f(2l + 1) = f(2l)g_1(l)$, and for the even case (b), we have $f(k) = f(2l + 2) = f(2l)g_2(l)$. Since $f(2l)$ is known from the induction hypothesis, we can compute $f(k)$ from $f(2l)$ and $g_1(l)$ or $g_2(l)$. $\qquad\square$

Here, before we state a more general variant of Lemma A.11 for any coloring $\chi$, we introduce the following notation. Let $\chi$ be a $\overline{\Sigma}$-coloring on a graph $G$, and $\mathcal{I}_{EA}(G, \chi)$ the invariant of efficient matrices $C_t^A(G, \chi)$, $C_t^I(G, \chi)$ in $\mathbb{R}^{S_t \times S_{t+1}}$ for $0 \leqslant t < n$. Let $a, b$ in $\Sigma$, and $t$ in $\mathbb{N}$ such that $0 \leqslant t < n$, we define $M_t(a)$ in $\mathbb{R}^{S_t \times S_{t+1}}$ by setting

$$M_t(a)[\boldsymbol{w}_1 a, \boldsymbol{w}_2 a] := C_t^I(G, \chi)[\boldsymbol{w}_1 a, \boldsymbol{w}_2 a], \tag{14}$$

for all $\boldsymbol{w}_1 a$ in $S_t$, and $\boldsymbol{w}_2 a$ in $S_{t+1}$ and letting all other entries be zero. Similarly, we define $M_t(a|b)$ in $\mathbb{R}^{S_t \times S_{t+1}}$ by setting

$$M_t(a|b)[\boldsymbol{w}_1 a, \boldsymbol{w}_2 ab] := C_t^A(G, \chi)[\boldsymbol{w}_1 a, \boldsymbol{w}_2 ab], \tag{15}$$

for all $\boldsymbol{w}_1 a, \boldsymbol{w}_2 ab$ in $S_{t+1}$ and letting all other entries be zero.

**Lemma A.12.** *Let $G$ be a graph and $\chi$ a $\Sigma$-coloring on $G$. Then for all $a, b$ in $\Sigma$ and each $t$ in $\mathbb{N}$ it holds that*

$$M_t(a) = B_t^+ P_a B_{t+1}, \quad and \quad M_t(a|b) = B_t^+ P_b A P_a B_{t+1},$$

*where $B_t = B_t(G, \chi)$.*

*Proof.* We start with the proof of the first identity. Let us fix $t$ in $\mathbb{N}$, then

$$C_t^I(G, \chi) = B_t^+ \left(\textstyle\sum_a P_a\right) B_{t+1} = \sum_{a \in \Sigma} B_t^+ P_a B_{t+1} = \sum_{a \in \Sigma} M_t(a). \tag{16}$$

On the other hand, since $B_t[-, \boldsymbol{w}_1 a] = (P_a B_t)[-, \boldsymbol{w}_1 a]$ for each $\boldsymbol{w}_1 a$ in $S_t$, we get for each $\boldsymbol{w}_1 x$ in $S_t$, and each $\boldsymbol{w}_2 y$ in $S_{t+1}$, that it holds

$$((B_t^\top B_t)^{-1} B_t^\top P_a B_{t+1})[\boldsymbol{w}_1 x, \boldsymbol{w}_2 y] = ((B_t^\top B_t)^{-1} B_t^\top P_x^\top P_a P_y B_t)[\boldsymbol{w}_1 x, \boldsymbol{w}_2 y],$$

from which it follows by Proposition A.2 being zero if $x \neq a$ or $y \neq a$. In conjunction with Eq. (16), it follows that left-hand side and right-hand side coincide entry-wise and thus $M_t(a) = B_t^+ P_a B_{t+1}$. The latter identity, we show similarly by fixing $t$ in $\mathbb{N}$,

$$
\begin{aligned}
C_t^A(G, \chi) &= B_t^+ (\sum_{a,b} P_b A P_a) B_{t+1} \\
&= \sum_{a,b \in \Sigma} B_t^+ P_x P_b A P_a P_y B_{t+1} = \sum_{a,b \in \Sigma} M_t(a|b).
\end{aligned}
\tag{17}
$$

Similarly, we have for each $w_1 x$ in $S_t$, and each $w_2 y$ in $S_{t+1}$,

$$
(B_t^+ P_a A P_b B_{t+1})[w_1 x, w_2 y] = (B_t^+ P_x^\top P_a A P_b P_y B_t)[w_1 x, w_2 y],
$$

which is zero if $x \neq a$ or $y \neq b$. In conjunction with Eq. (17), it follows that left-hand side and right-hand side coincide entry-wise and thus $M_t(a|b) = B_t^+ P_b A P_a B_{t+1}$. $\qquad \square$

**Lemma A.13.** *For any coloring $\chi$ we have $\mathcal{I}_{EA}(-, \chi) \sqsupseteq [\![\mathcal{A}(-, \chi)]\!]$.*

*Proof.* We first show that efficient matrices $\mathcal{I}_{EA} = \mathcal{I}_{EA}(G, \chi)$, for some graph $G$, encode a specific layered computation of minimal automata with the semantics of $\mathcal{A}(G, \chi)$. A direct consequence of Lemma A.12 is that by selecting the entries of efficient matrices in $\mathcal{I}_{EA}$ as in Eq. (14) and Eq. (15), we can obtain $M_t(a)$ and $M_t(a|b)$ for all $a, b$ in $\Sigma$ and where $0 \leqslant t < n$.

From the size of $\mathcal{I}_{EA}$, we get $n = |\mathcal{I}_{EA}| = |V(G)|$. For a word $w = w_1 w_2 \cdots w_{t+1}$ in $\Sigma^{t+1}$, we define the expression $\gamma_w(\mathcal{I}_{EA})$ in $\mathbb{R}^{\{\lambda\} \times S_{t+1}}$ as follows:

$$
\gamma_w(\mathcal{I}_{EA}) = n \cdot M_0(w_1|w_2) M_1(w_2|w_3) \cdots M_t(w_t|w_{t+1}).
$$

Next, we take $B_t = B_t(G, \chi)$ and $M : \Sigma \to \mathbb{R}^{V \times V}$ the transition matrices of $\mathcal{A}(G, \chi)$, it follows from Lemma A.12 that

$$
\begin{aligned}
\gamma_w(\mathcal{I}_{EA}) &= n B_0^+ P_{w_1} A P_{w_2} B_1 B_1^+ P_{w_2} A P_{w_3} B_2 B_2^+ \cdots P_{w_t} A P_{w_{t+1}} B_t \\
&= n \mathbf{1}^+ P_{w_1} A P_{w_2} \cdots P_{w_t} A P_{w_{t+1}} B_t \\
&= \mathbf{1}^T M(\overline{w}) B_t,
\end{aligned}
$$

where $\overline{w} = w_1|w_2|\cdots|w_{t+1}$ in $\overline{\Sigma}^{t+1}$.

We prove for every $t$, such that $0 \leqslant t < n$, that weight of any word $\overline{w}$ in $\overline{\Sigma}^t$, $[\![\mathcal{A}(G, \chi)]\!](\overline{w}) = \mathbf{1}^T M(\overline{w}) \mathbf{1}$ can be expressed a linear combination of $\gamma$-vectors that depend only on $\mathcal{I}_{EA}(G, \chi)$. We shall use the following notation, for the word $w = w_1 w_2 \cdots w_m$ in $\Sigma^m$, we denote the word $w_1|w_2|\cdots|w_m$ in $\overline{\Sigma}^m$ by $\overline{w}$, and the reverse words $w_m w_{m_1} \cdots w_1$ in $\Sigma^m$ by $w^R$, and finally, $w_m|w_{m-1}|\cdots|w_1$ in $\overline{\Sigma}^m$ by $\overline{w}^R$.

For the base case $t = 0$, we have $\overline{w} = \lambda$, and thus $\mathbf{1}^T M(\lambda) \mathbf{1} = \mathbf{1}^T \mathbf{1} = n$, which is equal to $\gamma_\lambda(\mathcal{I}_{EA}) \mathbf{1} = n \mathbf{1}^+ \mathbf{1} = n$.

In the induction step $t \geqslant 1$, we distinguish two cases: (a) $t = 2l$ is even, and (b) $t = 2l + 1$ is odd. For the even case (a), can write the given word as $\overline{w} = \overline{w}_p|a|\overline{w}_s$, for the suitable $w_p$ and $a w_s$ in $\Sigma^l$. Since the columns of $W_l = W_l(G, \chi)$ are spanned by the columns of $B_l$, there is a vector $x$ in $\mathbb{R}^{S_l}$ such that by Lemma A.5 it holds that

$$
M(a|\overline{w}_s) \mathbf{1} = W_l[-, w_s^R a] = B_l x.
$$

Note that the vector $x$ is independent of the choice of the base of the transition matrices $M$, as for any base-changing matrix $F$ in $\mathbb{R}^{V(H) \times V(G)}$, for some graph $H$ with $\mathcal{I}_{EA}(H, \chi) = \mathcal{I}_{EA}$, it holds $W_l(H, \chi) = F W_l = F B_l x = B_l(H, \chi) x$. Next, we obtain

$$
\begin{aligned}
\mathbf{1}^T M(\overline{w}) \mathbf{1} &= \mathbf{1}^T M(\overline{w}_p|a \cdot a|\overline{w}_s) \mathbf{1} \\
&= \mathbf{1}^T M(\overline{w}_p|a) M(a|\overline{w}_s) \mathbf{1} \\
&= \mathbf{1}^T M(\overline{w}_p|a) B_l x = \gamma_{\overline{w}_p|a}(\mathcal{I}_{EA}) x.
\end{aligned}
$$

Similarly, for the odd case (b), we write $\overline{\boldsymbol{w}} = \overline{\boldsymbol{w}}_p|a|\overline{\boldsymbol{w}}_s$, for the suitable $\boldsymbol{w}_p$ in $\Sigma^l$ and $a\boldsymbol{w}_s$ in $\Sigma^{l+1}$. Analogically, we can find a vector $\boldsymbol{x}$ in $\mathbb{R}^{S_{l+1}}$ such that $\boldsymbol{M}(a|\overline{\boldsymbol{w}}_s)\mathbf{1} = \boldsymbol{B}_{l+1}\boldsymbol{x}$ to obtain

$$
\begin{aligned}
\mathbf{1}^T \boldsymbol{M}(\overline{\boldsymbol{w}})\mathbf{1} &= \mathbf{1}^T \boldsymbol{M}(\overline{\boldsymbol{w}}_p|a \cdot a \cdot a|\overline{\boldsymbol{w}}_s)\mathbf{1} \\
&= \mathbf{1}^T \boldsymbol{M}(\overline{\boldsymbol{w}}_p|a)\boldsymbol{M}(a)\boldsymbol{M}(a|\overline{\boldsymbol{w}}_s)\mathbf{1} \\
&= \mathbf{1}^T \boldsymbol{M}(\overline{\boldsymbol{w}}_p|a)\boldsymbol{M}(a)\boldsymbol{B}_{l+1}\boldsymbol{x} \\
&= \gamma_{\overline{\boldsymbol{w}}_p|a}(\mathcal{I}_{\text{EA}})\boldsymbol{B}_l\boldsymbol{B}_l^+\boldsymbol{P}_a\boldsymbol{B}_{l+1}\boldsymbol{x} = \gamma_{\overline{\boldsymbol{w}}_p|a}(\mathcal{I}_{\text{EA}})\boldsymbol{M}_l(a)\boldsymbol{x}.
\end{aligned}
$$

Thus, we have shown that $\mathcal{I}_{\text{EA}} = \mathcal{I}_{\text{EA}}(G, \chi)$ determines the semantics of automata $\mathcal{A}(G, \chi)$, which is then identical for all graphs $H$ with $\mathcal{I}_{\text{EA}} = \mathcal{I}_{\text{EA}}(H, \chi)$. $\qquad\square$

**Theorem 4.2.** *For every coloring $\chi$ it holds that $\mathcal{I}_{EA}(-, \chi) \equiv \text{wr}(-, \chi)$.*

*Proof.* The proof follows from Lemma A.13 and Lemma A.10. $\qquad\square$

## B  HOMOMORPHISM COUNTS AND COLORED WALKS

This section contains the proofs for the statements of Section 4. We first show that homomorphism counts from caterpillars $\mathcal{C}_1$ into a given graph can be characterized by walk incidences with respect to the coloring $\chi_{\text{deg}}$. Our initial argument employs elementary techniques that are later generalized using methods from quantum graphs (Lovász and Szegedy, 2009; Dvořák, 2010) and logic (cf. (Immerman and Lander, 1990; Cai et al., 1992; Grohe, 2017)).

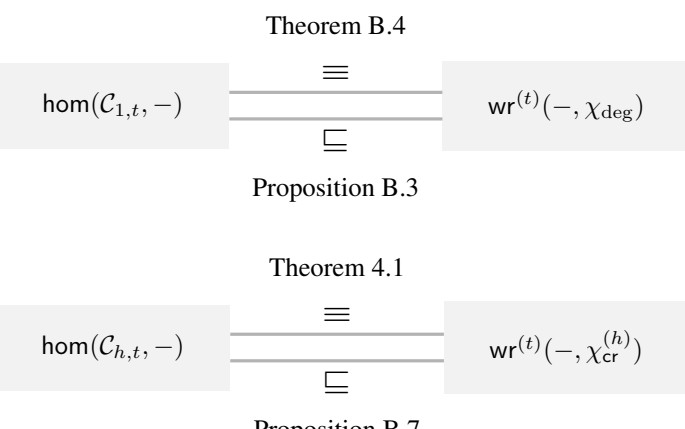

Figure 10: This section establishes the equivalence between homomorphism counts over caterpillars and colored walk refinement with respect to (specific) colorings. First, we focus on "folklore" caterpillars (top row), and then we generalize to $(h, t)$-caterpillars (bottom row) in Appendix B.1. For every $h, t \geqslant 0$, we relate the following two functions on graphs: homomorphism counts over $(h, t)$-caterpillars $\text{hom}(\mathcal{C}_{h,t}, -)$, and colored walk refinement $\text{wr}^{(t)}(-, \chi_{\text{cr}}^{(h)})$.

We begin by proving two technical lemmas.

**Lemma B.1.** *For every integer $t \geqslant 0$ and natural numbers $D_0, D_1, D_2, \ldots, D_t$ there are integral exponents $s_1, s_2 \ldots, s_t$ such that $D_0 \leqslant 2^{s_1}$, and such that all $t$-tuples of natural numbers $(d_1, d_2, \ldots, d_t)$ that satisfy $1 \leqslant d_i < D_i$ for $i$ in $[t]$, are injectively represented by the product $p = d_1^{s_1} \cdot d_2^{s_2} \cdots d_t^{s_t}$.*

*Proof.* We can instead take logarithm of $p$, since $\log$ is an injective function,

$$
\log_2 p = s_1 \log_2 d_1 + \cdots + s_t \log_2 d_t.
$$

The values of are called $\log_2 d_i$ in $\{\log_2 1, \log_2 2, \ldots, \log_2(D_i - 1)\}$ *digits* for $i$ in $[t]$. We construct a positional numeral system with a heterogeneous basis (i.e., $s_1, s_2, \ldots s_t$) special for each digit's position $i$. Since the intervals of digits at the $i$-th position are bounded by $\log_2 D_i$, we require to preserve the following inequality

$$s_{i+1} > \sum_{j=1}^{i} \log_2(D_j - 1) \cdot s_j.$$

To next meet the first condition $D_0 \leqslant 2^{s_1}$, we set $s_0 := \lceil \log_2 D_0 \rceil$. Furthermore, we obtain $s_{i+1} := 1 + \sum_{j=1}^{i} \lceil \log_2(D_j - 1) \rceil \cdot s_j$. Finally, we define $R_{t+1} = \log_2 p$ to inductively evaluate the following expression

$$\log_2 d_i = \frac{1}{s_i}\left(R_i - \sum_{j=0}^{i-1} s_j \log_2 d_j\right) \quad \text{where} \quad R_l = \frac{R_{l+1}}{s_{l+1}} - \left\lfloor \frac{R_{l+1}}{s_{l+1}} \right\rfloor,$$

for every $l$ in $[t]$, and every $i$ in $[t]$. Therefore, the values $s_i$ are sufficient. $\square$

**Lemma B.2.** *Let $\boldsymbol{x}$ be a $m$-tuple in $\mathbb{N}_{\geqslant 1}$ with mutually distinct elements, that is $\boldsymbol{x}[i] \neq \boldsymbol{x}[j]$ whenever $i \neq j$, and let $m$-tuples $\boldsymbol{a}, \boldsymbol{b}$ in $\mathbb{N}^m$ be such that $\boldsymbol{a} \neq \boldsymbol{b}$, that is $\boldsymbol{a}[i] \neq \boldsymbol{b}[i]$ for existing $i$ in $[m]$. Then there is $k$ in $\mathbb{N}$ such that*

$$\sum_{i=1}^{m} \boldsymbol{a}[i](\boldsymbol{x}[i])^k \neq \sum_{i=1}^{m} \boldsymbol{b}[i](\boldsymbol{x}[i])^k. \tag{18}$$

*Proof.* Denote tuple $\boldsymbol{a}$ by $(a_1, a_2, \ldots, a_m)$, $\boldsymbol{b}$ by $(b_1, b_2, \ldots, b_m)$, and the tuple $\boldsymbol{x}$ by $(x_1, x_2, \ldots, x_m)$. For a contradiction, suppose that for all $k$ in $\mathbb{N}$ the equality in equation 18 holds. Let us choose the index $j$ such that $|a_j - b_j| > 0$ and, importantly, such that $x_j$ is maximal (such $j$ exists only one since elements of $\boldsymbol{x}$ are distinct). Let us define functions $f$ and $g$ as follows

$$f(k) = \left|(a_j - b_j)x_j^k\right|, \quad g(k) = \left|\sum_{i \neq j}(b_i - a_i)x_i^k\right|. \tag{19}$$

As $k \to \infty$, the function $f$ grows faster than the function $g$. Therefore, there exists $k$ such that $f(k) > g(k)$, that is $(a_j - b_j)x_j^k > \sum_{i \neq j}(b_i - a_i)x_i^k$, which gives us the contradiction. $\square$

In Proposition B.3, we prove the formula for counting homomorphisms from the class $\mathcal{C}_1$ that uses the leg sequence (see Section 4 and Figure 5) of the caterpillar graph. Next, a *start graph* is a rooted tree of height at most 1.

Given a $(1, t)$-caterpillar $F$ in $\mathcal{C}_{1,t}$, we associate its leg sequence $(S_1, S_2, \ldots, S_t)$ of star graphs $S_i$, with the tuple $\boldsymbol{s}_F = (|S_1| - 1, |S_2| - 1, \ldots, |S_t| - 1)$ in $\mathbb{N}^t$, so that $|S_i| - 1$ is the number of leaves of the $i$-th leg of $F$. In terms of folklore caterpillars, the $i$-th entry of $\boldsymbol{s}_F$ is exactly the number of one-edge legs attached to the $i$-th vertex of the spine path. Importantly, every $(1, t)$-caterpillar graph $F$ is fully described and determined by $\boldsymbol{s}_F$. Furthermore, we recall that $\text{wr}^{(t)}(G, \chi_{\deg})$ is the multiset of all colored walks of length $t$ in $G$.

**Proposition B.3.** *Let $t$ in $\mathbb{N}$, then for every graph $G$ it holds*

$$\hom(\mathcal{C}_{1,t}, G)[F] = \sum_{\boldsymbol{w} \text{ in } \mathbb{N}^t} \text{wr}^{(t)}(\boldsymbol{w}) \cdot \prod_{i=1}^{t} \boldsymbol{w}[i]^{\boldsymbol{s}_F[i]}, \tag{20}$$

*for each $F$ in $\mathcal{C}_{1,t}$ with $\boldsymbol{s}_F$ in $\mathbb{N}^t$, where $\text{wr}^{(t)}(\boldsymbol{w}) = \text{wr}^{(t)}(G, \chi_{\deg})(\boldsymbol{w})$ for $\boldsymbol{w}$ in $\mathbb{N}^t$.*

*Proof.* Given a colored walk $\boldsymbol{w}$ in $\mathbb{N}^t$, consider its occurrence $\boldsymbol{u} = (u_1, u_2, \ldots, u_t)$ in $G$. We denote the vertices of the spine of $F$ that correspond to $\boldsymbol{s}_F$ by $l_1, l_2, \ldots, l_t$. Let $k_{\boldsymbol{u}}$ be the number of graph homomorphisms $\varphi : F \to G$, mapping the spine of $F$ to the occurrence $\boldsymbol{u}$, that is, such that $\varphi(l_i) = u_i$ for each $i$ in $[t]$.

Therefore, any two distinct homomorphisms contribute to $k_{\boldsymbol{u}}$ differ precisely by their mapping of vertices outside the spine, namely, the leaves of the stair graphs $S_i$ for $i$ in $[t]$.

Specifically, each leaf of $S_i$ can be mapped to any vertex in its neighborhood $N(u_i)$, hence, we have exactly $\boldsymbol{w}[i] = \deg_G(u_i)$ choices. Because $S_i$ contains $\boldsymbol{s}_F[i]$ such leaves, there are $\boldsymbol{w}[i]^{\boldsymbol{s}_F[i]}$ choices to map the legs of $\boldsymbol{P}_i$ independently of the other $\boldsymbol{P}_j$ ($j \neq i$). Therefore, we have $k_{\boldsymbol{u}} = \prod_{i=1}^{t} \boldsymbol{w}[i]^{\boldsymbol{s}_F[i]}$.

Finally, we sum over all independent $k_{\boldsymbol{u}}$ as follows:

$$\sum_{\boldsymbol{u} \text{ in } V(G)^t} k_{\boldsymbol{u}} = \sum_{\substack{\boldsymbol{u} \text{ occurrence} \\ \text{of } \boldsymbol{w} \text{ in } G}} \prod_{i=1}^{t} \boldsymbol{w}[i]^{\boldsymbol{s}_F[i]} = \sum_{\boldsymbol{w} \text{ in } \mathbb{N}^t} \mathsf{wr}^{(t)}(\boldsymbol{w}) \cdot \prod_{i=1}^{t} \boldsymbol{w}[i]^{\boldsymbol{s}_F[i]},$$

where in the second step, we distinguish occurrences $\boldsymbol{u}$ by their color $\boldsymbol{w}$, next, these occurrences are quantified by $\mathsf{wr}^{(t)}(\boldsymbol{w})$. Thus, we obtained an expression equal to the left-hand side of Equation 20. $\square$

**Theorem B.4.** *For every $t$ in $\mathbb{N}$, it holds that* $\mathsf{wr}^{(t)}(-, \chi_{\deg}) \equiv \hom(\mathcal{C}_{1,t}, -)$.

*Proof.* We first show $\mathsf{wr}^{(t)}(-, \chi_{\deg}) \sqsupseteq \hom(\mathcal{C}_{1,t}, -)$. Suppose that $\mathsf{wr}^{(t)}(G, \chi_{\deg}) = \mathsf{wr}^{(t)}(H, \chi_{\deg})$ for two given graphs $G$ and $H$. Then by Equation 20 of Proposition B.3 we have that $\hom(\mathcal{C}_{1,t}, G)[F] = \hom(\mathcal{C}_{1,t}, H)[F]$ for every its graph entry $F$ in $\mathcal{C}_{1,t}$.

For the other direction, suppose that $\mathsf{wr}^{(t)}(G, \chi_{\deg}) \neq \mathsf{wr}^{(t)}(H, \chi_{\deg})$ for two given $G$ and $H$. For clarity, we use the shortcut $m_{\boldsymbol{w}}^X = \mathsf{wr}^{(t)}(X, \chi_{\deg})(\boldsymbol{w})$ for graph $X$ in $\{G, H\}$. Using this notation, there exists $\boldsymbol{w}'$ in $\mathbb{N}^t$ such that $m_{\boldsymbol{w}'}^G \neq m_{\boldsymbol{w}'}^H$.

Let $n$ bound the number of vertices of $G$ and $H$, then the maximum degree of both graphs is at most $n - 1$. That means there is at most $(n-1)^t$ plain walks of length $t$ in each graph, implying both

$$m_{\boldsymbol{w}}^G \leqslant (n-1)^t, \quad \text{and} \quad m_{\boldsymbol{w}}^H \leqslant (n-1)^t.$$

Here, we use Lemma B.1, applied on $t$-tuples $\boldsymbol{w}$ in $[n-1]^t$. We choose $D_0 = n^t$ and bound entries by $D_1 = D_2 = \cdots = D_t = n$, to get an injective representation of each tuple. As a result, we obtain coefficients $(s_1, s_2, \ldots, s_t)$ such that the function

$$\left( m_{\boldsymbol{w}}^X, \boldsymbol{w}[1], \boldsymbol{w}[2], \ldots, \boldsymbol{w}[t] \right) \mapsto m_{\boldsymbol{w}}^X \cdot \boldsymbol{w}[1]^{s_1} \cdot \boldsymbol{w}[2]^{s_2} \cdots \boldsymbol{w}[t]^{s_t},$$

is injective for both $G$ and $H$ taken as $X$. Furthermore, we apply Lemma B.2 by setting vectors $\boldsymbol{x}, \boldsymbol{a}, \boldsymbol{b}$ in $[n-1]^t$ as follows:

$$\boldsymbol{x}[i] = \boldsymbol{w}_i[1]^{s_1} \cdot \boldsymbol{w}_i[2]^{s_2} \cdots \boldsymbol{w}_i[t]^{s_t}, \qquad \boldsymbol{a}[i] = m_{\boldsymbol{w}_i}^G, \qquad \boldsymbol{b}[i] = m_{\boldsymbol{w}_i}^H,$$

where $i$ is the index enumerating each $\boldsymbol{w}_i$ in $[n-1]^t$.

Since we assumed $m_{\boldsymbol{w}'}^G \neq m_{\boldsymbol{w}'}^H$, it holds $\boldsymbol{a} \neq \boldsymbol{b}$ and therefore by Lemma B.2 we can find a finite $k$ such that

$$\sum_{\boldsymbol{w} \in [n-1]^t} m_{\boldsymbol{w}}^G \cdot \left( \prod_{i=1}^{t} \boldsymbol{w}[i]^{s_i} \right)^k \neq \sum_{\boldsymbol{w} \in [n-1]^t} m_{\boldsymbol{w}}^H \cdot \left( \prod_{i=1}^{t} \boldsymbol{w}[i]^{s_i} \right)^k. \tag{21}$$

Finally, there is a caterpillar $F'$ in $\mathcal{C}_{1,t}$ determined by $\boldsymbol{s}_{F'} = (s_1 k, s_2 k, \ldots, s_t k)$, and since both sides of Eq. (21) can be rewritten by applying exponent $k$ as $\hom(\mathcal{C}_{1,t}, G)[F'] \neq \hom(\mathcal{C}_{1,t}, H)[F']$ by Proposition B.3, we obtain $\hom(\mathcal{C}_{1,t}, G) \neq \hom(\mathcal{C}_{1,t}, H)$. $\square$

### B.1 QUANTUM GRAPH HOMOMORPHISMS

In this subsection, we extend the proof of Theorem B.4 to the case of generalized caterpillars $\mathcal{C}_{h,t}$. Similarly to the previous case in Proposition B.3, where we counted possible mappings of each leg of the folklore caterpillar, we can adapt a more general approach by replacing star graphs with 1-labeled graphs. Finally, we replace the counts of all possible mappings of star graphs by the homomorphism counts of linear combinations of 1-labeled graphs, called quantum 1-labeled graphs.

**Labeled graphs.** We follow the algebraic approach to quantum graphs by Lovász and Szegedy (2009), instantiating for the 1-labeled case. A 1-*labeled graph*, or simply *labeled graph*, $G^\bullet$ is a graph $G$ with one distinguished vertex $u$ in $V(G)$, called a *label*, denoted by $\mathsf{lab}(G^\bullet)$.

Let $\mathcal{F}^\bullet$ be a class of labeled graphs, $F^\bullet$ in $\mathcal{F}^\bullet$ labeled, and $G^\bullet$ another labeled graph then we define vector $\mathsf{hom}(\mathcal{F}^\bullet, G^\bullet)$ in $\mathbb{N}^{\mathcal{F}^\bullet}$ entry-wise: each its entry $\mathsf{hom}(\mathcal{F}^\bullet, G^\bullet)[F^\bullet]$ is the number of graph homomorphisms $\varphi\colon V(G) \to V(H)$ that, moreover, preserve the label, i.e.,

$$\varphi(\mathsf{lab}(F^\bullet)) = \mathsf{lab}(G^\bullet).$$

In cases where we need to explicitly indicate the labeled vertex of $G^\bullet$, we write $G^u$ for $u = \mathsf{lab}(G^\bullet)$ in $V(G)$.

Next, a *product* $G_1^\bullet \cdot G_2^\bullet$ of two labeled graphs $G_1^\bullet$, $G_2^\bullet$ is the graph created by identification of $\mathsf{lab}(G_1^\bullet)$ and $\mathsf{lab}(G_2^\bullet)$ in the disjoint union of $G_1^\bullet$ and $G_2^\bullet$.

A *quantum graph* is a formal linear combination of finitely many graphs. A 1-*labeled quantum graph*, or simply *labeled quantum graph*, is a formal linear combination (with real coefficients) of finitely many 1-labeled graphs. The homomorphism counting extends linearly to quantum graphs:

$$\mathsf{hom}(\mathcal{F}^\bullet, G^\bullet)\Big[ \sum_{i=1}^{d} \alpha_i F_i^\bullet \Big] := \sum_{i=1}^{d} \alpha_i\, \mathsf{hom}(\mathcal{F}^\bullet, G^\bullet)[F_i^\bullet],$$

for the coefficients $\alpha_i$ in $\mathbb{R}$, and the 1-labeled graphs $F_i^\bullet$ for $i$ in $[d]$.

Moreover, quantum graphs $G_1^\bullet, G_2^\bullet$ can be naturally combined by sum, product, and exponentiation operations. A *sum* $G_1^\bullet + G_2^\bullet$ is the sum of their linear combinations. A *product* $G_1^\bullet \cdot G_2^\bullet$ is the product of their linear combinations, where we use the definition of the product of two labeled graphs. Finally, an *exponentiation* $(G_1^\bullet)^k$ for an integer $k \geqslant 1$ is the $k$-fold product of $G^\bullet$ with itself. We will use a standard identity for 1-labeled, possibly quantum, graphs:

$$\mathsf{hom}(\mathcal{F}^\bullet, G^\bullet)[(F^\bullet)^k] = (\mathsf{hom}(\mathcal{F}^\bullet, G^\bullet)[F^\bullet])^k . \tag{22}$$

The following result is due to Cai et al. (1992). Let $\mathcal{C}_{2,h}$ denote the class of formulas of two-variable first-order logic with counting quantifiers, where the quantifier depth is bounded by $h$.

**Theorem B.5** (Cai et al. (1992, Theorem 5.2)). *Let $G^u$, $H^v$ be a pair of labeled graphs and $\chi_{\mathsf{cr}}^{(h)}$ be a coloring for $h$ in $\mathbb{N}$. Then the following are equivalent:*

*(i) $\chi_{\mathsf{cr}}^{(h)}(G, u) = \chi_{\mathsf{cr}}^{(h)}(H, v)$,*

*(ii) $(G, u) \equiv_{\mathcal{C}_{2,h}} (H, v)$.*

The above result was followed by the work of Dvořák (2010) stating that the homomorphism counts of 1-labeled graphs are also equivalent to the first-order logic with counting quantifiers.

**Theorem B.6** (Dvořák (2010, Theorem 7)). *Let $G^u$, $H^v$ be a pair of labeled graphs, and let $\mathcal{T}_h^\bullet$ be the class of all 1-labeled trees of depth $h$ Then the following are equivalent*

*(i) $(G, u) \equiv_{\mathcal{C}_{2,h}} (H, v)$,*

*(ii) $\mathsf{hom}(\mathcal{T}_h^\bullet, G) = \mathsf{hom}(\mathcal{T}_h^\bullet, H)$.*

**Proposition B.7.** *Let $h, t$ in $\mathbb{N}$, and let $G$ be a graph and $\chi_{\mathsf{cr}}^{(h)}\colon V(G) \to \Sigma(G)$ a coloring. For each color $c$ in $\Sigma(G)$, choose a vertex $u$ in $V(G)$ such that $\chi_{\mathsf{cr}}^{(h)}(G, u) = c$, and denote by $G(c)$ the labeled graph $G^u$. Then, it holds:*

$$\mathsf{hom}(\mathcal{C}_{h,t}, G)[F] = \sum_{\boldsymbol{w} \text{ in } \Sigma^t} \mathsf{wr}^{(t)}(G, \chi_{\mathsf{cr}}^{(h)})(\boldsymbol{w}) \cdot \prod_{i=1}^{t} \mathsf{hom}(\mathcal{T}_h^\bullet, G(\boldsymbol{w}[i]))[T_i^\bullet], \tag{23}$$

*for every $F$ in $\mathcal{C}_{h,t}$, where $(T_1^\bullet, T_2^\bullet, \ldots T_t^\bullet)$ is a leg sequence of $F$ corresponding to the spine $(\mathsf{lab}(T_1^\bullet), \mathsf{lab}(T_2^\bullet), \ldots \mathsf{lab}(T_t^\bullet))$.*

*Proof.* Given a colored walk $\boldsymbol{w}$ in $\mathbb{N}^t$, consider its occurrence $\boldsymbol{u} = (u_1, u_2, \ldots, u_t)$ in $G$. Let $k_{\boldsymbol{u}}$ be the number of homomorphisms $\varphi \colon V(F) \to V(G)$ that map the spine of $F$ to the occurrence $\boldsymbol{u}$, that is, $\mathsf{lab}(T_i) = u_i$ for $i$ in $[t]$.

Specifically, for each $u_i$, independently, there is exactly $\mathsf{hom}(\mathcal{T}_h^\bullet, G^{u_i})[T_i^\bullet]$ ways to map the attached leg $T_i^\bullet$ into $G$:

$$\mathsf{hom}(\mathcal{C}_{h,t}, G)[F] = \sum_{\boldsymbol{u} \in V(G)^t} k_{\boldsymbol{u}} = \sum_{\substack{\boldsymbol{u} \text{ occurrence} \\ \text{of } \boldsymbol{w} \text{ in } G}} \prod_{i=1}^{t} \mathsf{hom}(\mathcal{T}_h^\bullet, G^{\boldsymbol{u}[i]})[T_i^\bullet]. \tag{24}$$

Furthermore, by the theorems Theorem B.5 and Theorem B.6, the number of tree homomorphisms $\mathsf{hom}(\mathcal{T}_h^\bullet, G^{u_i})[T_i^\bullet]$ only depends on the color of $u_i$, which is $\chi_{\mathsf{cr}}^{(h)}(G, u_i) = \boldsymbol{w}[i]$. Indeed we get, $\mathsf{hom}(\mathcal{T}_h^\bullet, G^{u_i})[T_i^\bullet] = \mathsf{hom}(\mathcal{T}_h^\bullet, G(\boldsymbol{w}[i]))[T_i^\bullet]$. Finally, reorganizing the sum in Eq. (24) to range over all possible colored walks $\boldsymbol{w}$ in $\Sigma^t$ using the known multiplicities $\mathsf{wr}^{(t)}(G, \chi_{\mathsf{cr}}^{(h)})(\boldsymbol{w})$, we obtain exactly the expression on the right-hand side of Eq. (23). $\qquad\square$

We also make use of the following result of Dvořák (2010), referred to as Lemma 6.

**Proposition B.8** (Dvořák (2010, Lemma 6))**.** *For every formula $\psi(x)$ in $\mathcal{C}_{2,h}$, there exists a quantum graph $T^\bullet$ with its base in $\mathcal{T}_h^\bullet$ such that*

$$\mathsf{hom}(T^\bullet, G^u) = \begin{cases} 1, & \text{if } (G, u) \models \psi(x), \\ 0, & \text{otherwise.} \end{cases}$$

We are prepared to restate and prove the main result of this section.

**Theorem 4.1.** *For every $h, t \geqslant 0$, it holds that $\mathsf{wr}^{(t)}(-, \chi_{\mathsf{cr}}^{(h)}) \equiv \mathsf{hom}(\mathcal{C}_{h,t}, -)$.*

*Proof.* Following the structure of the proof of Theorem B.4, we first show that $\mathsf{wr}^{(t)}(-, \chi_{\mathsf{cr}}^{(h)}) \sqsupseteq \mathsf{hom}(\mathcal{C}_{h,t}, -)$. Suppose that $\mathsf{wr}^{(t)}(G, \chi_{\mathsf{cr}}^{(h)}) = \mathsf{wr}^{(t)}(H, \chi_{\mathsf{cr}}^{(h)})$ for two given graphs $G$ and $H$. By Proposition B.7 we have that $\mathsf{hom}(\mathcal{C}_{h,t}, G)[F] = \mathsf{hom}(\mathcal{C}_{h,t}, H)[F]$ for every its graph entry $F$ in $\mathcal{C}_{h,t}$.

For the other direction, suppose that $\mathsf{wr}^{(t)}(G, \chi_{\mathsf{cr}}^{(h)}) \neq \mathsf{wr}^{(t)}(H, \chi_{\mathsf{cr}}^{(h)})$ for two given $G$ and $H$. We want to proof the existence of a caterpillar in $\mathcal{C}_{h,t}$ for which the homomorphism counts differ. We denote the common finite set of colors given by $\chi_{\mathsf{cr}}^{(h)}$ in both graphs by $\Sigma = \Sigma(G) \cup \Sigma(H)$. Given a color $c$ in $\Sigma$, we select a vertex $u$ in $V(G) \sqcup V(H)$ such that $\chi_{\mathsf{cr}}^{(h)}(G', u) = c$, where $G'$ is the disjoint union of $G$ and $H$. We denote $G'^\bullet$ such that $\mathsf{lab}(G'^\bullet) = u$ by $G'(c)$.

Additionally, for each color $c$ in $\Sigma$, there exists a formula $\psi_c$ in $\mathcal{C}_{2,h}$ such that, for every graph $G$ and vertex $u$ in $V(G)$, we have $(G, u) \models \psi_c(x)$ if and only if $\chi_{\mathsf{cr}}^{(h)}(G, u) = c$. This follows from Claim 2 in the proof of Theorem 5.5.3 given in (Grohe, 2017); see also (Cai et al., 1992; Immerman and Lander, 1990).

By Proposition B.8, it follows that for each $\psi_c(x)$ there is a quantum graph $T_c^\bullet$ of depth at most $h$ such that $\mathsf{hom}(T_c^\bullet, G^u) = 1$ if $\psi_c(x)$ holds and 0 otherwise.

We fix a linear order on $\Sigma$ by $\{c_1, c_2, \ldots, c_p\} = \Sigma$, where $p = |\Sigma|$. For each color $c_i$ there is a quantum graph $T_{c_i}^\bullet$ corresponding to $\psi_{c_i}$. We denote the quantum graph obtained by the scalar multiplication by $i$ as $L_{c_i}^\bullet = (i + 1) \cdot T_{c_i}^\bullet$, so that

$$\mathsf{hom}(\mathcal{T}_h^\bullet, G^u)[L_{c_i}^\bullet] = \begin{cases} i + 1, & \text{if } \chi_{\mathsf{cr}}^{(h)}(G, u) = c_i, \\ 0, & \text{otherwise.} \end{cases}$$

We apply Lemma B.1 to find exponents $s_1, \ldots, s_t$ for bounds $D_0 = |V(G)|^t$, and $D_1 = \cdots = D_t = p + 2$ to get such that the function

$$\big(m_{\boldsymbol{w}}^X, \boldsymbol{w}[1], \boldsymbol{w}[2], \ldots, \boldsymbol{w}[t]\big) \mapsto m_{\boldsymbol{w}}^X \cdot \prod_{i=0}^{t} \big(\mathsf{hom}(\mathcal{T}_h^\bullet, G'(\boldsymbol{w}[i]))[L_{\boldsymbol{w}[i]}^\bullet]\big)^{s_i}$$

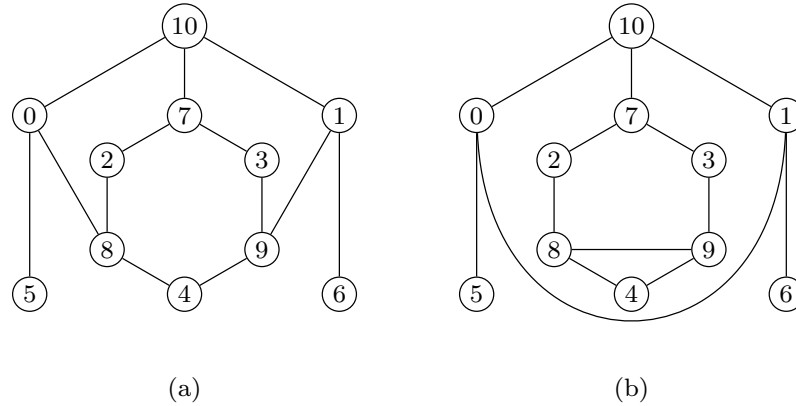

(a)                    (b)

Figure 11: An example of a pair of graphs $G$ and $H$ that strictly separates the graph functions $\mathsf{hom}(\mathcal{C}_1, -)$ and $\mathsf{hom}(\mathcal{C}_0, -)$. Note that the graph $H$ can be derived from $G$ by removing the edges $\{0, 8\}$ and $\{1, 9\}$, and then adding the edges $\{0, 1\}$ and $\{8, 9\}$.

is injective on both $X$ in $\{G, H\}$ and every $\boldsymbol{w}$ in $\Sigma^t$, where $m_{\boldsymbol{w}}^X = \mathsf{wr}^{(t)}(X, \chi_{\mathsf{cr}}^{(h)})(\boldsymbol{w})$. Next, for each $\boldsymbol{w} = w_1 w_2 \ldots w_t$ in $\Sigma^t$, we construct a quantum caterpillar graph $F_{\boldsymbol{w}}$ with a sequence of $t$ legs given by $\left((L_{w_1}^{\bullet})^{s_1}, (L_{w_2}^{\bullet})^{s_2}, \ldots, (L_{w_t}^{\bullet})^{s_t}\right)$.

Furthermore, we apply Lemma B.2 by setting $\boldsymbol{x}, \boldsymbol{a}, \boldsymbol{b}$ in $[p+2]^t$ as:

$$\boldsymbol{x}[i] = \prod_{j=0}^{t}(\mathsf{hom}(\mathcal{T}_h^{\bullet}, G'(\boldsymbol{w}_i[j]))[L_{\boldsymbol{w}_i[j]}^{\bullet}])^{s_j}, \quad \boldsymbol{a}[i] = m_{\boldsymbol{w}_i}^G, \quad \boldsymbol{b}[i] = m_{\boldsymbol{w}_i}^H,$$

where $i$ is the index enumerating each $\boldsymbol{w}_i$ in $[p+2]^t$. As a result we get sufficiently large exponent $k$ distinguishing the expressions involving $\boldsymbol{a}$ and $\boldsymbol{b}$. Specifically, we consider the quantum caterpillar given by the spine

$$F_{\boldsymbol{w}}^k = \left((L_{w_1}^{\bullet})^{k \cdot s_1}, (L_{w_2}^{\bullet})^{k \cdot s_2}, \ldots, (L_{w_t}^{\bullet})^{k \cdot s_t}\right),$$

for which we obtain the following

$$\mathsf{hom}(\mathcal{C}_{h,t}, G)[F_{\boldsymbol{w}}^k] = (\mathsf{hom}(\mathcal{C}_{h,t}, G)[F_{\boldsymbol{w}}])^k$$
$$\neq (\mathsf{hom}(\mathcal{C}_{h,t}, H)[F_{\boldsymbol{w}}])^k = \mathsf{hom}(\mathcal{C}_{h,t}, H)[F_{\boldsymbol{w}}^k],$$

where both equalities follow from Proposition B.7 and the identity in Eq. (22).

Finally, because every quantum graph is a linear combination of non-quantum ones, it follows immediately that there exists at least one non-quantum caterpillar graph $F'$ in $\mathcal{C}_{h,t}$ in the base of $F_{\boldsymbol{w}}^k$ for which

$$\mathsf{hom}(\mathcal{C}_{h,t}, G)[F'] \neq \mathsf{hom}(\mathcal{C}_{h,t}, H)[F'].$$

$\square$

**Corollary B.9.** *For every $h \geqslant 0$ it holds that* $\mathsf{wr}(-, \chi_{\mathsf{cr}}^{(h)}) \equiv \mathsf{hom}(\mathcal{C}_h, -)$.

*Proof.* It follows from Theorem 4.1 and the fact that $\mathcal{C}_h$ is a superclass of $\mathcal{C}_{h,t}$ for $t$ in $\mathbb{N}$. $\square$

### B.2 STRICT SEPARATION

For a strict separation, we define closure under disjoint unions of generalized caterpillar graphs in order to apply existing theory on minor-closed classes. Formally, let $G = (V, E)$ be a graph and let $e = uv$ in $E$ be an edge. The *contraction* of $e$ in $G$ is a graph $G/e$ with removed $e$ and identified vertices $u$ and $v$. Graph $G_m$ is a *minor* of $G$ if it can be obtained by a sequence of edge deletions, vertex deletions and edge contractions from $G$.

A class of graphs $\mathcal{C}$ is *minor-closed* if for every $G$ in $\mathcal{C}$ and every minor $G'$ of $G$, we have $G'$ in $\mathcal{C}$. We denote by $\mathcal{C}_{h,t}^{\sqcup}$ the closure of $\mathcal{C}_{h,t}$ under finite disjoint unions, that is, $\mathcal{C}_{h,t}^{\sqcup} \supseteq \mathcal{C}_{h,t}$ and if $(V_1, E_1), (V_2, E_2)$ in $\mathcal{C}_{h,t}^{\sqcup}$ then $(V_1 \sqcup V_2, E_1 \sqcup E_2)$ in $\mathcal{C}_{h,t}^{\sqcup}$. Analogicall, we denote by $\mathcal{C}_h^{\sqcup}$ the closure of $\mathcal{C}_h$ under finite disjoint unions.

**Proposition B.10.** *For all integers $h, t \geqslant 0$, the class $\mathcal{C}_{h,t}^{\sqcup}$ is minor-closed.*

*Proof.* Take a graph $G$ in $\mathcal{C}_{h,t}^{\sqcup}$ and consider either deletion or contraction of it edge $uv$. Edge $uv$ lies in one of the connected components $G'$, which in $\mathcal{C}_{h,t}$, therefore there is a sequence of legs $((L_1, s_1), \ldots (L_t, s_t))$. We now consider following cases:

1. Contraction of edge in $(L_i, s_i)$: we obtain by contraction $(L_i/uv, s_i)$ which is in $\mathcal{T}_h^{\bullet}$

2. Deletion of edge in $(L_i, s_i)$: we obtain two graphs, one of them contains $s_i$, $(L_i', s_i)$ in $\mathcal{T}_h^{\bullet}$, and for the second one containing $u$ (without loss of generality) $(L_i', u)$ in $\mathcal{T}_h^{\bullet}$.

3. Contraction of edge $s_i s_{i+1}$: we obtain only shorter spine, thus $G/s_i s_{i+1}$ in $\mathcal{C}_{h,t-1}^{\sqcup} \subseteq \mathcal{C}_{h,t}^{\sqcup}$.

4. Deletion of edge $s_i s_{i+1}$: we obtain two components, each is of them is again of a shorter spine.

5. Vertex deletions are analogous.

$\square$

The following lemma is a consequence of Roberson (2022, Lemma 5.14).

**Lemma B.11.** *It holds $\mathrm{hom}(\mathcal{C}_0^{\sqcup}, -) \subsetneqq \mathrm{hom}(\mathcal{C}_1^{\sqcup}, -)$.*

*Proof.* Follows from a stronger statement about the homomorphism distinguishability closedness of some classes closed on minors and disjoint unions. Specifically for $\mathcal{C}_0^{\sqcup}$, this was shown by Roberson (2022, Lemma 5.14 as remarked in Section 5.1). For illustration, we give a concrete separating example in Figure 11. $\square$

The following lemma is a consequence of Schindling (2025, Theorem 4.13.).

**Lemma B.12.** *It holds $\mathrm{hom}(\mathcal{C}_1^{\sqcup}, -) \subsetneqq \mathrm{hom}(\mathcal{C}_2^{\sqcup}, -)$.*

*Proof.* The class of unions of caterpillars $\mathcal{C}_1^{\sqcup}$ corresponds to the class graphs of pathwidth at most 1 (Proskurowski and Telle, 1999, Section 6). We apply a stronger statement about the homomorphism distinguishability closedness of $\mathcal{C}_1^{\sqcup}$, this was shown recently by (Schindling, 2025, Theorem 4.13.). $\square$

**Lemma B.13.** *It holds $\mathrm{hom}(\mathcal{C}_0, -) \subsetneqq \mathrm{hom}(\mathcal{C}_1, -) \subsetneqq \mathrm{hom}(\mathcal{C}_2, -)$.*

*Proof.* For a disjoint union of two graphs $F_1, F_2$ it holds $\mathrm{hom}(F_1 \sqcup F_2, G) = \mathrm{hom}(F_1, G) \cdot \mathrm{hom}(F_2, G)$, e.g. (Lovász, 2012, pg. 74). For any $h$ in $\mathbb{N}$, if $\mathrm{hom}(\mathcal{C}_h^{\sqcup}, G)[F_1 \sqcup F_2] \neq \mathrm{hom}(\mathcal{C}_h^{\sqcup}, H)[F_1 \sqcup F_2]$, then $\mathrm{hom}(\mathcal{C}_h^{\sqcup}, G)[F_1] \neq \mathrm{hom}(\mathcal{C}_h^{\sqcup}, H)[F_1]$ or $\mathrm{hom}(\mathcal{C}_h^{\sqcup}, G)[F_2] \neq \mathrm{hom}(\mathcal{C}_h^{\sqcup}, H)[F_2]$. Therefore, we get $\mathrm{hom}(\mathcal{C}_h^{\sqcup}, -) \equiv \mathrm{hom}(\mathcal{C}_h, -)$. Finally, we use Lemma B.11 and Lemma B.12. $\square$

### B.3 Expressivity Hierarchy

**Corollary B.14.** *There is the following expressivity hierarchy such that for any integer $h \geqslant 3$:*

$$
\begin{array}{ccccccccc}
\mathrm{hom}(\mathcal{P}, -) & \subsetneqq & \mathrm{hom}(\mathcal{C}_1, -) & \subsetneqq & \mathrm{hom}(\mathcal{C}_2, -) & \sqsubseteq \cdots \sqsubseteq & \mathrm{hom}(\mathcal{C}_h, -) & \sqsubseteq \cdots \sqsubseteq & \mathrm{hom}(\mathcal{T}, -) \\
\| \| \| & & \| \| \| & & \| \| \| & & \| \| \| & & \| \| \| \\
\mathcal{I}_{EA}(-, \chi_{\mathrm{triv}}) & \subsetneqq & \mathcal{I}_{EA}(-, \chi_{\mathrm{deg}}) & \subsetneqq & \mathcal{I}_{EA}(-, \chi_{\mathrm{cr}}^{(2)}) & \sqsubseteq \cdots \sqsubseteq & \mathcal{I}_{EA}(-, \chi_{\mathrm{cr}}^{(h)}) & \sqsubseteq \cdots \sqsubseteq & \mathrm{cr}(-),
\end{array}
$$

*Proof.* The second equivalence in the hierarchy is given by Theorem B.4, while the intermediate equivalences follow from Theorem 4.1 and Theorem 4.2. The strict separations follow from Lemma B.13. The last equivalence follows from Theorem B.6. □

## C  GRAPH NEURAL NETWORKS

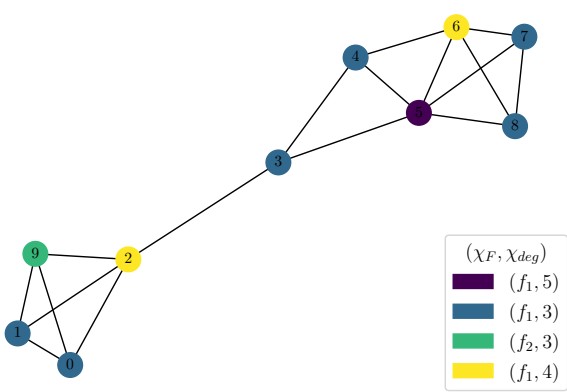

Figure 12: An example graph representation of a protein structure, colored by $(\chi_F, \chi_{\deg})$. The shown datapoint is taken from the Proteins dataset (Morris et al., 2020).

In this section, we further discuss variants of Caterpillar GNN, a practical architecture built upon the theoretical foundations of efficient aggregation. Graphs in real-world datasets such as Proteins (Morris et al., 2020), ZINC (Irwin et al., 2012), or ESOL (Wu et al., 2018) typically come attributed with *vertex features*. An example is given in Figure 12. Here, we assume that these features are seen as categorical, taken from a finite set $\Sigma$. Rather than encoding continuous-valued properties, these features represent discrete properties such an atom type or molecular class.

The vertex features of a given graph $G$, we represent as a coloring function $\chi_F(G, -)\colon V(G) \to \Sigma$. To seamlessly integrate vertex features with our scalable aggregation scheme, we introduce a parametrized combined coloring that incorporates both the vertex features and the refinement coloring:

$$\tilde{\chi}^{(h)}(G, u) = (\chi_F(G, u), \chi_{\mathrm{cr}}^{(h)}(G, u)), \tag{25}$$

for every vertex $u$ in $V(G)$ and $h$ in $\mathbb{N}$. The primary motivation for combining with $\chi_F$ is to prevent the computational graph from begin too downscaled to accommodate for distinct vertex features. Technically, in the following architecture, we employ vertex-feature matrices $\boldsymbol{Y} = \boldsymbol{Y}(G, \chi_F)$ indexed by (specifically selected) colored walks $\boldsymbol{a}c$ in $\Sigma^{t+1}$, and feature channels $i$. Crucially, each entry $Y[\boldsymbol{a}c, i]$ only needs to represent $i$-th channel of the vertex feature $c = \chi_F(G, u)$ for a suitable $u$ in $V(G)$. This is due to the structure to the prefix-successor relation we explained in Section 3.

### C.1  CATERPILLAR GCN

Let $L$ denote the number of layers in the network. For each layer $\ell$, where $0 \leqslant \ell \leqslant L$ and $t_\ell = L - \ell$, we define the following parameters: $c_\ell$ in $\mathbb{N}$, the number of feature channels; $\boldsymbol{W}^{(\ell)}$ in $\mathbb{R}^{c_\ell \times c_{\ell+1}}$, a learnable weight matrix; $\boldsymbol{Y}^{(\ell)}$, vertex feature matrix with channel embeddings for $c_\ell$; $\sigma$, an activation function, e.g. ReLU. Recall from Section 3 that $S_t = S_t(G, \tilde{\chi}) \subseteq \Sigma^t$ denotes canonical subsets of colored walks of size at most $|V(G)|$.

We derive a *Caterpillar GCN* as a sequence of layers transforming feature matrixes. Specifically, for each layer $\ell$, we the features $\boldsymbol{h}^{(\ell)}$ in $\mathbb{R}^{S_{t_\ell} \times c_\ell}$ as follows:

$$\boldsymbol{h}^{(0)} = \boldsymbol{Y}^{(0)}, \quad \boldsymbol{h}^{(\ell+1)} = \sigma\big(\boldsymbol{C}_{t_\ell}^{\tilde{\boldsymbol{A}}} \boldsymbol{h}^{(\ell)} \boldsymbol{W}^{(\ell)}\big) \square \boldsymbol{Y}^{(\ell)}, \quad \boldsymbol{h}^{(L)} = \boldsymbol{C}_0^{\boldsymbol{I}} \boldsymbol{h}^{(L)} \boldsymbol{W}^{(L)},$$

where $\square$ represents a standard addition or a concatenation. Finally, the resulting graph-level feature is

$$\boldsymbol{h}(L, \theta; G, \chi) := \boldsymbol{h}^{(L)} \text{ in } \mathbb{R}^{\{\lambda\} \times c_L}, \tag{26}$$

where $\theta$ is the set of all learnable parameters. For the $\ell$-th layer, we used the $t_\ell$-th efficient matrix (Equation 2). We specifically set $\boldsymbol{M} := \tilde{\boldsymbol{A}}$ for the efficient variant $\boldsymbol{C}_t^{\boldsymbol{M}}$, where $\tilde{\boldsymbol{A}}$ is the augmented normalized adjacency matrix, see Kipf and Welling (2017), defined as follows: $\tilde{\boldsymbol{A}} = \hat{\boldsymbol{D}}^{-1/2} \hat{\boldsymbol{A}} \hat{\boldsymbol{D}}^{-1/2}$, where $\hat{\boldsymbol{A}} = \boldsymbol{A} + 2\boldsymbol{I}$, and $\hat{\boldsymbol{D}} + 2\boldsymbol{I}$. For an example of such a computational graph, see Figure 13.

Indexing of the efficient variants of graph matrices ensures the correct structure of aggregation from longer to shorter words, and, ultimately, $\boldsymbol{C}_0^{\boldsymbol{I}}$ maps to the space of dimension $S_0 = \{\lambda\}$, as shown in Figure 14 and Figure 4. Note that if we set $\tilde{\chi} = (\chi_F, \chi_{\mathrm{id}})$ and operation $\square$ to ignore $\boldsymbol{Y}$, our architecture becomes nothing else than a network of GCNConv (Kipf and Welling, 2017) with global readout.

## C.2 Density of Computational Graphs

As defined in the previous subsection, while computational graphs of GCNs are given by matrix $\tilde{\boldsymbol{A}}$ the computational graphs of Caterpillar GCNs are given by the efficient variants $\boldsymbol{C}_t^{\tilde{\boldsymbol{A}}}$ (see Figure 4 and Figure 13).

Even though the number of nodes of the computational graph may *decrease* significantly with height, as shown in Figure 7, and further detailed in Table 7 and Table 9, this does not necessarily imply a proportional decrease of the number of edges. To study this phenomenon, we introduce two metrics to quantify the sparsity of computational graphs.

For any matrix $\boldsymbol{M}$ in $\mathbb{R}^{R \times C}$, we denote the number of its *non-zero entries* by

$$\mathrm{nnz}(\boldsymbol{M}) := |\{(r, c) \mid \boldsymbol{M}[r, c] \neq 0, \ r \text{ in } R, \ c \text{ in } C\}|.$$

Then the *density* of $\boldsymbol{M}$ is defined as the ratio of its non-zero entries to the total number of entries, i.e., $\mathrm{nnz}(\boldsymbol{M}) \cdot |R \times C|^{-1}$. Complementarily, the *sparsity* of $\boldsymbol{M}$ is defined as $1 - \mathrm{nnz}(\boldsymbol{M}) \cdot |R \times C|^{-1}$.

**Absolute density.** We extend density to the computational graph of Caterpillar GCN by taking the average of layer densities. Let $G$ be an input graph with a $\Sigma$-coloring $\chi$, and let $L$ denote the number of layers.

$$\text{absolute\_density}(L, G, \chi) := \frac{1}{L} \sum_{t=1}^{L} \frac{\mathrm{nnz}\left(\boldsymbol{C}_t^{\tilde{\boldsymbol{A}}}\right)}{|S_t \times S_{t+1}|}. \tag{27}$$

**Relative density.** By relative density $\delta_{h,L}$, we measure the ratio of the number of edges in the computational graph of Caterpillar GCN to that of (standard) GCN.

$$\text{relative\_density}(L, G, \chi) := \frac{1}{L} \sum_{t=1}^{L} \frac{\mathrm{nnz}\left(\boldsymbol{C}_t^{\tilde{\boldsymbol{A}}}\right)}{\mathrm{nnz}\left(\boldsymbol{A}\right)}, \quad \text{and}$$

$$\delta_{h,L} := \text{relative\_density}\left(L, G, (\chi_F, \chi_{\mathrm{cr}}^{(h)})\right). \tag{28}$$

Note that for the computational graph of GCN, the relative density equals one. By Theorem 3.3, we can state this formally as $\text{relative\_density}(L, G, \chi_{\mathrm{id}}) = 1.0$. Moreover, since $\mathrm{nnz}(\boldsymbol{C}_t^{\tilde{\boldsymbol{A}}}) = \mathrm{nnz}(\boldsymbol{C}_t^{\hat{\boldsymbol{A}}}) = \mathrm{nnz}(\boldsymbol{C}_t^{\boldsymbol{A}}) + \mathrm{nnz}(\boldsymbol{C}_t^{\boldsymbol{I}})$, these metrics extend naturally to computational graphs of other variants of GNNs.

**Densities and dimensions of $\boldsymbol{C}_t^{\tilde{\boldsymbol{A}}}$ on real-world datasets.** To better understand the sparsity (density) of computational graphs induced by our model, we empirically evaluate how the density evolves with increasing height parameter $h$ for fixed $L = 5$. Specifically, for a fixed sequence length $L = 5$ and varying height parameter $h$, we measure the densities (defined above) of the resulting computational graphs across TUDataset, MoleculeNet, and ZINC, as summarised in Table 3.

In our experiments, we observe that the relative density $\delta_{h,5}$ is, in most cases, strictly below 1.0, and can overall be bounded by a small constant, for instance $\delta_{h,5} \lessgtr 2.0$. This implies that the

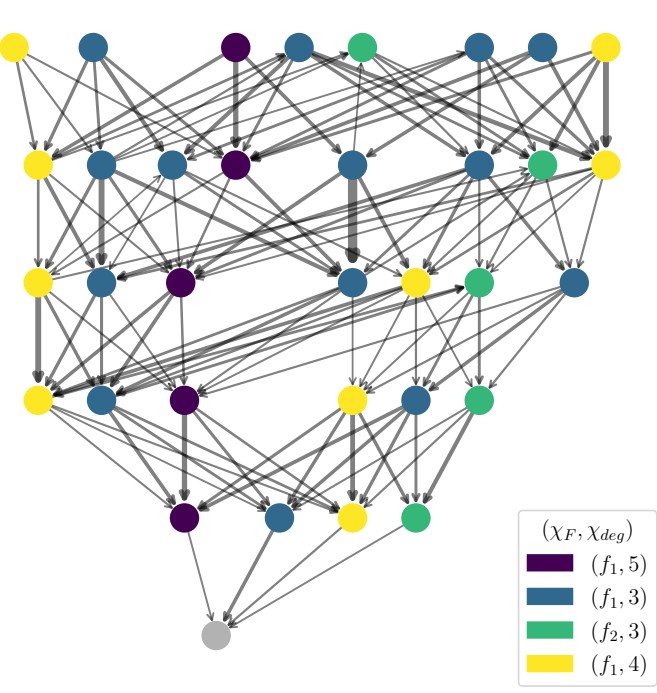

Figure 13: Computational graph (**ours**) of Caterpillar GCN with $h = 1$ that uses $\boldsymbol{C}_{t_\ell}^{\tilde{\boldsymbol{A}}}$ matrix at $\ell$-th layer. This diagram is analogical to Figure 4. Weights are represented by line width, and signs by arrow direction. The input graph is shown in Figure 12.

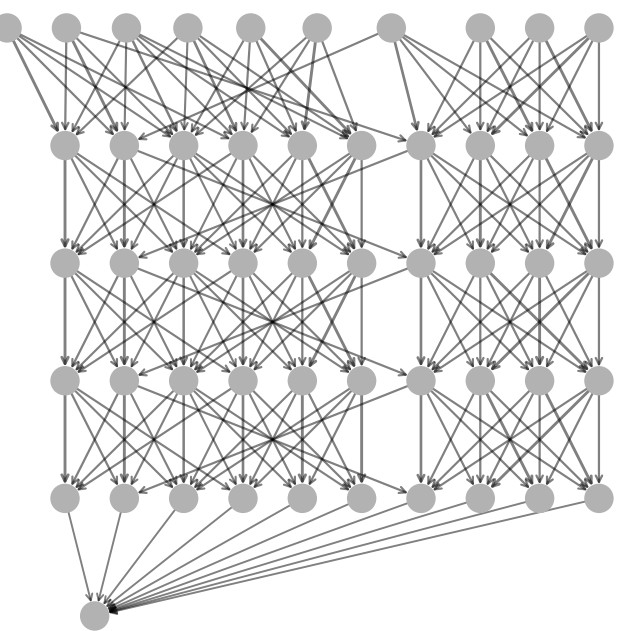

Figure 14: Computational graph (message-passing) of GCN that uses copies of matrix $\tilde{\boldsymbol{A}}$ for its layers. The input graph is shown in Figure 12.

| type | MUTAG | | PROTEINS | | ENZYMES | | IMDB-BINARY | | COLLAB | |
|---|---|---|---|---|---|---|---|---|---|---|
| | $\delta_{h,5}$ [↓] | absolute [↑, %] | $\delta_{h,5}$ [↓] | absolute [↑, %] | $\delta_{h,5}$ [↓] | absolute [↑, %] | $\delta_{h,5}$ [↓] | absolute [↑, %] | $\delta_{h,5}$ [↓] | absolute [↑, %] |
| $C_0$ | 0.73 ± 0.63 | 44.4 ± 10.5 | 1.26 ± 1.80 | 53.9 ± 29.4 | 1.46 ± 1.68 | 48.4 ± 26.1 | 0.01 ± 0.00 | 100.0 ± 0.0 | 0.35 ± 0.39 | 51.4 ± 24.7 |
| $C_1$ | 1.23 ± 0.82 | 36.8 ± 10.9 | 1.59 ± 1.69 | 38.2 ± 22.6 | 1.40 ± 0.77 | 31.4 ± 15.0 | 0.11 ± 0.10 | 73.4 ± 16.3 | 0.35 ± 0.39 | 51.4 ± 24.7 |
| $C_2$ | 0.94 ± 0.33 | 27.5 ± 6.9 | 0.81 ± 0.24 | 27.5 ± 23.9 | 0.84 ± 0.20 | 20.8 ± 13.0 | 0.10 ± 0.09 | 72.0 ± 17.7 | 0.30 ± 0.29 | 50.8 ± 26.2 |
| $C_3$ | 0.85 ± 0.22 | 24.4 ± 7.0 | 0.78 ± 0.21 | 27.0 ± 24.1 | 0.82 ± 0.17 | 20.4 ± 13.1 | 0.10 ± 0.09 | 72.0 ± 17.7 | 0.30 ± 0.29 | 50.9 ± 26.2 |
| $C_4$ | 0.83 ± 0.20 | 23.6 ± 7.3 | 0.78 ± 0.21 | 26.9 ± 24.1 | 0.82 ± 0.17 | 20.4 ± 13.1 | 0.10 ± 0.09 | 72.0 ± 17.7 | 0.30 ± 0.29 | 50.9 ± 26.2 |
| $C_5$ | 0.83 ± 0.19 | 23.4 ± 7.4 | 0.78 ± 0.21 | 26.9 ± 24.1 | 0.82 ± 0.16 | 20.4 ± 13.1 | 0.10 ± 0.09 | 72.0 ± 17.7 | 0.30 ± 0.29 | 50.9 ± 26.2 |
| $C_6$ | 0.83 ± 0.19 | 23.4 ± 7.4 | 0.78 ± 0.21 | 26.9 ± 24.1 | 0.82 ± 0.16 | 20.4 ± 13.1 | 0.10 ± 0.09 | 72.0 ± 17.7 | 0.30 ± 0.29 | 50.9 ± 26.2 |
| $C_7$ | 0.83 ± 0.19 | 23.4 ± 7.4 | 0.78 ± 0.21 | 26.9 ± 24.1 | 0.82 ± 0.16 | 20.4 ± 13.1 | 0.10 ± 0.09 | 72.0 ± 17.7 | 0.30 ± 0.29 | 50.9 ± 26.2 |
| $C_8$ | 0.83 ± 0.19 | 23.4 ± 7.4 | 0.78 ± 0.21 | 26.9 ± 24.1 | 0.82 ± 0.16 | 20.4 ± 13.1 | 0.10 ± 0.09 | 72.0 ± 17.7 | 0.30 ± 0.29 | 50.9 ± 26.2 |
| $C_9$ | 0.83 ± 0.19 | 23.4 ± 7.4 | 0.78 ± 0.21 | 26.9 ± 24.1 | 0.83 ± 0.16 | 20.4 ± 13.1 | 0.10 ± 0.09 | 72.0 ± 17.7 | 0.30 ± 0.29 | 50.9 ± 26.2 |
| $C_{10}$ | 0.83 ± 0.19 | 23.4 ± 7.4 | 0.78 ± 0.21 | 26.9 ± 24.1 | 0.83 ± 0.16 | 20.4 ± 13.1 | 0.10 ± 0.09 | 72.0 ± 17.7 | 0.30 ± 0.29 | 50.9 ± 26.2 |
| MP | 1.00 ± 0.00 | 18.9 ± 4.7 | 1.00 ± 0.00 | 24.4 ± 20.6 | 1.00 ± 0.00 | 19.0 ± 11.8 | 1.00 ± 0.00 | 54.7 ± 23.4 | 1.00 ± 0.00 | 51.8 ± 29.9 |

| type | FreeSolv | | Lipo | | ESOL | | BACE | | ZINC | |
|---|---|---|---|---|---|---|---|---|---|---|
| | $\delta_{h,5}$ [↓] | absolute [↑, %] | $\delta_{h,5}$ [↓] | absolute [↑, %] | $\delta_{h,5}$ [↓] | absolute [↑, %] | $\delta_{h,5}$ [↓] | absolute [↑, %] | $\delta_{h,5}$ [↓] | absolute [↑, %] |
| $C_0$ | 0.80 ± 0.34 | 51.0 ± 20.5 | 1.13 ± 0.39 | 18.9 ± 6.9 | 0.88 ± 0.44 | 40.3 ± 19.8 | 1.09 ± 0.32 | 14.1 ± 4.5 | 1.33 ± 0.50 | 23.0 ± 6.1 |
| $C_1$ | 0.80 ± 0.34 | 51.0 ± 20.5 | 1.13 ± 0.39 | 18.9 ± 6.9 | 0.88 ± 0.44 | 40.3 ± 19.8 | 1.09 ± 0.32 | 14.1 ± 4.5 | 1.29 ± 0.47 | 22.4 ± 5.9 |
| $C_2$ | 0.77 ± 0.27 | 49.4 ± 21.1 | 0.93 ± 0.17 | 15.4 ± 5.4 | 0.81 ± 0.28 | 37.6 ± 20.0 | 0.97 ± 0.15 | 11.9 ± 3.3 | 0.99 ± 0.17 | 16.9 ± 3.6 |
| $C_3$ | 0.75 ± 0.25 | 48.9 ± 21.5 | 0.88 ± 0.12 | 14.5 ± 5.2 | 0.78 ± 0.25 | 36.6 ± 20.4 | 0.90 ± 0.08 | 11.0 ± 3.0 | 0.91 ± 0.10 | 15.5 ± 3.3 |
| $C_4$ | 0.75 ± 0.25 | 48.9 ± 21.6 | 0.87 ± 0.11 | 14.3 ± 5.1 | 0.78 ± 0.24 | 36.5 ± 20.5 | 0.89 ± 0.07 | 10.9 ± 3.0 | 0.90 ± 0.09 | 15.3 ± 3.3 |
| $C_5$ | 0.75 ± 0.25 | 48.9 ± 21.6 | 0.87 ± 0.11 | 14.2 ± 5.1 | 0.78 ± 0.24 | 36.5 ± 20.5 | 0.89 ± 0.07 | 10.9 ± 3.0 | 0.90 ± 0.09 | 15.3 ± 3.3 |
| $C_6$ | 0.75 ± 0.25 | 48.9 ± 21.6 | 0.87 ± 0.11 | 14.2 ± 5.1 | 0.78 ± 0.24 | 36.5 ± 20.5 | 0.89 ± 0.07 | 10.9 ± 3.0 | 0.90 ± 0.09 | 15.2 ± 3.3 |
| $C_7$ | 0.75 ± 0.25 | 48.9 ± 21.6 | 0.87 ± 0.11 | 14.2 ± 5.1 | 0.78 ± 0.24 | 36.5 ± 20.5 | 0.89 ± 0.07 | 10.8 ± 3.0 | 0.90 ± 0.09 | 15.2 ± 3.3 |
| $C_8$ | 0.75 ± 0.25 | 48.9 ± 21.6 | 0.87 ± 0.11 | 14.2 ± 5.1 | 0.78 ± 0.24 | 36.5 ± 20.5 | 0.89 ± 0.07 | 10.9 ± 3.0 | 0.90 ± 0.09 | 15.2 ± 3.3 |
| $C_9$ | 0.75 ± 0.25 | 48.9 ± 21.6 | 0.87 ± 0.11 | 14.2 ± 5.1 | 0.78 ± 0.24 | 36.4 ± 20.5 | 0.89 ± 0.07 | 10.9 ± 3.0 | 0.90 ± 0.09 | 15.2 ± 3.3 |
| $C_{10}$ | 0.75 ± 0.25 | 48.9 ± 21.6 | 0.87 ± 0.11 | 14.2 ± 5.1 | 0.78 ± 0.24 | 36.4 ± 20.5 | 0.89 ± 0.07 | 10.9 ± 3.0 | 0.90 ± 0.09 | 15.2 ± 3.3 |
| MP | 1.00 ± 0.00 | 39.0 ± 16.3 | 1.00 ± 0.00 | 12.8 ± 4.3 | 1.00 ± 0.00 | 28.9 ± 14.9 | 1.00 ± 0.00 | 9.9 ± 2.8 | 1.00 ± 0.00 | 14.1 ± 2.9 |

Table 3: *Density of computational graphs across datasets.* By $C_h$ is denoted the (caterpillar height) parameter $h$ in $\mathbb{N}$ of efficient aggregation (ours), while MP denotes the full message-passing GCN. The subcolumns correspond to $\delta_{h,L}$ (lower is better) and relative_density (higher is better once $\delta_{h,L}$ is bounded) for $L = 5$ layers. We report absolute density in %.

matrices $C_t^{\tilde{A}}$ *typically contain fewer non-zero entries* than the original adjacency matrix $\tilde{A}$. On unattributed graphs such as IMDB-BINARY and COLLAB, the ratio $\delta_{h,5}$ tends to be even lower (e.g., approximately $0.1$ and $0.3$, respectively).

On the other hand, the absolute_density tends to increase as the height parameter $h$ decreases. This indicates that the computational graphs become significantly *denser*, which is beneficial for execution on hardware accelerators, provided that the relative density $\delta_{h,L}$ does not increase by more than a small constant factor. On typically sparse datasets such as PROTEINS and ENZYMES, we observed a substantial rise in absolute density (from approximately $20\%$ for MP to around $50\%$ for $C_0$), suggesting that such computational graphs may be more desirably processed using dense tensor operations rather than sparse ones.

## C.3   IMPACTS OF RELATIVE DENSITY ON ASYMPTOTIC COMPLEXITY

In this subsection, we discuss the asymptotic time complexity of Caterpillar GNN (including pre-computation) in comparison to some other methods learning on sequential patterns in graphs. We first focus on relative density $\delta_{h,L}$, and later, we also explain impact of the number of nodes in the computational graph.

When the number of channels $d$ in the GNN is taken as a constant, we give short overview Appendix C.3. Assuming that an input graph $G$ on $n$ vertices is connected and $n \geqslant 3$, it holds for the maximal degree $\Delta$ that $2 \leqslant \Delta \leqslant n - 1$. A multiplicative factor in our comparison $\delta_{h,L}$ is the average relative density, as defined in Appendix C.2. For Caterpillar GNNs, it is fixed that $L = T$. The best known theoretical bound to our knowledge is the trivial one: $\delta_{h,L} \leqslant n^2/|E|$. Importantly, on TUDataset, MoleculeNet and ZINC, we see that $\delta_{h,5}$ is typically a fraction smaller than $1.0$ and overall $\delta_{h,5} \lessgtr 2.0$, see Table 3.

**Complexity under large width.**    In the regime where the number of channels $d$ (i.e., the width) of our GNN is large, the multiplication of learnable parameters $W^{(\ell)}$ in $\mathbb{R}^{d \times d}$ with features $h^{(\ell)}$ in $\mathbb{R}^{|S_{t_\ell}| \times d}$ (see Appendix C.1) becomes a dominant contributor to both time and space complexity. Concretely, the cost of one layer is

$$\mathcal{O}\big(\mathrm{nnz}(C_{t_\ell}^A) \cdot d + |S_{t_{\ell+1}}| \cdot d^2\big), \tag{29}$$

| Method | Pre-comp. | Time per layer | Worst case | Equivariant |
|---|---|---|---|---|
| MPGNN (Gilmer et al., 2017) | — | $\lvert E \rvert$ | $n^2$ | ✓ |
| CRaWL (Tönshoff et al., 2023) | — | $n \cdot f_G(\Delta^T) \cdot T$ | $n f_G((n-1)^T) T$ | X |
| NeuralWalker (Chen et al., 2024) | — | $n \cdot f_G(\Delta^T) \cdot T$ | $n f_G((n-1)^T) T$ | X |
| Path NN (Michel et al., 2023) | $n \cdot \Delta^T$ | $n \cdot \Delta^T \cdot T$ | $n^{T+1} T$ | ✓ |
| Path-based GNN (Graziani et al., 2024) | $n \cdot \Delta^T$ | $n \cdot \Delta^T \cdot T$ | $n^{T+1} T$ | ✓ |
| Caterpillar GNN (ours) | $T \cdot n^{2.372}$ | $\lvert E \rvert \cdot \delta_{h,L}$ | $n^2$ | ✓ |

Table 4: Asymptotic comparison of methods processing sequential patterns of length $T$. The function $f_G$ gives the number of sampled walks on $G = (V, E)$ from the given upper bound. Symbol $\Delta$ stands for the maximal degree and $\delta_{h,L}$ is the relative density on $G$ (see Equation 28).

hence for a network with $L$ layers ($t_\ell = L - t$) and sufficiently large $d$, the term $d^2 \cdot \sum_{t=0}^{L} \lvert S_t \rvert$ dominates the overall complexity. Hence, the *number of nodes in the computational graph*, $\sum_{t=0}^{L} \lvert S_t \rvert$, appears as a multiplicative factor in the time and memory complexity and is asymptotically significant with respect to the width $d$.

Empirically, on real-world datasets TUDataset, MoleculeNet, and ZINC, we observe that the number of nodes in the computational graph tends to *decrease* as the height parameter $h$ decreases, as illustrated in Figure 7 and detailed further in Tables 7 and 9.

### C.4 TOWARDS MODERN ARCHITECTURES AND LARGE GRAPH-LEVEL DATASETS

To broaden our empirical evaluation, this subsection extends our analysis of computational graphs to more practical settings (Table 5). In the previous analysis, we fixed as many hyperparameters as possible, such as the number of layers. Here, in contrast, for structural comparison against a strong message-passing baseline, we adopt the experimental settings of the recent GCN-based architecture (GCN$^+$) proposed by Luo et al. (2025). Their work demonstrates that, with careful design and tuning, message-passing backbones can be highly competitive on graph-level prediction tasks.

Following their setup, we preserve the configuration parameters such as the number of layers, and the dimensionality of the preprocessed node features, and we reuse their protocol to ensure a controlled structural comparison where only the aggregation mechanism differs. To further isolate the impact of lower expressivity, we intentionally disable positional encodings and edge features for all architectures, consistently with the GCN$^+$ ablation setting used in Table 6 of Section 5.2 in (Luo et al., 2025). Hence, we obtain a clear assessment of how efficient aggregation contributes to graph-level performance uninfluenced by other structural augmentations.

We analyze the computational graphs of the MalNet-Tiny dataset (Freitas et al., 2021), which contain graphs with up to $5,000$ nodes ($1,410$ on average). In addition, we also include an analysis of datasets with discrete node features: Peptides-func, Peptides-func from Open Graph Benchmark Hu et al. (2021), and datasets ogbg-code2, ogbg-molhiv, ogbg-molpcba from Long-Range Graph Benchmark (Dwivedi et al., 2022), see Table 5. These datasets allow us to examine how the structure of proposed inductive biases behaves under substantially larger graph structures and more diverse data modalities. We report the preprocessing times of a *serial* precomputation on the platform of an Intel Xeon Platinum 8168 @ 2.70 GHz processor (24 cores, 48 threads, 33 MB L3 cache, 64B cache line).

**Receipt for equivariant density reduction.** Our proposed efficient variants of graph matrices are fully (permutation) equivariant. In practice, we can therefore further reduce the value of relative density $\delta_{h,L}$ (Equation 28) by, e.g., thresholding small-magnitude entries of our matrices. This optional step decreases computational cost while preserving the symmetries inherent to our construction. On the other hand, during our measurements in Table 5, similarly to our previous analysis, we observe that the relative density $\delta_{h,L}$ of the computational graphs is a very small constant. The only exception we found was in the case of $\delta_{h,8}$ for the MalNet-Tiny and the height $h = 0$, for which we took special attention to eliminate possible hardware errors.

| | MalNet-Tiny | | | | | Peptides-func | | | | | Peptides-struct | | | | |
|---|---|---|---|---|---|---|---|---|---|---|---|---|---|---|---|
| | features: 5 graphs: 5,000 avg. \|V\|: 1410.3 avg. \|E\|: 2859.9 layers: $L = 8$ (GCN$^+$) | | | | | features: 9 graphs: 15,535 avg. \|V\|: 150.9 avg. \|E\|: 307.3 layers: $L = 3$ (GCN$^+$) | | | | | features: 9 graphs: 15,535 avg. \|V\|: 150.9 avg. \|E\|: 307.3 layers: $L = 5$ (GCN$^+$) | | | | |
| | edge density | | nodal efficiency | | precomp. | edge density | | nodal efficiency | | precomp. | edge density | | nodal efficiency | | precomp. |
| type | $\delta_{h,8}$[↓] | abs.[↑, %] | #n.[↓] | %s.[↑] | time[↓] | $\delta_{h,3}$[↓] | abs.[↑, %] | #n.[↓] | %s.[↑] | time[↓] | $\delta_{h,5}$[↓] | abs.[↑, %] | #n.[↓] | %s.[↑] | time[↓] |
| $C_0$ | 30.52 | 24.7 | 5971.2 | 53.0 | 6.97h | 1.634 | 15.4 | 233.7 | 61.4 | 8.2min | 2.116 | 12.4 | 471.5 | 48.0 | 13.1min |
| $C_1$ | 2.002 | 7.7 | 6757.5 | 46.8 | 7.85h | 1.634 | 15.4 | 233.7 | 61.4 | 9.9min | 2.116 | 12.4 | 471.5 | 48.0 | 15.0min |
| $C_2$ | 1.500 | 6.4 | 7436.9 | 41.4 | 9.63h | 1.418 | 8.7 | 292.3 | 51.7 | 17.0min | 1.708 | 7.4 | 542.2 | 40.2 | 22.0min |
| $C_3$ | 1.516 | 6.3 | 7900.7 | 37.8 | 10.39h | 1.416 | 5.9 | 366.1 | 39.5 | 24.4min | 1.533 | 5.3 | 629.0 | 30.6 | 34.3min |
| $C_4$ | 1.519 | 6.3 | 7953.9 | 37.3 | 11.32h | 1.344 | 4.5 | 429.3 | 29.0 | 36.5min | 1.412 | 4.2 | 698.9 | 22.9 | 45.4min |
| $C_5$ | 1.519 | 6.3 | 7959.8 | 37.3 | 10.93h | 1.350 | 3.9 | 478.3 | 20.9 | 50.9min | 1.393 | 3.8 | 754.0 | 16.8 | 58.4min |
| $C_6$ | 1.519 | 6.3 | 7960.7 | 37.3 | 10.94h | 1.372 | 3.6 | 515.9 | 14.7 | 54.4min | 1.398 | 3.5 | 797.1 | 12.1 | 1.15h |
| MP | 1.000 | 0.8 | 12693.8 | 0.0 | – | 1.000 | 2.1 | 604.8 | 0.0 | – | 1.000 | 2.1 | 906.6 | 0.0 | – |

| | ogbg-code2 | | | | | ogbg-molhiv | | | | | ogbg-molpcba | | | | |
|---|---|---|---|---|---|---|---|---|---|---|---|---|---|---|---|
| | features: 2 graphs: 452,741 avg. \|V\|: 121.6 avg. \|E\|: 333.9 layers: $L = 4$ (GCN$^+$) | | | | | features: 9 graphs: 41,127 avg. \|V\|: 25.5 avg. \|E\|: 54.9 layers: $L = 4$ (GCN$^+$) | | | | | features: 9 graphs: 437,929 avg. \|V\|: 26.0 avg. \|E\|: 56.2 layers: $L = 10$ (GCN$^+$) | | | | |
| | edge density | | nodal efficiency | | precomp. | edge density | | nodal efficiency | | precomp. | edge density | | nodal efficiency | | precomp. |
| type | $\delta_{h,4}$[↓] | abs.[↑, %] | #n.[↓] | %s.[↑] | time[↓] | $\delta_{h,4}$[↓] | abs.[↑, %] | #n.[↓] | %s.[↑] | time[↓] | $\delta_{h,10}$[↓] | abs.[↑, %] | #n.[↓] | %s.[↑] | time[↓] |
| $C_0$ | 1.638 | 9.4 | 445.2 | 26.9 | 10.21h | 1.467 | 28.7 | 84.2 | 34.5 | 10.7min | 1.758 | 22.0 | 235.3 | 17.9 | 4.46h |
| $C_1$ | 1.476 | 8.3 | 454.8 | 25.3 | 12.09h | 1.467 | 28.7 | 84.2 | 34.5 | 11.4min | 1.758 | 22.0 | 235.3 | 17.9 | 4.57h |
| $C_2$ | 1.318 | 6.9 | 486.6 | 20.1 | 16.14h | 1.223 | 21.1 | 92.6 | 27.9 | 16.1min | 1.365 | 16.2 | 245.0 | 14.5 | 6.32h |
| $C_3$ | 1.274 | 6.1 | 526.0 | 13.6 | 22.11h | 1.188 | 18.7 | 98.9 | 23.1 | 20.4min | 1.291 | 14.7 | 251.9 | 12.1 | 7.97h |
| $C_4$ | 1.293 | 5.7 | 571.7 | 6.1 | 56.26h | 1.190 | 18.1 | 101.7 | 20.9 | 22.5min | 1.283 | 14.4 | 254.5 | 11.2 | 8.78h |
| $C_5$ | 1.299 | 5.6 | 589.0 | 3.3 | 61.11h | 1.192 | 18.0 | 102.8 | 20.0 | 24.3min | 1.282 | 14.4 | 255.3 | 10.9 | 9.10h |
| $C_6$ | 1.300 | 5.6 | 595.5 | 2.2 | 64.88h | 1.194 | 17.9 | 103.4 | 19.6 | 49.9min | 1.282 | 14.4 | 255.6 | 10.9 | 8.66h |
| MP | 1.000 | 4.3 | 608.9 | 0.0 | – | 1.000 | 9.8 | 128.6 | 0.0 | – | 1.000 | 8.8 | 286.7 | 0.0 | – |

Table 5: *Density and nodal efficiency of computational graphs across large datasets.* By $C_h$ is denoted the (caterpillar height) parameter $h$ in $\mathbb{N}$ of efficient aggregation (ours), while MP denotes the full message-passing GCN$^+$ (Luo et al., 2025). The subcolumns correspond to $\delta_{h,L}$ (lower is better) and relative_density (higher is better once $\delta_{h,L}$ is bounded) for $L = 5$ layers. We report absolute density in %. Columns "#n." denote number of *nodes* of the computation graph, and columns "%s." percent of nodes of the computation graph *saved* comparing to message-passing (MP).

# D ADDITIONAL DETAILS ON EXPERIMENTS

## D.1 INCIDENCE TOPOLOGY

Although our discussion of topology mainly serves heuristic and interpretative purposes, we provide formal definitions of the relevant notions in Section 5.1.

*Topology.* Let $V$ be a set and $\tau \subseteq 2^V$ be a family of subsets of $V$. Then $\tau$ is a *topology* on $V$ if the following three conditions hold:

(T1) Set $V$ in $\tau$, and empty set $\varnothing$ in $\tau$.

(T2) The family $\tau$ is closed on all unions of its sets.

(T3) The family $\tau$ is closed on finite intersections of its sets.

A subfamily $\mathcal{B} \subseteq \tau$ is a *subbase* for $\tau$ if $\tau$ is the intersection of all topologies on $V$ containing $\mathcal{B}$. We say that $\mathcal{B}$ *generates* $\tau$.

For a graph $G = (V, E)$ on $V$, the neighborhood topology $\tau(E)$ is generated by the following family of sets for every $v$ in $V$ and $r$ in $\mathbb{N}$ such that $r > 0$:

$$B_r(v) = \{u \mid u \text{ in } V, (\boldsymbol{I} + \boldsymbol{A})^r[u, v] \neq 0\},$$

where $\boldsymbol{A}$ is the adjacency and $\boldsymbol{I}$ is the (self-loop) identity matrix of shape $V \times V$ for $G$.

Let $\chi$ be a $\Sigma$-coloring on $G$ and let $T$ be the maximum length of colored walks. Then we denote the following family of sets for every $\boldsymbol{a}$ in $\Sigma^{\leqslant T}$

$$B(\boldsymbol{a}, T) := \{u \mid u \text{ in } V, (\boldsymbol{I} + \boldsymbol{A})\boldsymbol{W}[u, \boldsymbol{a}] \neq 0\}. \tag{30}$$

where we recall the walk-incidence matrix $\boldsymbol{W}$ of shape $V \times \Sigma^*$ defined in Section 3.1.

We call the topology generated by the subbase $\mathcal{B} = \{B(\boldsymbol{a}, T) \mid \boldsymbol{a} \text{ in } \Sigma^{\leqslant T}\}$ the *incidence topology* $\tau(\chi)$ on $V$. We state the following proposition to highlight the similarity with Observation 3.2 and Proposition 3.3.

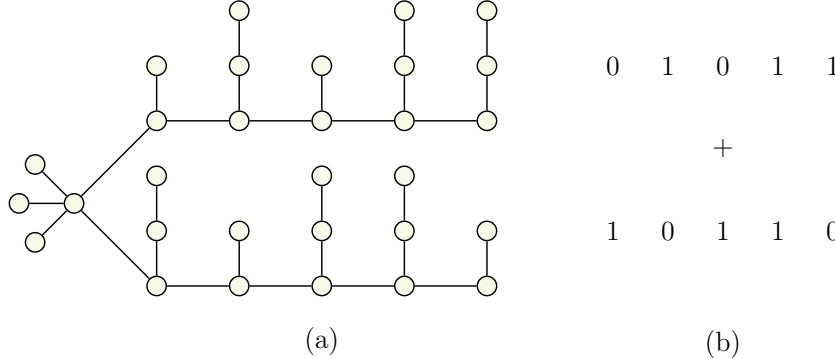

Figure 15: NSTEPSUM: An example of a graph $G$ (a). The graph $G$ encodes two 5-bit numbers (b), namely 11 (top) and 22 (bottom). If $N = 33$ for a classification dataset then $G$ is a positive example, as $11 + 22 = 33$.

**Proposition D.1.** *Let $G = (V, E)$ be a graph $G$ without isolated vertices, and $T > 0$ Then the topology $\tau(\chi_{\mathrm{triv}})$ is trivial, and the topology $\tau(\chi_{\mathrm{id}}) = \tau(E)$.*

*Proof.* Let us denote the colored walk of a single color of length $t$ by $\boldsymbol{z}_t$ (as in Observation 3.2). Since every node $u$ in $V$ has at least one neighbor, it is adjacent to colored walk $\boldsymbol{z}_{2t}$ and incident to colored walk $\boldsymbol{z}_{2t+1}$. That is, $(\boldsymbol{AW})[u, \boldsymbol{z}_{2t}] \neq 0$ and $(\boldsymbol{IW})[u, \boldsymbol{z}_{2t+1}] \neq 0$ and thus $\tau(\chi_{\mathrm{triv}})$ contains only. Therefore, the subbase is of $B(\boldsymbol{z}_t, T) = V$, and thus $\tau(\chi_{\mathrm{triv}}) = \{\varnothing, V\}$.

For the second part, we recall that $\chi_{\mathrm{id}}$ is a $V$-coloring. For $v$ in $V$ we take any colored walk $\boldsymbol{a}v$, and we get $(\boldsymbol{I} + \boldsymbol{A})W[u, \boldsymbol{a}v] = (\boldsymbol{I} + \boldsymbol{A})[u, v]$. Finally, every ball $B_r(v)$ in $\tau(E)$ is (already) generated by the union of some 1-balls of the form $B_1(v) = B(\boldsymbol{a}v, T) = \{u \mid u \text{ in } V, (\boldsymbol{I} + \boldsymbol{A})[u, v] \neq 0\}$. $\qquad\square$

## D.2 GENERATION OF NSTEPSUM DATASET

We construct the dataset for a binary classification task, where each graph encodes two integers represented in binary. For an example of a graph $G$ encoding two integers 11 and 22 in the 5-bit binary case, see Figure 15. To construct a graph of our dataset of a target sum $N = 2^{B-1}$, we randomly generate two integers $x_1$ and $x_2$ in the range $[0, 2^B - 1]$, such that $x_1 + x_2 = N$. With 0.5 probability, we add offset to one of the numbers in the range of $[0, \frac{2}{3}N]$ to deviate from the target sum $N$ by not more than 66.7%. Accordingly, we assign the class of the graph to be 1 if $x_1 + x_2 = N$, and 0 otherwise. We do not use node any features, so the classification relies solely on graph structure. Samples: 6,000 graphs with integers generated using $B = 15$ bits. The dataset we provide is balanced with respect to the class labels.

## D.3 IMPLEMENTATION DETAILS

We use the PyTorch Geometric library (Paszke et al., 2019) to implement our models. As it might be considered common, we precomputed the normalization of the (sparse) adjacency matrix for the GCNConv layer, to later use it in the forward pass with the option norm=False, and adding the normalized weights to the message-passing instead. For the cases where Caterpillar GCN is not identical to message passing, we added an efficient (sparse) version of the adjacency matrix $C_{t_\ell}^{\bar{A}}$, specialized for every layer $\ell$, in a way similar to the normalization of the adjacency matrix. We preprocessed datasets in a unified fashion for message-passing and Caterpillar GCN ($\mathsf{C}_0$ up to $\mathsf{C}_{10}$).

We trained our models on the NSTEPSUM dataset described above using deeper architectures (18 layers), with small hidden dimensions (width=8), batch size of 64, and moderate regularization settings (final dropout 0.3, weight decay $10^{-6}$). Training was performed for 10 Figure 6, and 50 epochs using 10-fold cross-validation. Complete results are reported in Table 6. We deliberately adopted a minimal configuration in order to cleanly isolate the topological effect of computational graph scaling behind our approach. We remark that the definition of incidence topology in general supports richer benchmarks. The ongoing work may extend these experiments to include multiple

| type | epochs $= 10$ | | | epochs $= 50$ | | |
|---|---|---|---|---|---|---|
| | mean $\pm$ std | #n. | %s. | mean $\pm$ std | #n. | %s. |
| $\mathcal{C}_0$ | $0.590 \pm 0.068$ | 18.0 | 98.7 | $0.601 \pm 0.124$ | 18.0 | 98.7 |
| $\mathcal{C}_1$ | $\mathbf{0.929 \pm 0.013}$ | 1011.7 | 25.1 | $\mathbf{0.971 \pm 0.010}$ | 1011.7 | 25.1 |
| $\mathcal{C}_2$ | $0.688 \pm 0.046$ | 1116.4 | 17.3 | $0.869 \pm 0.061$ | 1116.4 | 17.3 |
| $\mathcal{C}_3$ | $0.637 \pm 0.084$ | 1189.9 | 11.9 | $0.884 \pm 0.056$ | 1189.9 | 11.9 |
| $\mathcal{C}_4$ | $0.807 \pm 0.053$ | 1242.6 | 8.0 | $0.897 \pm 0.029$ | 1242.6 | 8.0 |
| $\mathcal{C}_5$ | $0.839 \pm 0.016$ | 1275.5 | 5.6 | $0.887 \pm 0.027$ | 1275.5 | 5.6 |
| $\mathcal{C}_8$ | $0.584 \pm 0.045$ | 1299.4 | 3.8 | $0.645 \pm 0.023$ | 1299.4 | 3.8 |
| $\mathcal{C}_{10}$ | $0.568 \pm 0.025$ | 1299.8 | 3.8 | $0.633 \pm 0.014$ | 1299.8 | 3.8 |
| $\mathcal{C}_{12}$ | $0.569 \pm 0.021$ | 1299.9 | 3.7 | $0.626 \pm 0.022$ | 1299.9 | 3.7 |
| $\mathcal{C}_{14}$ | $0.562 \pm 0.016$ | 1299.9 | 3.7 | $0.625 \pm 0.013$ | 1299.9 | 3.7 |
| $\mathcal{C}_{16}$ | $0.563 \pm 0.024$ | 1299.9 | 3.7 | $0.634 \pm 0.021$ | 1299.9 | 3.7 |
| $\mathcal{C}_{18}$ | $0.563 \pm 0.024$ | 1299.9 | 3.7 | $0.634 \pm 0.021$ | 1299.9 | 3.7 |
| MP | $0.524 \pm 0.011$ | 1350.5 | 0.0 | $0.629 \pm 0.102$ | 1350.5 | 0.0 |

Table 6: Results for the NSTEPSUM dataset. By $\mathsf{C}_h$ is denoted the (caterpillar height) parameter $h$ in $\mathbb{N}$ of efficient aggregation (ours), while MP denotes the full message-passing GCN. We report mean validation accuracy with standard deviation of different 10 splits. The model of 18 layers was trained for 10 and 50 epochs. The best results are highlighted in bold. The columns "#n." denotes number of *nodes* of the computation graph, and columns "%s." percent of nodes of the computation graph *saved* comparing to message-passing (MP).

occurrences (multiple graph branches to sum over multiple numbers) and vertex labels (for larger number systems than binary), e.g. in line with the experimental contributions of Alon and Yahav (2021).

## D.4 EXPERIMENTAL SETTING ON REAL-WORLD DATASETS

We performed empirical experiments across multiple standard graph datasets, categorized by their evaluation metrics. Accuracy-based evaluation was used for bioinformatics and social network datasets, including the TUDataset (Morris et al., 2020) benchmarks such as ENZYMES, MUTAG, PROTEINS, COLLAB, and IMDB-BINARY (see Table 7). We report running times for both preprocessing and training (see Table 8). For chemical property prediction, we evaluated performance using mean squared error (MSE) on regression tasks from MoleculeNet (Paszke et al., 2019) (ESOL, FreeSolv, Lipo) and the ZINC dataset (Irwin et al., 2012) (see Table 9). Training code, along with exact hyperparameter configurations for reproducibility, is available in the supplementary material.

For each dataset, we trained graph convolutional networks (GCNs) with variants of our proposed model Caterpillar GCN. To compare the number of saved nodes consistently, for all experiments, we *fixed the number of layers to five* ($L = 5$), gradient clipping to 1.0, and based on the validation set performance, we employed early stopping with a fixed `patience` parameter of 20 to prevent overfitting. We conducted extensive k-fold cross-validation (typically 10 folds), ensuring robust performance evaluation. For ZINC dataset, we always used 5 random seed initializations for the public splits. Models were trained with standard settings, using the Adam optimizer (Kingma and Ba, 2017), moderate learning rate, and weight decay to balance training stability and convergence speed. Complete hyperparameter details are provided in the supplementary material. Experiments were repeated using fixed random seeds to ensure reproducibility. All models were trained using an Intel Xeon E5-2690 @ 2.90 GHz processor (16 cores, 32 threads, 20 MB L3 cache, 64 B cache line) equipped with 64 GB of RAM.

## D.5 EFFICIENCY OF AGGREGATION

The efficient aggregation (EA) we propose significantly lowers training complexity relative to full message-passing (MP) graph neural networks when the number of hidden channels grows, without necessarily sacrificing predictive performance. Experiments on accuracy-based classification datasets (Table 7) and regression-based datasets evaluated by mean absolute error (Table 9) demonstrate that using lower-height caterpillar aggregations substantially reduces the size of computation graphs. That is, up to approximately 93.8% fewer nodes compared to MP for *unattributed* dataset IMDB-BINARY, and 38.1% fewer nodes for *categorically attributed* dataset such as PROTEINS. This reduction may translate into improved computational efficiency, memory usage, and scalability. In our experiments without additional extensive hyperparameter optimization, we observed even a positive impact on prediction accuracy or regression performance across datasets. These results underscore the practical

| type | MUTAG | | | PROTEINS | | | ENZYMES | | | IMDB-BINARY | | | COLLAB | | |
|---|---|---|---|---|---|---|---|---|---|---|---|---|---|---|---|
| | mean ± std | #n. | %s. | mean ± std | #n. | %s. | mean ± std | #n. | %s. | mean ± std | #n. | %s. | mean ± std | #n. | %s. |
| $\mathcal{C}_0$ | 0.851 ± 0.091 | 23.7 | 67.4 | 0.649 ± 0.127 | 28.0 | 82.2 | 0.288 ± 0.030 | 30.6 | 76.6 | **0.740 ± 0.029** | 5.0 | 93.8 | 0.647 ± 0.035 | 114.7 | 61.6 |
| $\mathcal{C}_1$ | 0.856 ± 0.082 | 38.4 | 47.2 | **0.774 ± 0.034** | 97.4 | 38.1 | 0.317 ± 0.035 | 88.2 | 32.7 | 0.699 ± 0.036 | 19.6 | 75.6 | 0.647 ± 0.035 | 114.7 | 61.6 |
| $\mathcal{C}_2$ | **0.899 ± 0.050** | 47.7 | 34.4 | 0.747 ± 0.045 | 127.3 | 19.0 | 0.333 ± 0.042 | 108.6 | 17.1 | 0.705 ± 0.037 | 20.4 | 74.5 | 0.652 ± 0.042 | 130.1 | 56.5 |
| $\mathcal{C}_3$ | 0.846 ± 0.093 | 55.0 | 24.4 | 0.739 ± 0.046 | 138.2 | 12.1 | 0.360 ± 0.042 | 114.7 | 12.5 | 0.697 ± 0.043 | 20.5 | 74.5 | 0.649 ± 0.032 | 130.4 | 56.4 |
| $\mathcal{C}_4$ | 0.835 ± 0.103 | 58.4 | 19.7 | 0.738 ± 0.045 | 140.6 | 10.6 | 0.373 ± 0.045 | 116.0 | 11.5 | 0.697 ± 0.043 | 20.5 | 74.5 | 0.649 ± 0.037 | 130.4 | 56.4 |
| $\mathcal{C}_5$ | 0.861 ± 0.098 | 59.6 | 18.1 | 0.751 ± 0.045 | 141.5 | 10.0 | 0.373 ± 0.042 | 116.5 | 11.0 | 0.697 ± 0.043 | 20.5 | 74.5 | 0.651 ± 0.035 | 130.4 | 56.4 |
| $\mathcal{C}_6$ | 0.862 ± 0.071 | 60.1 | 17.4 | 0.739 ± 0.052 | 142.0 | 9.7 | 0.363 ± 0.015 | 116.9 | 10.8 | 0.697 ± 0.043 | 20.5 | 74.5 | 0.651 ± 0.035 | 130.4 | 56.4 |
| $\mathcal{C}_7$ | 0.862 ± 0.083 | 60.3 | 17.1 | 0.751 ± 0.047 | 142.2 | 9.5 | 0.377 ± 0.033 | 117.1 | 10.6 | 0.697 ± 0.043 | 20.5 | 74.5 | 0.651 ± 0.035 | 130.4 | 56.4 |
| $\mathcal{C}_8$ | 0.851 ± 0.076 | 60.3 | 17.0 | 0.748 ± 0.052 | 142.3 | 9.5 | **0.398 ± 0.037** | 117.2 | 10.5 | 0.697 ± 0.043 | 20.5 | 74.5 | 0.651 ± 0.035 | 130.4 | 56.4 |
| $\mathcal{C}_9$ | 0.862 ± 0.076 | 60.4 | 17.0 | 0.740 ± 0.047 | 142.4 | 9.4 | 0.368 ± 0.043 | 117.4 | 10.4 | 0.697 ± 0.043 | 20.5 | 74.5 | 0.651 ± 0.035 | 130.4 | 56.4 |
| $\mathcal{C}_{10}$ | 0.856 ± 0.087 | 60.4 | 17.0 | 0.745 ± 0.047 | 142.5 | 9.4 | 0.390 ± 0.043 | 117.4 | 10.4 | 0.697 ± 0.043 | 20.5 | 74.5 | 0.651 ± 0.035 | 130.4 | 56.4 |
| MP | 0.776 ± 0.100 | 72.7 | 0.0 | 0.742 ± 0.033 | 157.2 | 0.0 | 0.377 ± 0.037 | 131.0 | 0.0 | 0.640 ± 0.053 | 80.1 | 0.0 | **0.687 ± 0.043** | 299.0 | 0.0 |

Table 7: Results for *graph-level classification* datasets. By $\mathsf{C}_h$ is denoted the (caterpillar height) parameter $h$ in $\mathbb{N}$ of efficient aggregation (ours), while $\mathsf{MP}$ denotes the full message-passing GCN. We report mean validation accuracy with standard deviation of different 10 splits. Columns "#n." denote number of *nodes* of the computation graph, and columns "%s." percent of nodes of the computation graph *saved* comparing to message-passing ($\mathsf{MP}$).

| | MUTAG | | PROTEINS | | ENZYMES | | IMDB-BINARY | | COLLAB | |
|---|---|---|---|---|---|---|---|---|---|---|
| | features: 7 graphs: 188 | | features: 3 graphs: 1113 | | features: 3 graphs: 600 | | features: 0 graphs: 1000 | | features: 0 graphs: 5000 | |
| | avg. $\|V\|$: 17.9 avg. $\|E\|$: 39.6 | | avg. $\|V\|$: 39.1 avg. $\|E\|$: 145.6 | | avg. $\|V\|$: 32.6 avg. $\|E\|$: 124.3 | | avg. $\|V\|$: 19.8 avg. $\|E\|$: 193.1 | | avg. $\|V\|$: 74.5 avg. $\|E\|$: 4914.4 | |
| | parameters: 15.0k | | parameters: 53.4k | | parameters: 14.4k | | parameters: 49.6k | | parameters: 66.3k | |
| type | precomp. | avg. training | precomp. | avg. training | precomp. | avg. training | precomp. | avg. training | precomp. | avg. training |
| $\mathsf{C}_0$ | 2.98 s | 42.53 s | 13.21 s | 2.90 min | 5.99 s | 1.17 min | 7.64 s | 1.80 min | 1.82 min | 11.40 min |
| $\mathsf{C}_1$ | 6.87 s | 38.95 s | 1.52 min | 3.15 min | 48.33 s | 55.14 s | 1.21 min | 1.81 min | 100.51 min | 11.70 min |
| $\mathsf{C}_2$ | 10.83 s | 43.59 s | 2.93 min | 2.25 min | 1.36 min | 59.12 s | 1.64 min | 2.04 min | 155.25 min | 12.08 min |
| $\mathsf{C}_3$ | 15.19 s | 41.88 s | 3.91 min | 2.27 min | 1.75 min | 1.07 min | 1.66 min | 2.07 min | 198.42 min | 14.59 min |
| $\mathsf{C}_4$ | 18.85 s | 40.59 s | 4.40 min | 2.57 min | 1.96 min | 1.07 min | 1.81 min | 2.07 min | 220.47 min | 11.88 min |
| $\mathsf{C}_5$ | 20.82 s | 42.31 s | 4.75 min | 2.48 min | 2.09 min | 1.05 min | 1.66 min | 2.11 min | 220.70 min | 13.01 min |
| $\mathsf{C}_6$ | 22.23 s | 41.46 s | 4.96 min | 2.44 min | 2.14 min | 1.18 min | 1.65 min | 2.06 min | 220.22 min | 12.54 min |
| MP | — | 35.96 s | — | 2.49 min | — | 1.08 min | — | 1.59 min | — | 27.78 min |

Table 8: *Running times across datasets.* By $\mathsf{C}_h$ is denoted the (caterpillar height) parameter $h$ in $\mathbb{N}$ of efficient aggregation (ours), while $\mathsf{MP}$ denotes the full message-passing GCN. Columns "precomp" report the preprocessing time for the computation of efficient aggregation matrices, and columns "avg. training" report the average training time accross multiple runs.

value and scalability potential of our efficient aggregation method in graph-based machine learning tasks.

| type | FreeSolv | | | Lipo | | | ESOL | | | ZINC | | |
|---|---|---|---|---|---|---|---|---|---|---|---|---|
| | mean ± std | #n. | %s. | mean ± std | #n. | %s. | mean ± std | #n. | %s. | mean ± std | #n. | %s. |
| $\mathcal{C}_0$ | 2.693 ± 0.223 | 24.3 | 32.5 | 1.116 ± 0.108 | 77.3 | 29.1 | 1.664 ± 0.106 | 35.8 | 34.0 | 0.570 ± 0.012 | 65.7 | 29.8 |
| $\mathcal{C}_1$ | 2.693 ± 0.223 | 24.3 | 32.5 | 1.116 ± 0.108 | 77.3 | 29.1 | 1.664 ± 0.106 | 35.8 | 34.0 | 0.571 ± 0.006 | 65.7 | 29.8 |
| $\mathcal{C}_2$ | 2.704 ± 0.237 | 26.5 | 26.5 | 1.069 ± 0.043 | 87.0 | 20.3 | 1.666 ± 0.212 | 40.1 | 26.0 | 0.584 ± 0.007 | 75.9 | 18.9 |
| $\mathcal{C}_3$ | 2.576 ± 0.283 | 27.5 | 23.7 | 1.105 ± 0.098 | 93.7 | 14.1 | 1.516 ± 0.100 | 42.6 | 21.4 | 0.602 ± 0.003 | 82.5 | 11.9 |
| $\mathcal{C}_4$ | 2.630 ± 0.350 | 27.8 | 22.8 | 1.051 ± 0.079 | 95.8 | 12.3 | 1.410 ± 0.152 | 43.4 | 20.0 | 0.570 ± 0.012 | 84.6 | 9.6 |
| $\mathcal{C}_5$ | 2.597 ± 0.178 | 27.9 | 22.6 | 1.097 ± 0.067 | 96.3 | 11.8 | 1.458 ± 0.102 | 43.6 | 19.6 | 0.551 ± 0.012 | 85.1 | 9.1 |
| $\mathcal{C}_6$ | 2.650 ± 0.266 | 27.9 | 22.6 | 1.070 ± 0.077 | 96.5 | 11.6 | 1.528 ± 0.207 | 43.6 | 19.5 | 0.546 ± 0.013 | 85.2 | 9.0 |
| $\mathcal{C}_7$ | 2.719 ± 0.206 | 27.9 | 22.6 | 1.072 ± 0.111 | 96.5 | 11.6 | 1.446 ± 0.101 | 43.7 | 19.4 | 0.539 ± 0.012 | 85.2 | 9.0 |
| $\mathcal{C}_8$ | 2.676 ± 0.196 | 27.9 | 22.6 | 1.108 ± 0.104 | 96.5 | 11.6 | **1.392 ± 0.129** | 43.7 | 19.4 | 0.543 ± 0.006 | 85.2 | 9.0 |
| $\mathcal{C}_9$ | 2.652 ± 0.284 | 27.9 | 22.6 | 1.094 ± 0.069 | 96.5 | 11.6 | 1.416 ± 0.148 | 43.7 | 19.4 | 0.538 ± 0.003 | 85.2 | 9.0 |
| $\mathcal{C}_{10}$ | 2.675 ± 0.276 | 27.9 | 22.6 | 1.063 ± 0.091 | 96.5 | 11.6 | 1.514 ± 0.142 | 43.7 | 19.4 | 0.528 ± 0.004 | 85.2 | 9.0 |
| MP | **2.550 ± 0.374** | 36.0 | 0.0 | **1.031 ± 0.059** | 109.2 | 0.0 | 1.449 ± 0.140 | 54.2 | 0.0 | **0.477 ± 0.020** | 93.6 | 0.0 |

Table 9: Results for *graph-level regression* datasets. By $\mathsf{C}_h$ is denoted the (caterpillar height) parameter $h$ in $\mathbb{N}$ of efficient aggregation (ours), while $\mathsf{MP}$ denotes the full message-passing GCN. We report validation mean absolute error (MAE) with standard deviation of different 10 splits and random seeds, in the case of ZINC of distinct 5 random seed repetitions. Columns "#n." denote number of *nodes* of the computation graph, and columns "%s." percent of nodes of the computation graph *saved* comparing to message-passing ($\mathsf{MP}$).

