# OpenReview forum: "Caterpillar GNN: Replacing Message Passing with Efficient Aggregation"
_ICLR.cc/2026/Conference — Submitted to ICLR 2026_

### Official Review · Reviewer_7fZ6 · 2025-10-25

**Soundness:** 2
**Presentation:** 2
**Contribution:** 3
**Rating:** 4
**Confidence:** 3

**Summary:**

The paper introduces Caterpillar GNNs as a novel aggregation scheme over walk–incidence matrices. The design intentionally trades some expressivity for a stronger inductive bias that can be tuned via a height parameter derived from color refinement.
Theoretically, the authors prove that homomorphism counts from generalized caterpillar graphs are exactly as expressive as counting colored walks on height-h colorings, and that their efficient aggregation recovers this power while remaining comparatively efficient.
Empirically, Caterpillar GNNs solve a topology-driven benchmark that challenges MPNNs and, on common graph-classification datasets, achieve comparable accuracy with fewer computational nodes per graph.

**Strengths:**

The work introduces original ideas on designing GNNs. The presented model processes the occurrences of colored walks in a manner that de-duplicates repeated structural patterns in the input graph, effectively reducing the size of the computational graph. This class of GNN is very distinct from prior approaches. The fine-grained characterization of expressivity below 1-WL is also an intriguing new direction for the theory of GNNs and their inductive biases. The presentation is also clear to follow.

**Weaknesses:**

I think its claimed efficiency of Caterpillar GNNs is not convincingly demonstrated, despite being presented as a major feature of the approach. The preprocessing runtime is shown to be cubic in the number of vertices, which is limiting for large graphs and far above the trivial preprocessing needed for standard message-passing GNNs.

While the number of nodes in the computational graph is one factor that determines the efficiency, the sparsity of the underlying aggregation matrices is another critical factor that is not discussed in sufficient detail. As real-world graphs often have a high degree of sparsity, message-passing GNNs can scale to large graphs with >>10K vertices on single GPUs by using optimized sparse matrix operations. The paper does not analyze the sparsity of the $C_t^A$ matrices and the compute needed when using them in the suggested aggregation scheme.

The experiments also do not provide any runtime measurements for preprocessing and model training, which would be key to establish the scaling of Caterpillar GNNs in practice. The experiments also focus entirely on small-scale graphs with at most a few hundred vertices.

I do think the presented class of GNNs and the expressivity results are novel and interesting, but the unconvincing efficiency claims are a significant weakness as the paper makes efficiency a key part of the motivation.

**Questions:**

1. For the experiments on real-world datasets, what is the end-2-end runtime of training on each dataset when including the pre-processing time for obtaining the canonical walk subsets?
2. How sparse are the matrices $C_t^A$ compared to the adjacency matrix $A$?
3. Can the suggested walk scheme handle edge-features or continuous node features, which are both common on real-world graph learning tasks?

---

> ### Author Response · Authors · 2025-11-20
>
> We thank the reviewer for their comments, for highlighting the originality of our structural approach that lead us to a distinct class of GNNs, and for their positive recognition of a new theoretical direction established through the expressivity analysis.
>
>
>
> We appreciate the comments on complexity, which we addressed directly in the revision:
>
> - We **improved the analysis** of the preprocessing step, making it independent of $|\Sigma|\le n$ and reducing the expensive subroutines to optimized QR-decomposition. At the same time this improves by over a factor $n^{1.5}$ upon the previous asymptotic bound and aligns with matrix multiplication.
>
> We thank the reviewer for constructive and helpful direction concerning the sparsity of matrices.
> - We **added** analysis and comparison of the **sparsity** (absolute and relative to that of $A$) of the constructed matrices (Sec C.2).
> We **assessed** the asymptotic implications of the measure (Sec C.3).
>
> **Question 1**
>
> We **added report** (Tab. 7) that includes preprocessing times.
> In our current set of experiments, the preprocessing is comparable to a single training of our model.
>
> That being said, our method targets graph-level tasks, where graphs are not typically as large as for node-level tasks,
> making current QR-decomposition-based preprocessing likely to be manageable.
>
>
>
> **Question 2**
>
> As noted above, we **accessed** an additional **study** for 5 layers and datasets TUDataset, MoleculeNet, and ZINC.
> The average ratio of edges in $C^A_t$ to $A$, i.e., $\operatorname{nnz}(C^A_t)/\operatorname{nnz}(A)$, is typically below 1.0 (fewer entries of $C^A_t$ on average) or appears to be bounded by a very small constant.
> Next, we see that $C^A_t$ are typically denser, placing similar number of entries in smaller matrices.
> In the case of height=0, this density often reaches 50%, suggesting dense tensor processing would be preferable in such a setting.
> For more details, we refer to Sec. C.2.
>
>
> **Question 3**
>
>
> Within the presented theory and given a moderate preprocessing time, we would suggest for features $\xi\colon E \to \Omega$ to subdivide edges to get a graph with $\hat{\chi} \colon (E\cup V) \to (\Sigma \cup \Omega)$. We note that this scheme would contain additional words, e.g. $\chi(v_1) \xi(v_1 v_2) \chi(v_1)$, containing “half edges”.
> In case these are undesired, we recommend a more formal assessment, such as using directed edges (as answered in the Q2 of reviewer iegB).
>
> We would like to point out that continuous node features are supported, yet nodes might be more likely uniquely identified, aligning the computation to that of MP (Proposition 3.3.)

---

> > ### Comment · Reviewer_7fZ6 · 2025-11-27
> >
> > I thank the authors for the thorough response and additions to the manuscript, which address my main concerns.
> >
> > I have increased my rating accordingly.

---

### Official Review · Reviewer_iegB · 2025-10-28

**Soundness:** 3
**Presentation:** 3
**Contribution:** 3
**Rating:** 8
**Confidence:** 3

**Summary:**

Caterpillar graphs are a subclass of tree graphs and thus computing homomorphisms from caterpillar graphs is strictly less powerful than using all trees.
Nevertheless, the paper shows that this reduced expressivity is often enough and introduces an architecture that works by counting the occurences of colored walks in the input graph which in practice is as powerful as using standard message-passing.
The paper focuses on a clean and thorough introduction of the method, its modifications to make it efficient (including proofs) ,and its connection to existing work in expressiveness research for graphs.

**Strengths:**

A fundamentally new architecture, an indicator that more expressiveness is not everything (at least for many situations), rigorous theoretic derivation of expressiveness properties and connections to homomorphism counting.

**Weaknesses:**

The experiments could have been more extensive. This includes both runtime (completely missing) as well as additional datasets. E.g. Table 3 in the appendix makes it look like the caterpillar-GNN achieves SOTA results which is clearly not the case.

Details:
- Figure 2: Just looking at the figure makes it hard to guess what is happening here, even including the caption did not help too much. It would be nice to state that on top of the table its "all possible walk patterns" and possibly to visualize the example given in the caption about two walks "gb" ending in vertex 3 by showing those two walks and connecting it to the corresponding position
- 191: its sounds odd to talk about walks of color $\textbf a$, maybe "color pattern" or just "pattern"?
- 206: it would be nice to state that the overall matrix $W$ will then be reduced to $|V|\cdot t$ (from $|V|^t$)
- 231: what does this paragraph aim for? Or rather, what part about the MPGNN aggregation are you unhappy about and thus suggest to change it?
- 239: I find this whole paragraph on EA pretty hard to parse and would have liked a bit more high-level information on what the paragraph is about. What is "t-th efficient"? Is it possible to add "(e.g. A and I from MP above)" to the first sentence introducing M?
- 358: would it be possible to clarify which aspect of fig 1 is important in this context? I think this could be more clear.
- Fig 7: the datasets feel a little outdated and the MPNN performance tends to be rather low (even though the description makes clear that some effort was put into it).

**Questions:**

1) how much capacity for tuning the practical part of the method do you see?
2) What are the main obstacles against extending the method to directed graphs?
3) Given the low performance on ZINC (which is about counting cycles), do you expect it to perform well in real-world tasks in general? Especially as many of those rely on small cycles. Would it be possible to also run experiments on e.g. MolPCBA and OGBN-Arxiv? (as examples of somewhat more recent datasets)

---

> ### Author Response · Authors · 2025-11-20
>
> We thank the reviewer for their assessment and for recognizing the theoretical and architectural novelty of our approach.
>
>
> We are grateful for the noted suggestions, which we addressed in revision:
> - improved Figure 2, and we want to improve it further;
> - 191: we changed to “colored walk”;
> - 206: we added this statement;
> - 231: here, we are not willing to change MP, but help the reader by relating to it operators as reviewer suggests to clarify in 239;
> - 239: added such explanation to help reader connect to the previous point;
> - 358: clarified the reference to Fig 1 and its caption;
> - Fig 7.: to support the computational-graph-based comparison,
> we needed to fix the number of layers for all datasets, we believe that per-height tuning could improve results.
>
>
> **Question 1:**
>
> In our experiments, the setting is fixed across models to enable structural comparison of computational graphs. We believe that EA offers a meaningful capacity to tune the hyperparameters, particularly per height (currently done according to MP).
> From a computational perspective, we provide a tighter bound (via reduction to QR-decompositions) improving both theoretical complexity and simplifying practical implementations.
>
>
> **Question 2:**
>
> We thank the reviewer for the question. Our arguments in Sec. A rely on adjacency symmetry (A^T = A) to guarantee restricted expressivity of efficient variants of graph matrices. To that end, using our construction of A(G, \chi) that links to general weighted automata theory, we believe that one could obtain canonization for general settings. Though the resulting method would require a more thorough assessment.
> Informally, by running EA simply on a directed graph would not break equivariance, but we lose guarantees of lower expressivity bounded by the colored walk refinement.
>
> **Question 3:**
>
> We agree that short cycles are important for molecules and that they might not be captured any better than with the standard MPGNN.
> For real-world tasks in general, our strict focus on expressivity below MP can be relaxed by, e.g., suitable positional encodings, or by running on products of input graphs.
> While OGBN-Arxiv is a node-level dataset, we are progressing preliminary experiments on OGB-MolHIV, OGB-MolPCBA, and more recent datasets. If time allows, we will be happy to share these results.

---

### Official Review · Reviewer_57dt · 2025-10-29

**Soundness:** 4
**Presentation:** 2
**Contribution:** 3
**Rating:** 6
**Confidence:** 3

**Summary:**

The authors propose EA, a mechanism to replace Message-Passing in GNNs, which uses an aggregation over walk incidence-based matrices and trades some expressive power for a stronger, more structured inductive bias. They provide proof of its expressiveness using a hierarchy of generalized caterpillar graphs and their graph homomorphism counts.

**Strengths:**

- The mechanism presented is novel and well-founded, connecting random walk-based models with message-passing GNNs.
- There is a rigorous theoretical analysis of the expressive power of the method using homomorphism counts.
- The paper contains multiple interesting theoretical insights. I also find the use of automata theory to prove results from Sec. 3 and Sec. 4 interesting.
- Different from some other random-walk-based methods, Caterpillar GNNs can preserve (permutation) equivariance and determinism, which can be desirable properties.
- The paper introduces the new nStepAddition benchmark, showcasing scenarios where Caterpillar GNNs outperform MPNNs.
- The paper is very pleasing from an aesthetic perspective; the figures and formatting are beautiful.

**Weaknesses:**

The main weaknesses of this paper are scoping and empirics:
- **Scoping:** The paper is primarily theoretical, proposing a novel aggregation mechanism along with the new Caterpillar GNNs and providing an extensive theoretical analysis. However, its presentation within the constraints of a short conference paper feels limited. While the main text introduces some of the key ideas and results, much of the theoretical novelty and discussion are deferred to the Supplementary Material. I recognize the challenge of fitting a substantial body of work into a 9-page format, but this raises the question of whether the contribution would be better suited as (1) a longer journal submission or (2) multiple, more focused conference papers with narrower scope. As it stands, the overall narrative feels somewhat diffuse, and the contributions appear scattered rather than cohesive.
* **Empirics**: The paper is notably weak in terms of empirical evaluation. The stated goal from an empirical perspective is to study the scaling properties of EA and compare the Caterpillar GNNs with MPNNs. While some results are presented in this direction (e.g., Figures 4 and 7, Appendix Tables 2–4), they remain relatively superficial and unconvincing from a practitioner’s perspective. Both MoleculeNet and TUDataset provide limited grounds for quantitatively assessing modern architectures. Moreover, the paper lacks runtime comparisons, parameter counts, or other metrics that would allow a fair assessment of computational efficiency, reporting only the number of nodes in the computation graphs. In addition, there are no comparisons with more expressive baselines such as _k_-WL-based models (e.g., GNN-SSWL [1], IDMPNN [2]) or subgraph-aware GNNs (e.g., ESAN [3]). Comparisons with Random-Walk-based methods like CRaWL [6] or NeuralWalker [7] are also missing. Finally, the evaluation omits standard expressivity benchmarks, including EXP and CSL [4], which are widely used to assess a model’s ability to capture higher-order structural patterns.

[1]: https://proceedings.mlr.press/v202/zhang23k/zhang23k.pdf
[2]: https://proceedings.mlr.press/v202/zhou23n/zhou23n.pdf
[3]: https://arxiv.org/pdf/2110.02910
[4]: https://arxiv.org/pdf/2010.01179
[6]: https://arxiv.org/abs/2102.08786
[7]: https://arxiv.org/pdf/2406.03386

**Questions:**

Please see weaknesses above.

I find the core idea of the paper appealing, and I believe the theoretical contributions are both interesting and valuable for certain subcommunities. However, the presentation of the theoretical ideas could be more focused, or alternatively, the paper could benefit from a more comprehensive empirical analysis. I am somewhat ambivalent overall but lean toward accepting the paper primarily for its theoretical merits. That said, I believe there is substantial room for improvement, and I would also find a rejection reasonable.

---

> ### Author Response · Authors · 2025-11-20
>
> We thank the reviewer for their thoughtful, balanced feedback and their kind words.
> In particular, we are encouraged by the recognition of novelty and the soundness of the structural approach to the design of graph learning models.
>
> The reviewer mentions that the paper carries a broad theoretical scope. In revision, we took steps to improve the presentation of the theoretical ideas. To further emphasize the contribution, we **added a proof sketch** for Theorem 4.2. While part of this work could naturally be extended in future research, we believe that the current combination of results is essential to present a coherent contribution.
>
> We are grateful for the reviewers' suggestions regarding empirical depth.
> - we **added report** of runtime, preprocessing time, and parameters;
> - we **included density measures** to further empirically assess the computational graphs (Table 2, Sec. C.2).
>
> We believe that for practitioners our work offers a convenient and controlled parametrization towards lower expressivity by changing the computational graphs, without relying on, e.g., new layer types complicating deployment.
> While the reviewer mentions comparisons to more expressive baselines, we primarily focus on standard message-passing GNNs, since message-passing component is present in all of the referenced methods.
> A broader quantitative study across diverse architectures would be an interesting next step.
> We want to clarify that we do not include EXP or CSL benchmarks, as they are designed to test expressivity strictly above that of MPGNNs, and are therefore not informative for evaluating architectures of expressivity strictly below that of MPGNNs.
> To compensate for this absence of benchmarks, we introduce the nStepAddition benchmark in Sec. 5.1 guided by the walk-incidence-based topology.

---

> > ### Comment · Reviewer_57dt · 2025-11-25
> >
> > I thank the authors for their response. I still believe that the theoretical contributions can be interesting for certain readers, however the experimental evaluation is still very limited. My suggestion was not to only expand it (only) to datasets such as EXP, but more broadly to other (practical) datasets too.
> >
> > I will maintain my score and confidence.

---

> > > ### Author Response · Authors · 2025-12-03
> > >
> > > We thank the reviewer for their follow-up, for their acknowledgement of the theoretical contributions.
> > > The reviewer clarifies their suggestion to expand the empirical evaluation, not necessarily towards higher-order, but more broadly, which they mentioned in the context of providing grounds for a quantitative assessment of modern architectures.
> > >
> > > In the second revision, we therefore expanded Section C beyond the original 10 datasets
> > > by including the large MalNet-Tiny dataset and five additional datasets (Peptides-func, Peptides-struct, ogbg-molhiv, ogbg-molpcba, ogbg-code2).
> > > For these practical datasets, we adopt the configuration of a modern GCN-backbone architecture [1], as detailed in Section C.4.
> > >
> > > We hope this broadening addresses the limitations of the empirical section, without further diffusing the theoretical focus of our work.
> > >
> > >
> > > [1] Can Classic GNNs Be Strong Baselines for Graph-level Tasks? Simple Architectures Meet Excellence, ICML 2025

---

### Official Review · Reviewer_8Yfw · 2025-10-30

**Soundness:** 3
**Presentation:** 1
**Contribution:** 2
**Rating:** 2
**Confidence:** 3

**Summary:**

This paper presents Caterpillar GNN, which augments the message passing process to incorporate node-walk incidence matrices derived from random walks. Because the number of random walks increases exponentially in the length of the walks, the authors propose a deterministic walk selection approach, which limits the number of random walks of length 1 to T to n, where n is the number of graph nodes and T is the maximum length of the random walks. Using a least-squares based approximation, the selected walks B_t are mapped into new matrices C_t that are used instead of the adjacency matrices during the message passing operations. The authors choose the random walks in a deterministic and permutation equivariant manner.

**Strengths:**

- The approach appears to be novel and incorporating important insights from graph theory and automata theory fields.

- Deterministic and permutation-equivariant walk selection adds a principled structural constraint.

**Weaknesses:**

1) Poor organisation and clarity

- The exposition is dense and introduces key definitions late, which makes it difficult to follow the main ideas.

- The first three pages (i.e., from the beginning until Section 3) introduce the terminology and concepts used in the paper.  However, a formal definition of “caterpillar”, a fundamental concept of this submission, does not appear until page 6 in Section 4. On page 1, the authors refer to caterpillars simply as a subclass of trees, without providing any further details.

- In Section 1, an informal theorem (Theorem 4.1) is illustrated by a figure (Figure 1). However, neither a proof is provided nor there is an explanation for the figure. As a result, the relationship between colored walks and caterpillars remains unclear.

- In Section 2, an unnecessarily complex and unconventional notation is used to describe the way MPGNNs operate.

- The proof of the arguably the most important theorem of the paper (Theorem 4.2) is given in the appendix with minimal explanation and by citing some other Lemmas given in the appendix.

2) Efficiency and scalability issues

- The method presented is expensive despite the complexity reduction referred to as efficient aggregation. With sufficiently discriminative node or edge features, the number of colors will be in the order of the graph nodes. The complexity of walk selection will then be O(n^4) according to Theorem 3.1, where n is the number of graph nodes.

3) Limited experimental validation

- Experiments are sparse and inconclusive. It would help if the authors provided a comparison or ablation showing why deterministic selection outperforms stochastic alternatives.

**Questions:**

- Why is the deterministic, permutation-equivariant walk selection introduced in Section 3.1 considered the best possible choice?

- How does Caterpillar GNN compare, in tractability and expressiveness, to simpler alternatives such as using powers of the adjacency matrix?

---

> ### Author Response · Authors · 2025-11-20
>
> We thank the reviewer for their thorough assessment, for their questions, and for recognition of the novelty of our approach and its grounding in graph theory and automata theory.
> We address the points in order below.
>
>
> _Organization and clarity:_
>
> For the given advancement, we use more rigour than can be found as usual; therefore, we appreciate the concerns raised regarding organization, for which we made concrete changes:
>
> - we **added** the formal and simple **definition** of caterpillars right into the introduction;
> - we **detailed the caption of Fig. 1**, and described components of caterpillar;
> - we **added a reference** below the informal theorem (Theorem 4.1) to a (simpler) proof of its example case.
> - we **added sketch of the proof of Theorem 4.2** to give readers more intuition.
> - to navigate the structure of the proof, which we agree takes a major part of Section A,
> we **will add** a diagram of dependencies.
>
>
> _Complexity of precomputation:_
>
> Our method improves $O(Tn^4)$ upon $O(n^T)$ when $n$ is the number of nodes and $T$ the length limit. Moreover, we **revisited and improved** our analysis to give an even tighter bound $O(Tn^{2.372})$ in Thm 3.1. Our proof is constructive, and the theoretical coefficient is derived by reducing the potentially least-scalable parts to QR-decomposition, which is an extensively optimized algorithm.
>
>
>
>
> **Question 1**
>
> Our main contribution is implying that stochastic walk-based non-equivariant methods have a deterministic equivariant counterpart, which moreover does not come with exponential asymptotics. The walk selection introduced in Sec. 3.1, we would not exactly call the “best possible”, but for the column space of $W_t$, it induces a minimal and spanning set of columns. This property is precisely what ensures that EA is exactly as expressive as the colored walk refinement, as shown by Thm. 4.2.
>
>
> **Question 2**
>
> We would like to point out that the adjacency matrix approach is only viable for unlabeled cases, i.e., where vertices do not carry labels or features. Our approach matches in expressiveness(*) and tractability(**) in this simple setting by taking height=0, but we present much more: For a labeled/vertex-colored graph on input (with vertex-coloring other than trivial), our method with height=0 is already more expressive. Moreover, for height>0, it is always strictly more expressive, see diagram "Expressivity Hierarchy" in Sec. 4.
>
> *) Simplified arguments are given for each direction Observation 3.2, and Lemma A.12 in the Appendix, respectively.
>
> **) Algorithm 1 reduces to adjacency multiplication.

---

> > ### Comment · Reviewer_8Yfw · 2025-11-23
> >
> > Thank you for the clarifications, which address some of my concerns regarding scalability and the lack of a sketch of the proof of Theorem 4.2. However, my concerns about the quality of the writing and the limited scope of the experimental evaluation remain.
> >
> > 1. **Example of unclear or sloppy writing.**
> >    The revised manuscript includes the following definition of a *caterpillar*:
> >
> > > “A tree graph is a caterpillar if removing its leaves yields a path called spine.”
> >
> > Here, the authors introduce new terminology without providing a clear definition. What precisely is meant by “spine”? Is it simply a path? If so, why introduce a new term instead of using the standard one? As written, the definition raises unnecessary ambiguity.
> >
> > 2. **Further example of unclear writing.**
> >    The definition of Theorem 4.1 in the revision includes the sentence:
> >
> > > “See Figure 1, and the proof (B.4) of this case.”
> >
> > It is unclear what “(B.4)” refers to. Should the reader consult Appendix B.4 (which does not exist), Theorem B.4 in Appendix B, or the proof of Theorem 4.1 in Appendix B.1? Such ambiguities significantly hinder readability.
> >
> > 3. **Inconsistent numbering of theorems.**
> >    Theorem 4.1 appears in Section 1, whereas Theorem 3.1 appears later in Section 3. This inconsistency makes the paper difficult to navigate and suggests that the manuscript has not been carefully edited.
> >
> > 4. **Comments on the appendix.**
> >    The appendix is quite long (~25–30 pages), with roughly half dedicated to automata theory and quantum automata. It is not clear how much of this material is novel. If a substantial portion is original, it should be reviewed by appropriate domain experts, potentially published separately, and then cited in the main paper. If the majority is not original, the authors should include only the genuinely novel content and cite the rest rather than reproducing it in full.

---

> ### Author Response · Authors · 2025-11-24
>
> We thank the reviewer for their response. We are pleased that some of their concerns have been addressed.
>
> _Experimental evaluation_
>
> The reviewer suggests an ablation study on an alternative stochastic walk selection. As briefly noted in our previous response, the walk selection (introduced in Sec. 3.1) ensures that each submatrix $B_t$ is a basis for the column space of $W_t$, hence guaranteeing full rank as required by Eq. 2. Moreover, the selection is prefix-closed and can therefore be computed without additional overhead.
>
> While we agree that there may be meaningful stochastic walk selections, incorporating them would require extending the approach beyond its present scope rather than simplifying it.
> To that end, in Sec 5.2, we compare multiple walk selection approaches by varying the height parameter (Sec 3.3), which significantly impacts the walk selection process (Obs 3.2, Prop 3.3), and affects the resulting computational graphs, as illustrated in Fig. 7.
>
> _1-3. Surroundings of the informal version of Thm 4.1  in the introduction_
>
> We will refine the introduction of caterpillars to meet the requested level of detail by removing the “spine” term and adding the label for the requested proof reference in the next revision. Statements are currently numbered by section when stated formally (to help the reader navigate to that section).
>
>
> _4. Comments on the appendix_
>
> Appendix A (\~10 pages, including the improved analysis of Thm 3.1) uses concepts from (weighted) automata theory for the derivation of our method, and Appendix B (\~6 pages) uses graph (homomorphism) theory for the analysis of our method.
>
> For the derivation of efficient aggregation (EA) and Theorems 4.2 and 3.1, we start with the standard definitions of weighted automata. Since our automata (capturing walks in the graphs) are not general, we then hold stronger assumptions, and thus we can obtain significantly improved results and more desired properties (e.g., per-layer matrices) than using plain automata theory. These targeted results allow for the definition of EA without automata theory in the main text, purely as a self-standing parametrization over computational graphs. The improved analysis of Thm 3.1 is another such example.
>
> For Theorem 4.1, we develop novel combinatorial arguments in graph theory, further extending widely used techniques of labeled graphs (c.f. Paragraph 2 of Sec 6). These results provide an initial understanding (characterization) of the inductive biases below message-passing; thus, serving as the motivating insight, as well as the tool of well-established homomorphism expressivity analysis for GNNs.
> For (strict) separations in the “Expressivity hierarchy” of Sec 4, we prove a single property (Prop B.10) of our graph classes that allows us to (almost directly) apply the results of other works (as we clarify explicitly in the main text and appendix).
>
> In the Appendix, we recall existing results and definitions with explicit and transparent references without reproducing the proof itself (to the best of our knowledge). Additionally, in cases where our concept adapts an existing construction, we aim to clarify it explicitly and transparently, e.g., Algorithm 1 as a “layered” adaptation of weighted automata minimization. When appropriate, in addition to the original reference, we provide a reference to a summarizing literature to improve accessibility.
> In special cases (Thm B.4 and Thm 4.1, or Lemma 12 and Lemma 14), we first state and prove a simpler version of the result relying on more elementary techniques.
> We will further improve clarity, e.g., by adding diagrams to appendix.
> We respectfully clarify that our paper does not concern quantum automata.

---

### Official Review · Reviewer_V12x · 2025-10-31

**Soundness:** 2
**Presentation:** 1
**Contribution:** 2
**Rating:** 2
**Confidence:** 4

**Summary:**

This paper proposes sacrificing some expressivity to achieve more efficient computation. I believe it makes substantive advances in both ideas and theory; however, objectively, it needs better organization and polishing, and it should provide more convincing experiments.

**Strengths:**

Relaxing strict graph isomorphism tests to obtain more efficient computation is an interesting idea that could help the graph learning community strike a better balance between theory and practice.

**Weaknesses:**

* To be fair, the paper’s clarity could be much improved; I spent considerable time ensuring I understood the main ideas. I recognize that a theory-heavy paper cannot be fully popularized, but there are concrete steps that would markedly raise readability. For example:

  * Provide a glossary at the start of the appendix. It may be long in this case, but it is necessary.
  * “Caterpillar graphs” are not formally defined until the end of page 6, yet the term appears frequently before that. The same issue applies to Theorem 4.1. I suggest reorganizing the paper so that Section 4 appears earlier in the main text.
  * Section 3 walks readers through the derivation of EFFICIENT AGGREGATION (EA). I am not convinced these details are needed in the main text for all readers; they may fit better in the appendix.
  * Add a table or relationship diagram clarifying the contributions of all theorems/lemmas/corollaries in the main text and appendix, and how they relate to each other.
  * What kind of benchmark is “NSTEPADDITION”? Is it introduced by this paper, or reported previously? Please cite prior work or provide a more detailed description in the appendix.

* The trade-off between expressivity and computational efficiency is a long-standing challenge in graph learning, so the idea of “EFFICIENT AGGREGATION” is appealing. Unfortunately, I do not see the paper emphasizing this advantage with sufficient evidence. Why is EA “efficient”? Readers will want to know how much computational improvement Caterpillar GNNs achieve over prior GNNs known for high expressivity. Expanding Table 1 to compare both expressivity and complexity would help.

* The experiments are few and confusing.

  * In Section 5.1, what exactly does “bottleneck” mean? The NSTEPADDITION benchmark is also described vaguely. What are the precise MP settings used for comparison? Is the conclusion “more expressivity hurts” restricted to NSTEPADDITION and Caterpillar GNNs, or is it general?
  * In Section 5.2, I do not see the rationale for using the average number of nodes in the computation graph on the x-axis. Computational cost (or efficiency) is not linearly tied to the number of nodes. I suggest replacing it with the height of the Caterpillar GNN. Also, §5.2 lacks comparisons against other highly expressive GNNs (and even MP), which makes it hard to substantiate the advantages of Caterpillar GNNs.
  * Experiments are conducted only on the (almost) smallest graph-classification datasets, with no larger datasets or richer task settings.
  * In the appendix, Table 4 shows MP almost always outperforming Caterpillar GNNs across settings. This raises doubts about the practical utility of Caterpillar GNNs in real-world scenarios.
  * Regarding the authors’ claim of efficient aggregation, the paper should also report the comparison of time and memory consumption with other GNNs.

**Questions:**

See weakness

---

> ### Author Response · Authors · 2025-11-20
>
> We thank the reviewer for their careful analysis and for highlighting the conceptual value of the proposed relaxation of the graph isomorphism test (color refinement) to obtain more assumptions and control over inductive biases used in graph learning.
> We address the noted sections towards improvements in the corresponding parts. For clarity, submit our comment in two parts, repeating key points where necessary.
>
> _Readibility:_
> We appreciate concrete steps towards readability improvements, which we implemented in the revision.
>
> - **Added** a one-page glossary at the beginning of the appendix
>
> - Caterpillars are **now defined directly** in the introduction and further emphasized below the Fig. 1.
>
> - We agree that streamlining the results of Section 4 earlier is desired; yet we wanted to at least give readers the key technical ideas underlying our approach. The actual complete derivation is already postponed to Appendix A. We went over Section 3 and simplified as much as possible.
> - We **will add** a diagram explaining the contributions and their relationship of theorems/lemmas/corollaries to the appendix.
>
> - Benchmark nStepAddition is part of our contributions and is indeed introduced in this paper; the formal rationale is detailed in the first part of Appendix D.
>
> _Trade-off:_
> We appreciate the reviewers' recognition of the benefits of tradeoffs following from balanced expressivity.
> - We respectfully note that a tradeoffs that paper directly concerns are
> between expressivity and stronger more structured inductive bias (015, Sec. 5.1) and between predictive performance and the (subsequent) number of nodes in the computation graphs (021, 089, Sec. 5.2).
>
> >  Why is EA “efficient”?
>
> - Sequential patterns are important component of some recent graph learning models.
> As a practical consequence, graph learning methods until now aggregate sequential patterns in a graph naively by naive O(n^T) enumeration which is highly inefficient (or do subsampling). To emphasize this contribution of our improvement to polynomial-time, we **improved explanation** in the introduction, and we **added** an explanatory sketch of the proof of Theorem 4.2. We **added Sec C.3** discussing complete asymptotics for this matter.
>
> - To that end, given the discovered characterization of lower expressivity of Thm 4.1, also noted in Table 1, EA is the first known polynomial canonical form that achieves expressivity of caterpillar homomorphisms (also known as a graph class of pathwidth at most 1). This also implies polynomial-time for the decision problem of this relaxed isomorphism of a pair of graphs (homomorphism indistinguishability), which was shown to be polynomial only recently [1] using finite fields.

---

> > ### Author Response · Authors · 2025-11-20
> >
> > (the second part)
> >
> > _Experiments:_
> > We are grateful for the list of points to clarify, which we are going to address in order:
> >
> > In Section 5.1:
> >
> > > what exactly does “bottleneck” mean
> >
> > * With a bottleneck, we refer to an over-squashing detailed in a long line of work, e.g. [2], that ties it to the undesired properties of topology on the graph.
> >
> > > ...benchmark is also described vaguely. What are the precise MP settings ...
> >
> > * A formal description of nStepAddition is provided in Sec D.2, including an example in Figure 13. MP is GCN, settings for the experiment are given in Sec D.3 as well as implementation in the supplementaries. That being said, we find it important that the depicted descent occur already for Caterpillar GNNs with higher height, not only for MP.
> >
> >
> > > Is the conclusion “more expressivity hurts” restricted to NSTEPADDITION and Caterpillar GNNs, or is it general?
> >
> > * Description “Figure 6: NSTEPADDITION: more expressivity hurts”, is only referring to nStepAddition.
> >
> > In Section 5.2:
> >
> > > Computational cost (or efficiency) is not linearly tied to the number of nodes.
> >
> > * As the number of channels $d$ increases, the number of nodes is tied to $d^2$ linearly. We **added clarifying paragraph, Eq. 30** to Sec C.3 for formal assessment. For the regime with constant $d$, we **added measures** to Sec C.2 along with **additional empirical measurements**.
> >
> >
> > > I suggest replacing it with the height of the Caterpillar GNN. Also, §5.2 lacks comparisons against other highly expressive GNNs (and even MP), which makes it hard to substantiate the advantages of Caterpillar GNNs.
> >
> > * We understand the suggestion and respectfully note that the height and MP for comparison is already explicit in Figure 7.
> >
> > > Experiments are conducted only on the (almost) smallest graph-classification datasets, with no larger datasets or richer task settings.
> >
> > - Used datasets demonstrate rich data-specific behavior. To communicate this compactly to readers of a theory-heavy paper, we deliberately limited experimental settings to maintain clarity in a theory-focused paper. Experiments show that (1) height changes/consolidates the computational graphs significantly on those datasets. And (2) these changes do not obstruct learning through backpropagation, moreover, allow to keep the usual hyperparameter setting, i.e. replace MP with EA.
> >
> > > Table 4 shows MP almost always outperforming Caterpillar GNNs across settings. This raises doubts about the practical utility of Caterpillar GNNs in real-world scenarios.
> >
> > - We believe that Table 4 reflects how our method allows deliberately restrict its expressivity. Following our theoretical insights, more expressive methods are better suited for such datasets (e.g. ZINC).
> >
> >
> > > Regarding the authors’ claim of efficient aggregation, the paper should also report the comparison of time and memory consumption with other GNNs.
> >
> > * We **added report** of time comparison (Sec D.4, Table 8) for experiments in Sec 5.2, moderate memory consumption is by the reported hardware.
> >
> >
> >
> >
> >
> >
> > We hope that these changes address the reviewer's concerns and improve the clarity of our work.
> >
> >
> > References:
> >
> > [1] An Algorithmic Meta Theorem for Homomorphism Indistinguishability, MFCS 2024
> >
> > [2] Understanding over-squashing and bottlenecks on graphs via curvature, ICLR 2022

---

### Author Response · Authors · 2025-11-21

We thank all reviewers for their great efforts in reviewing our paper, for their constructive comments and questions, and for their recognition of the theoretical and conceptual advances of our contributions.

- In response to concrete helpful suggestions on readability and descriptive clarity, we revised corresponding parts, highlighted the corresponding changes directly in the manuscript, and noted/detailed/or explained them in our comments below each review. We added a glossary and a table of contents to the appendix.

- To address concerns regarding the complexity of precomputation, we refined the theoretical analysis, tightening the asymptotic bound from $O(Tn^4)$ to $O(Tn^{2.372})$, building upon the earlier reduction from exponential enumeration. The extended proof is now included in the appendix.

- We substantially expanded Appendix C with additional measures of the computational graphs of Caterpillar GNNs to answer Question 2 of reviewer 7fZ6 and complemented subsequent analysis, refined by empirical estimates based on our datasets. We also report preprocessing times, average training times, with parameter counts used in Figure 7.

---

### Author Response · Authors · 2025-12-03

We thank the reviewers for their responses and suggestions. In the _second revision_, we address the remaining concerns as follows.

- _Empirical analysis._ We now also include the large graph-level dataset MalNet-Tiny (up to 5,000 nodes per graph; 1,410 on average) and five additional datasets from LRGB and OGB. For structural comparison, we adopt the settings of a recent strong GCN-based message-passing architecture (Section C.4).
- _Clarity in the introduction._ We polished the surroundings of Theorem 4.1 and Figure 1, fine-tuning the level of detail in the definitions.
- _Navigation of the results._ We added relationship diagrams to Appendix A and Appendix B (along with further clarifications) to help readers follow the contributions.

---

### Meta-Review · Area_Chair_aLGB · 2026-01-07

**Summary:**

This submission introduces Caterpillar GNNs and an “efficient aggregation” mechanism based on deterministically selected colored walks / walk-incidence matrices, motivated by trading expressivity for a structured inductive bias and improved efficiency. Reviewers agreed the theoretical direction is interesting, but the decision should be rejection because the paper does not convincingly support its central efficiency and practicality claims. The main drivers were: unclear and hard-to-follow presentation (key definitions and theorem structure appear late, heavy reliance on appendix), limited and confusing experiments on mostly small datasets, missing or insufficiently motivating baselines, and a lack of end-to-end evidence that the approach is actually efficient in realistic regimes (runtime, memory, sparsity-aware scaling). While one reviewer gave a very high score, that review was comparatively brief and did not engage deeply with the major empirical and efficiency shortcomings raised by others, so it should carry less weight in the final decision.

**Reviewer Concerns:**

The authors made good efforts to improve clarity and navigation by moving key definitions earlier, adding a glossary, expanding figure captions, and providing proof sketches and additional explanations in the appendix. They also clarified the scope of some claims, gave more detail on the synthetic benchmark, and added runtime, preprocessing, and sparsity statistics to better explain what they mean by “efficiency.

Efficiency, however, is still not convincingly demonstrated in realistic settings: reductions in computational graph size do not clearly translate to end-to-end speed or memory gains, especially given the costs of preprocessing and dense operations. The experimental evaluation remains limited, focusing primarily on small datasets and niche benchmarks, with missing or weak baselines. Presentation problems persist, as noted explicitly by a reviewer who responded after the rebuttal and maintained concerns about writing quality and experimental scope.

**Reviewer Scores:**

The two reviewers who recommended rejection would likely keep their scores unchanged. While some improvements were acknowledged, neither the efficiency claims nor the empirical weaknesses were resolved, and one reviewer explicitly reaffirmed their concerns after the rebuttal.

The borderline-positive reviewer would likely keep the same score. They acknowledged the theoretical interest but reiterated that the empirical evaluation remains too limited, and they maintained their rating in follow-up.

The very positive reviewer would likely remain positive, but this review is comparatively brief and does not engage more with the main weaknesses raised by others. Overall, the score distribution would remain largely unchanged, indicating a clear rejection.

---

### Decision · Program_Chairs · 2026-01-26

Reject